# Provable Meta-Learning with Low-Rank Adaptations

**Jacob L. Block**
UT Austin
jblock@utexas.edu

**Sundararajan Srinivasan**
UT Austin
sundararajans@utexas.edu

**Liam Collins**
Snap, Inc.
lcollins2@snapchat.com

**Aryan Mokhtari**
UT Austin & Google Research
mokhtari@austin.utexas.edu

**Sanjay Shakkottai**
UT Austin
sanjay.shakkottai@utexas.edu

## Abstract

The power of *foundation models* (FMs) lies in their capacity to learn highly expressive representations that can be adapted to a broad spectrum of tasks. However, these pretrained models require additional training stages to become effective for downstream applications. In the multi-task setting, prior works have shown empirically that specific meta-learning approaches for preparing a model for future adaptation through parameter-efficient fine-tuning (PEFT) can outperform standard retraining methods, but the mechanism of the benefits of meta-learning has been largely unexplored. We introduce a framework for generic PEFT-based meta-learning to learn a model that can easily adapt to unseen tasks. For linear models using LoRA, we show that standard retraining is provably suboptimal for finding an adaptable set of parameters and provide strict performance guarantees for our proposed method. We verify these theoretical insights through experiments on synthetic data as well as real-data vision and language tasks. We observe significant performance benefits using a simple implementation of our proposed meta-learning scheme during retraining relative to the conventional approach.

## 1 Introduction

*Foundation Models* (FMs) learn rich representations that are useful for a variety of downstream tasks. The first stage of FM training is referred to as *pretraining*, where a combination of massive public, proprietary, and synthetic sources of data is used to learn a general-purpose model from scratch [1–4]. However, due to the enormous cost of training state-of-the-art models on such large datasets, pretraining is largely infeasible for most. Thus, the most popular and viable way to utilize FMs for specific applications is to adapt an existing pretrained model.

We consider this problem of adapting a pretrained FM to a set of related tasks. We refer to this as *retraining*, where given a number of tasks with many samples, our goal is to recover a model that learns the task structure and can be quickly adapted to future tasks with limited samples. In other works this has been referred to as pre-finetuning [5] or supervised fine-tuning [6]. After retraining, we adapt the model to a new task with few samples in what we denote the *fine-tuning* stage. In this last stage, we typically employ parameter-efficient fine-tuning (PEFT) methods – training heuristics which sacrifice learning expressiveness for improved computational efficiency [7, 8]. Ultimately, the purpose of retraining is to prepare the model for efficient future adaptation, and the effectiveness of a retraining method is measured by the model's performance on the fine-tuning task.

Standard approaches to retraining involve fitting the model to the aggregation of the different retraining tasks. While this seems reasonable and has been empirically successful [9, 10], it does not leverage knowledge of the downstream fine-tuning procedure to tailor the retrained model to perform well after

such adaptation. Rather, it retrains the model to minimize the average loss across the retraining tasks regardless of the PEFT method to be employed later. Thus, there is no assurance that the recovered solution is indeed adaptable to future unseen tasks relative to other possible retraining solutions.

*Meta-learning* is a natural framework to address this issue, as it explicitly aims to learn adaptable models, typically in low-resource, few-shot settings using gradient-based adaptations [11, 12]. The success of meta-learning algorithms is largely attributed to their ability to learn useful representations, as even model-agnostic gradient-based meta-learning algorithms like MAML [11] and Reptile [13] have been shown to implicitly learn representations in linear settings [14, 15]. Recent works have shown empirical benefits to retraining using specific PEFT-based meta-learning methods [16–20], but theoretical guarantees showcasing these gains have not been established.

**Contributions.** We propose a framework for PEFT-based meta-learning during retraining which explicitly accounts for the specific PEFT method that will be used for future fine-tuning. While our framework is general to any PEFT method, we focus on Low-Rank Adaptation (LoRA) [7] for our theoretical results. Specifically, we consider multiple linear regression tasks where each ground truth regressor is a rank-$k$ perturbation of a common matrix $\boldsymbol{A}^* \in \mathbb{R}^{d \times d}$, where $d$ is the ambient dimension. Given a set of $T$ tasks, our goal is to recover $\boldsymbol{A}^*$ so that we can easily fine-tune to a new, unseen task by learning a low-rank perturbation using LoRA. With this in mind, we show the following:

- We prove that standard retraining, which does not incorporate any knowledge of the PEFT method for fine-tuning, fails to recover parameters that are low-rank adaptable, as the recovered model is not low-rank away from $\boldsymbol{A}^*$ and consequently the test model (Proposition 1). As a result, applying low-rank adaptations cannot account for the large rank discrepancy to the test task (Proposition 2). Further, fine-tuning with a very large rank to account for this discrepancy defeats the purpose of PEFT and results in squared prediction error which grows as $\mathcal{O}\left(\frac{d \min\{kT, d\}}{n}\right)$ with high probability, where $n$ is the number of test samples (Remark 2). Thus, standard retraining performs *worse* when given access to more tasks.

- For our meta-learning framework, we guarantee that any minimizer of the meta-learning loss in the infinite sample case is low-rank adaptable to unseen tasks (Theorem 2). Further, we show that if there are at least three retraining tasks, the ground truth parameters are the unique global minima up to orthogonal symmetry (Theorem 3). As a result, LoRA fine-tuning is effective in adapting to the test task and with high probability achieves squared prediction error which grows as $\mathcal{O}\left(\frac{kd}{n}\right)$ (Corollary 4). In contrast to standard retraining, we achieve the optimal rate which does not include any dependence on $T$.

- We prove in the infinite sample case, every second-order stationary point of our meta-learning loss when applied to two retraining tasks is in fact globally optimal (Theorem 4). In this case there are no spurious local minima of our meta-learning loss and optimality is completely determined by second-order information. Thus, local optimization methods like perturbed gradient descent can efficiently find global minima.

To the author's knowledge, these are the first results proving that PEFT-based meta-learning outperforms standard retraining methods in any setting. Proofs are presented in Appendix B. To verify our theoretical insights, we compare the performance of the standard retraining and LoRA-based meta-learning objectives in a synthetic multi-output linear regression setting. We show clear improvements using LoRA-based meta-learning for all data parameter settings. Then, we apply an implementation of our general PEFT-based meta-learning framework to a vision task on CIFAR-10 [21] as well as a natural language experiment using the RoBERTa [22] language model on the ConvAI2 [23] dataset with two PEFT schemes: LoRA and last layer fine-tuning. In both cases, we observe that PEFT-based meta-learning outperforms standard retraining.

## 1.1 Related Work

Meta-learning is a technique for learning models that can be rapidly adapted to unseen tasks by leveraging access to prior tasks during training. For example, Model-Agnostic Meta-Learning (MAML) [11] and Reptile [13] are popular methods that aim to find a model that can be adapted to a new task after a small number of steps of gradient descent on the new task's loss function.

Prior analyses of gradient-based meta-learning for linear models consider learning a shared feature subspace across tasks [14, 24, 15]. This reduces to a subspace recovery problem, as each task reveals

a linear measurement of this shared space. In contrast, we study tasks that are low-rank perturbations of a shared arbitrary matrix—a novel setting with distinct challenges. For instance, in our setting just three retraining tasks suffice to exactly recover the ground-truth parameters (Theorem 3), whereas subspace-based approaches require the number of tasks to exceed the subspace dimension. These differences motivate new analytical tools and lead to qualitatively different results.

Many meta-learning approaches specific to FMs for incorporating PEFT-based adaptation have been proposed. Hong and Jang [17], Bansal et al. [18], and Gheini et al. [19] applied meta-learning with architecture adaptations that inject task-specific trainable layers within the FM architecture. Hou et al. [16] combined architecture adaptations with parameter perturbations similar to LoRA. They considered a complicated loss that updates the adapters and FM weights over different splits of the data and showed empirical gains over standard retraining and other gradient-based MAML-style algorithms. Aghajanyan et al. [5] proposed a multi-task objective that trains an FM on different tasks simultaneously to encourage learning a universally applicable representation. It forces the FM to learn a shared data representation but allows for task-specific prediction heads. Overall, these works each proposed a meta-learning or multi-task objective and showed empirical gains over standard retraining strategies. However, their experimental results motivate a deeper theoretical exploration of when standard retraining is insufficient relative to meta-learning approaches, how many tasks are needed to learn a rich representation, and how to best adapt to tasks unseen in the training stage.

Lastly, although we focus on LoRA, different PEFT methods have been proposed, including variants of LoRA [25–27] and architecture adaptations [28] among others. Further, recent works have analyzed the theoretical aspects of LoRA in the fine-tuning stage [29, 30], but they explored orthogonal directions to the analysis of LoRA-based meta-learning during retraining. Extended discussion of these prior works is in Appendix A.

**Notation.** We use bold capital letters for matrices and bold lowercase letters for vectors. $\mathcal{N}(\boldsymbol{\mu}, \boldsymbol{\Sigma})$ refers to the multivariate Gaussian distribution with mean $\boldsymbol{\mu}$ and covariance matrix $\boldsymbol{\Sigma}$. $\boldsymbol{I}_d$ refers to the $d \times d$ identity matrix. $\|\cdot\|_F$ refers to the Frobenius norm. $S_d$ refers to the set of $d \times d$ symmetric matrices, and $S_d^+$ is the set of $d \times d$ positive semi-definite matrices. $O_d$ refers to the set of $d \times d$ orthogonal matrices. $[n]$ refers to the set $\{1, \ldots, n\}$. For a matrix $\boldsymbol{X} \in \mathbb{R}^{m \times n}$, $\mathrm{im}(\boldsymbol{X})$ and $\ker(\boldsymbol{X})$ refer to the image and kernel of $\boldsymbol{X}$, while $\mathrm{vec}(\boldsymbol{X}) \in \mathbb{R}^{mn}$ denotes the column-wise vectorization of $\boldsymbol{X}$. For subspaces $M, N$, $\dim(M)$ refers to the dimension of $M$ and $M + N = \{\boldsymbol{x} + \boldsymbol{y} | \boldsymbol{x} \in M, \boldsymbol{y} \in N\}$. If $M \cap N = \{\boldsymbol{0}\}$, we write the direct sum $M \oplus N$.

## 2 Retraining and Fine-Tuning Schemes

We briefly recap the optimization process for standard retraining of an FM across multiple tasks followed by fine-tuning on a downstream task. We then introduce a general framework for PEFT-based meta-learning which adjusts the retraining phase to incorporate insights from fine-tuning.

### 2.1 Standard Retraining Then Fine-Tuning

Consider a collection of $T$ tasks of interest $\mathcal{T} = \{\mathcal{T}_t\}_{t=1}^T$ where each task $\mathcal{T}_t$ is drawn from task distribution $\mathcal{D}$ and consists of $n_t$ labeled examples $\mathcal{T}_t = \{(\boldsymbol{x}_{t,j}, \boldsymbol{y}_{t,j})\}_{j=1}^{n_t}$. Without loss of generality, we assume consistent dimensions across tasks, so $\boldsymbol{x}_{t,j} \in \mathbb{R}^{d_x}$, $\boldsymbol{y}_{t,j} \in \mathbb{R}^{d_y}$ for all tasks $t$ and sample indices $j$. Let $\boldsymbol{X}_t \in \mathbb{R}^{d_x \times n_t}$ and $\boldsymbol{Y}_t \in \mathbb{R}^{d_y \times n_t}$ denote the concatenation of the respective input samples and labels from task $t$, and consider a model $\Phi(\cdot ; \boldsymbol{W}) : \mathbb{R}^{d_x} \to \mathbb{R}^{d_y}$ parameterized by weights $\boldsymbol{W}$ that maps feature vectors to predicted labels. We abuse notation and write $\Phi(\boldsymbol{X}_t ; \boldsymbol{W})$ to denote the concatenation of $\Phi(\boldsymbol{x}_{t,j} ; \boldsymbol{W})$ for $j \in [n_t]$. Typically $\boldsymbol{W} = (\boldsymbol{W}_1, \ldots, \boldsymbol{W}_m)$ is a list of matrices where $\boldsymbol{W}_i \in \mathbb{R}^{d \times d}$ parameterizes the $i^{\text{th}}$ layer of a neural network. We assume each $\boldsymbol{W}_i$ is square for convenience.

**Retraining Phase.** Given a loss function $\mathcal{L}$, standard retraining attempts to minimize the aggregated loss over a collection of training tasks [22, 2]. This amounts to solving

$$\hat{\boldsymbol{W}}_{\text{SR}} = \min_{\boldsymbol{W}} \sum_{t=1}^T \mathcal{L}\left(\Phi(\boldsymbol{X}_t; \boldsymbol{W}), \boldsymbol{Y}_t\right), \tag{1}$$

where SR stands for Standard Retraining. The above optimization problem seeks a set of universal parameters that define a unique mapping function capable of translating inputs to outputs across all tasks involved in the retraining phase. We denote the corresponding model as $\Phi(\,\cdot\,;\hat{\boldsymbol{W}}_{\text{SR}})$.

**Fine-Tuning Phase.** In the subsequent fine-tuning, PEFT is used to refine the model to fit a downstream task with fewer labeled samples. Formally, consider an unseen task $\mathcal{T}_{T+1}$ drawn from the same task distribution $\mathcal{D}$. We define PEFT as any method which fits the model to task $\mathcal{T}_{T+1}$ by fixing $\boldsymbol{W} = \hat{\boldsymbol{W}}_{\text{SR}}$ in the original parameterization and fine-tuning the mapping $\Phi(\,\cdot\,;\hat{\boldsymbol{W}}_{\text{SR}})$ using additional parameters $\boldsymbol{\theta}$. For example, $\boldsymbol{\theta}$ could parameterize trainable perturbations of $\hat{\boldsymbol{W}}_{\text{SR}}$ or new trainable layers inserted into the architecture of the retrained model [7, 25, 5]. We denote the fine-tuned model as $\Phi_{\text{FT}}(\,\cdot\,;\hat{\boldsymbol{W}}_{\text{SR}},\boldsymbol{\theta}):\mathbb{R}^{d_x} \to \mathbb{R}^{d_y}$ and again abuse notation by writing $\Phi_{\text{FT}}(\boldsymbol{X}_{T+1}\,;\hat{\boldsymbol{W}}_{\text{SR}},\boldsymbol{\theta})$ to denote the concatenation of $\Phi_{\text{FT}}(\boldsymbol{x}_{T+1,j}\,;\hat{\boldsymbol{W}}_{\text{SR}},\boldsymbol{\theta})$ for each $j \in [n_{T+1}]$. During the *fine-tuning stage*, the goal is to find the optimal additional parameters, $\boldsymbol{\theta}$, that minimize the loss for the downstream task $\mathcal{T}_{T+1}$, solving

$$\min_{\boldsymbol{\theta}} \mathcal{L}(\Phi_{\text{FT}}(\boldsymbol{X}_{T+1}\,;\hat{\boldsymbol{W}}_{\text{SR}},\boldsymbol{\theta}),\boldsymbol{Y}_{T+1}). \tag{2}$$

In particular, when LoRA is used for fine-tuning, the model is adapted to task $\mathcal{T}_{T+1}$ by fixing the architecture and the retrained weights $\hat{\boldsymbol{W}}_{\text{SR}}$ and only training low-rank perturbations for each of the matrices $\hat{\boldsymbol{W}}_{\text{SR},1},\dots,\hat{\boldsymbol{W}}_{\text{SR},m}$. For rank-$r$ adaptations, we parameterize $\boldsymbol{\theta} = ((\boldsymbol{Q}_1,\boldsymbol{V}_1),\dots,(\boldsymbol{Q}_m,\boldsymbol{V}_m))$, where $\boldsymbol{Q}_i,\boldsymbol{V}_i \in \mathbb{R}^{d\times r}$ are the factors of the low-rank adaptation of the $i^{\text{th}}$ matrix in $\hat{\boldsymbol{W}}_{\text{SR}}$. The fine-tuned model is just the original model where the $i^{\text{th}}$ weight matrix $\boldsymbol{W}_i$ is now perturbed to be $\boldsymbol{W}_i + \boldsymbol{Q}_i\boldsymbol{V}_i^{\top}$. For $\boldsymbol{Q},\boldsymbol{V} \in (\mathbb{R}^{d\times r})^m$, define the LoRA loss

$$\mathcal{L}_{\text{LoRA}}(\boldsymbol{Q},\boldsymbol{V}\,;\boldsymbol{W}) = \mathcal{L}\left(\Phi\left(\boldsymbol{X}_{T+1}\,;\left(\boldsymbol{W}_i + \boldsymbol{Q}_i\boldsymbol{V}_i^{\top}\right)_{i=1}^{m}\right),\boldsymbol{Y}_{T+1}\right). \tag{3}$$

The LoRA fine-tuning optimization problem is then

$$\min_{\boldsymbol{Q},\boldsymbol{V}} \mathcal{L}_{\text{LoRA}}(\boldsymbol{Q},\boldsymbol{V}\,;\hat{\boldsymbol{W}}_{\text{SR}}). \tag{4}$$

This pipeline seems reasonable as we first fit the model to the aggregation of the retraining tasks which we hope will promote learning the general structure of the tasks drawn from $\mathcal{D}$. However, nothing about standard retraining promotes learning an adaptable solution relative to other candidate solutions that fit the retraining tasks. Next, we introduce a general meta-learning framework which explicitly incorporates the adaptation mechanism during retraining.

## 2.2 PEFT-Based Meta-Learning

Since the ultimate goal of retraining is to perform well on an unseen downstream task, we study a general PEFT-based meta-learning (PEFT-ML) objective that explicitly fits weights and adapter parameters to the training tasks. Rather than training a single model on the aggregation of the retraining tasks, we instead incorporate the adapters during the retraining process and learn adapted models for each task. Let $\boldsymbol{\theta}^{(t)}$ be the set of adapter parameters for the $t^{\text{th}}$ training task $\mathcal{T}_t$. The PEFT-ML objective searches for a single set of base weights $\hat{\boldsymbol{W}}_{\text{Meta}}$ such that for all $t \in [T]$, the $t^{\text{th}}$ adapted model $\Phi_{\text{FT}}(\,\cdot\,;\hat{\boldsymbol{W}}_{\text{Meta}},\boldsymbol{\theta}^{(t)})$ minimizes the loss over the training task $\mathcal{T}_t$. More precisely, we define the proposed PEFT-ML objective as

$$\hat{\boldsymbol{W}}_{\text{Meta}} = \min_{\boldsymbol{W}} \sum_{t=1}^{T} \mathcal{L}_t(\boldsymbol{W}), \tag{5}$$

where $\mathcal{L}_t(\boldsymbol{W})$ denotes the optimal loss on task $t$ after fine-tuning:

$$\mathcal{L}_t(\boldsymbol{W}) = \min_{\boldsymbol{\theta}^{(t)}} \mathcal{L}\left(\Phi_{\text{FT}}\left(\boldsymbol{X}_t\,;\boldsymbol{W},\boldsymbol{\theta}^{(t)}\right),\boldsymbol{Y}_t\right).$$

When we use LoRA as the adaptation method, we define $\boldsymbol{Q}^{(t)},\boldsymbol{V}^{(t)} \in (\mathbb{R}^{d\times r})^m$ as the list of factors of the low-rank adapter $\boldsymbol{Q}_i^{(t)}(\boldsymbol{V}_i^{(t)})^{\top}$ applied to the $i^{\text{th}}$ weight matrix for the $t^{\text{th}}$ task. Then the inner objective $\mathcal{L}_t(\boldsymbol{W})$ reduces to

$$\mathcal{L}_t(\boldsymbol{W}) = \min_{\boldsymbol{Q}^{(t)},\boldsymbol{V}^{(t)}} \mathcal{L}_{\text{LoRA}}(\boldsymbol{Q}^{(t)},\boldsymbol{V}^{(t)}\,;\boldsymbol{W}). \tag{6}$$

In this case, we refer to the objective function in (5) as LoRA-ML. This proposed optimization problem is designed to replace the standard retraining objective in (1). After solving (5) we recover base parameters $\hat{\boldsymbol{W}}_{\text{Meta}}$ that are explicitly designed to be adaptable to downstream tasks drawn from the same distribution as those seen in retraining. To perform finetuning, we then run the exact same minimization in (2) but using retrained weights $\hat{\boldsymbol{W}}_{\text{Meta}}$ instead of $\hat{\boldsymbol{W}}_{\text{SR}}$.

## 3 Main Results

To establish our theoretical results, we consider $T \geq 1$ multi-output linear regression retraining tasks $\{\mathcal{T}_t\}_{t=1}^T$ and one downstream test task $\mathcal{T}_{T+1}$, where the ground-truth regressor for each task is a low-rank perturbation of a common shared matrix. Precisely, for each $t \in [T+1]$ we assume task $\mathcal{T}_t$ is independently drawn from distribution $\mathcal{D}_{\boldsymbol{A}^*}$ which is associated with some arbitrary fixed matrix $\boldsymbol{A}^* \in \mathbb{R}^{d \times d}$ and intrinsic adaptation rank $k \ll d$. Each task $\mathcal{T}_t \sim \mathcal{D}_{\boldsymbol{A}^*}$ is parameterized by the shared matrix $\boldsymbol{A}^*$ and a task-specific rank-$r$ matrix $\boldsymbol{R}_t^*$ such that the samples $(\boldsymbol{x}_{t,j}, \boldsymbol{y}_{t,j}) \in \mathbb{R}^d \times \mathbb{R}^d$ from $\mathcal{T}_t$ are related by the noisy linear transformation

$$\boldsymbol{y}_{t,j} = (\boldsymbol{A}^* + \boldsymbol{R}_t^*)\boldsymbol{x}_{t,j} + \boldsymbol{\epsilon}_{t,j},$$

where $\boldsymbol{\epsilon}_{t,j}$ are independently generated noise terms $\boldsymbol{\epsilon}_{t,j} \sim \mathcal{N}(\boldsymbol{0}, \sigma_\epsilon^2 \boldsymbol{I}_d)$. We assume the inputs $\boldsymbol{x}_{t,j}$ are i.i.d. with zero-mean $\mathbb{E}\left[\boldsymbol{x}_{t,j}\right] = \boldsymbol{0}$ and covariance $\mathbb{E}\left[\boldsymbol{x}_{t,j}\boldsymbol{x}_{t,j}^\top\right] = \sigma_x^2 \boldsymbol{I}_d$. Lastly, we generate the ground truth rank-$r$ adaptations $\boldsymbol{R}_t^*$ as the symmetric outer product of random Gaussian factors $\boldsymbol{U}_t^*$:

$$\boldsymbol{R}_t^* = \boldsymbol{U}_t^* \boldsymbol{U}_t^{*\top} \quad \text{s.t.} \quad \text{vec}(\boldsymbol{U}_t^*) \sim \mathcal{N}(\boldsymbol{0}, \boldsymbol{I}_{dk}).$$

The above generative model defines the input-output relationships for each task as similar linear models, differing from each other only by a low-rank perturbation $\boldsymbol{R}_t^*$. We construct the adapters $\boldsymbol{R}_t^*$ as symmetric for convenience of analysis, but note that the limitations of standard retraining which we demonstrate in Section 3.1 also hold for general adapters (Appendix C).

**Remark 1.** *For convenience, we require a mild sense of task diversity and assume that any collection of $r \leq d$ number of columns chosen from the set of all columns of $\{\boldsymbol{U}_t^*\}_{t=1}^{T+1}$ are linearly independent. We note that our generative model for each $\boldsymbol{U}_t^*$ ensures this assumption holds with probability 1.*

The learner uses the linear model $\Phi(\boldsymbol{x}; \boldsymbol{A}) = \boldsymbol{A}\boldsymbol{x}$ for $\boldsymbol{A} \in \mathbb{R}^{d \times d}$ and retrains on tasks $\mathcal{T}_1, \ldots, \mathcal{T}_T$ with the ultimate goal of efficient adaptation to $\mathcal{T}_{T+1}$ using LoRA. The aim is to recover the parameter value $\hat{\boldsymbol{A}} = \boldsymbol{A}^*$ in the retraining phase so that the fine-tuned model $\Phi_{\text{FT}}(\boldsymbol{x}; \hat{\boldsymbol{A}}, \boldsymbol{Q}, \boldsymbol{V}) = (\hat{\boldsymbol{A}} + \boldsymbol{Q}\boldsymbol{V}^\top)\boldsymbol{x}$ fits the data distribution of any downstream task also drawn from $\mathcal{D}$ for proper rank-$k$ adapter $\boldsymbol{Q}\boldsymbol{V}^\top$.

We define the finite-sample loss function for task $t$ as

$$\mathcal{L}_t^{n_t}(\boldsymbol{A}) = \frac{1}{2n_t} \sum_{j=1}^{n_t} \|\boldsymbol{y}_{t,j} - \boldsymbol{A}\boldsymbol{x}_{t,j}\|_2^2, \tag{7}$$

and we define $\mathcal{L}_t^*(\boldsymbol{A})$ as the shifted and scaled infinite sample loss:

$$\mathcal{L}_t^*(\boldsymbol{A}) = \frac{1}{\sigma_x^2}\left(\mathbb{E}_{\boldsymbol{x},\boldsymbol{y}}\left[\mathcal{L}_t^{n_t}(\boldsymbol{A})\right] - \frac{\sigma_\epsilon^2}{2}\right) = \frac{1}{2}\left\|\boldsymbol{A}^* + \boldsymbol{U}_t^*\boldsymbol{U}_t^{*\top} - \boldsymbol{A}\right\|_F^2. \tag{8}$$

We consider the setting where for the retraining tasks $t \leq T$ we have large $n_t$, but for the test task $n_{T+1}$ is small. This reflects practical scenarios where we have access to large retraining datasets compared to the low-resource fine-tuning task. Thus, we assume access to the infinite sample loss functions $\mathcal{L}_t^*$ for the retraining tasks $t \leq T$. Then, for ease of notation, define $n = n_{T+1}$ as the number of test task samples. We ultimately aim to use LoRA to fit the finite-sample test task loss $\mathcal{L}_{T+1}^n$ efficiently in $n$. Given a learned representation $\hat{\boldsymbol{A}} \in \mathbb{R}^{d \times d}$ from retraining, the fine-tuning problem using LoRA with rank $r$ reduces to

$$\min_{\boldsymbol{Q},\boldsymbol{V} \in \mathbb{R}^{d \times r}} \mathcal{L}_{T+1}^n(\hat{\boldsymbol{A}} + \boldsymbol{Q}\boldsymbol{V}^\top) \tag{9}$$

Since $\boldsymbol{Q}\boldsymbol{V}^\top$ can parameterize any rank-$r$ matrix, (9) is a specific parametrization for what is commonly known as reduced rank regression [31]. It is clear that to even realize the optimal regressor $\boldsymbol{A}^* - \hat{\boldsymbol{A}} + \boldsymbol{U}_{T+1}^*\boldsymbol{U}_{T+1}^{*\top}$ for $\boldsymbol{Q}\boldsymbol{V}^\top$, we need $r \geq \text{rank}(\boldsymbol{A}^* - \hat{\boldsymbol{A}} + \boldsymbol{U}_{T+1}^*\boldsymbol{U}_{T+1}^{*\top})$ . Further, results in reduced rank regression and matrix sensing in general reveal the importance of $\text{rank}(\boldsymbol{A}^* + \boldsymbol{U}_{T+1}^*\boldsymbol{U}_{T+1}^{*\top} - \hat{\boldsymbol{A}})$ in terms of the hardness of minimizing $\mathcal{L}_{T+1}^n(\hat{\boldsymbol{A}} + \boldsymbol{Q}\boldsymbol{V}^\top)$.

**Lemma 1** (Bunea et al., 2011). *Consider $\hat{\boldsymbol{A}} \in \mathbb{R}^{d \times d}$ and let $r = \text{rank}(\boldsymbol{A}^* + \boldsymbol{U}_{T+1}^*\boldsymbol{U}_{T+1}^{*\top} - \hat{\boldsymbol{A}})$. Let $\boldsymbol{Q}^*, \boldsymbol{V}^* \in \mathbb{R}^{d \times r}$ minimize $\mathcal{L}_{T+1}^n(\hat{\boldsymbol{A}} + \boldsymbol{Q}\boldsymbol{V}^\top)$ over all rank-$r$ factors $\boldsymbol{Q}, \boldsymbol{V} \in \mathbb{R}^{d \times r}$ and let $\boldsymbol{X}_{T+1} = [\boldsymbol{x}_{T+1,1}, \dots, \boldsymbol{x}_{T+1,n}]$ denote the matrix of test task inputs. Denote the matrix of prediction errors $\boldsymbol{E} = (\boldsymbol{A}^* + \boldsymbol{U}_{T+1}^*\boldsymbol{U}_{T+1}^{*\top})\boldsymbol{X}_{T+1} - (\hat{\boldsymbol{A}} + \boldsymbol{Q}^*\boldsymbol{V}^{*\top})\boldsymbol{X}_{T+1}$. Then $\forall \gamma > 0$,*

$$\mathbb{P}\left(\frac{1}{n}\|\boldsymbol{E}\|_F^2 \leq \frac{24(1+\gamma)^2\sigma_\epsilon^2 rd}{n} \,\middle|\, \boldsymbol{X}_{T+1}\right) \geq 1 - e^{-\gamma^2 d} \tag{10}$$

The squared prediction error scales linearly with $rd$. This matches the information-theoretic lower bound to learn $rd$ number of parameters and is minimax optimal over all rank-$r$ matrices when the singular values of $\boldsymbol{X}_{T+1}$ are uniformly bounded [33]. Thus, a larger rank of $\boldsymbol{A}^* - \hat{\boldsymbol{A}} + \boldsymbol{U}_{T+1}^*\boldsymbol{U}_{T+1}^{*\top}$ inflates the fine-tuning prediction error, as we hope to recover $\hat{\boldsymbol{A}} = \boldsymbol{A}^*$ so that $\text{rank}(\boldsymbol{A}^* - \hat{\boldsymbol{A}} + \boldsymbol{U}_{T+1}^*\boldsymbol{U}_{T+1}^{*\top}) = k$. We next compare the standard retraining (1) and LoRA-ML (6) objectives.

### 3.1 Negative Results for Standard Retraining then Fine-Tuning

Consider standard retraining then fine-tuning as a candidate for ultimately minimizing (9). The learner first finds a single matrix $\hat{\boldsymbol{A}}_{\text{SR}}$ that minimizes the sum of losses $\sum_{t=1}^T \mathcal{L}_t^*$:

$$\hat{\boldsymbol{A}}_{\text{SR}} = \arg\min_{\boldsymbol{A}} \frac{1}{2}\sum_{t=1}^T \left\|\boldsymbol{A}^* + \boldsymbol{U}_t^*\boldsymbol{U}_t^{*\top} - \boldsymbol{A}\right\|_F^2. \tag{11}$$

Then when given test task $\mathcal{T}_{T+1}$, the learner solves $\min_{\boldsymbol{Q},\boldsymbol{V}\in\mathbb{R}^{d\times r}} \mathcal{L}_{T+1}^n(\hat{\boldsymbol{A}}_{\text{SR}} + \boldsymbol{Q}\boldsymbol{V}^\top)$. However, this strategy suffers substantial loss in both the retraining and fine-tuning stages. Notice the loss in (11) is convex and quadratic in $\boldsymbol{A}$, so the first-order optimality condition shows that

$$\hat{\boldsymbol{A}}_{\text{SR}} = \boldsymbol{A}^* + \frac{1}{T}\sum_{t=1}^T \boldsymbol{U}_t^*\boldsymbol{U}_t^{*\top}. \tag{12}$$

Thus, $\hat{\boldsymbol{A}}_{\text{SR}}$ recovers $\boldsymbol{A}^*$ added to the average of the retraining ground truth adaptations $\boldsymbol{U}_t^*\boldsymbol{U}_t^{*\top}$. However, $\hat{\boldsymbol{A}}_{\text{SR}}$ performs poorly on all of the retraining tasks, as standard retraining is unable to disentangle the common structure $\boldsymbol{A}^*$ from the task-specific adapters $\boldsymbol{U}_t^*\boldsymbol{U}_t^{*\top}$.

**Theorem 1.** *Let $\boldsymbol{U}^* = (\boldsymbol{U}_1^*, \dots, \boldsymbol{U}_T^*)$. Then,*

$$\mathbb{E}_{\boldsymbol{U}^*}\left[\sum_{t=1}^T \mathcal{L}_t(\hat{\boldsymbol{A}}_{SR})\right] = (T-1)kd(d+1) = \Omega\left(Tkd^2\right)$$

Thus, $\hat{\boldsymbol{A}}_{\text{SR}}$ suffers significant loss on the retraining tasks when averaged over the generation process of ground truth parameters $\boldsymbol{U}^*$. Further, $\hat{\boldsymbol{A}}_{\text{SR}}$ is not low-rank adaptable to the test task. Crucially, the intrinsic dimension of the test task is $\text{rank}(\boldsymbol{A}^* + \boldsymbol{U}_{T+1}^*\boldsymbol{U}_{T+1}^{*\top} - \hat{\boldsymbol{A}}_{\text{SR}}) = \min\{d, k(T+1)\}$, so an adaptation rank of $\min\{d, k(T+1)\}$ is required to even achieve the ground truth test task parameters.

**Proposition 1.** *If test fine-tuning rank $r < \min\{d, k(T+1)\}$, then $\mathcal{L}_{T+1}^*(\boldsymbol{Q}, \boldsymbol{V}; \hat{\boldsymbol{A}}_{SR}) > 0$ for all rank-$r$ adapter factors $\boldsymbol{Q}, \boldsymbol{V} \in \mathbb{R}^{d\times r}$.*

Even though the test task parameters are only rank-$k$ away from $\boldsymbol{A}^*$, standard retraining fails to exploit this structure and inflates the necessary rank to $\min\{d, k(T+1)\}$. Thus, standard retraining actually recovers worse representations as the number of tasks $T$ grows. In this case, failing to fine-tune with large enough rank causes significant loss.

**Proposition 2.** *For a large number of retraining tasks $T \to \infty$ and test fine-tuning rank $r$, $\mathcal{L}_{T+1}^*(\boldsymbol{Q}, \boldsymbol{V}; \hat{\boldsymbol{A}}_{SR}) = \Omega\left((d-r)k^2\right)$ for all $\boldsymbol{Q}, \boldsymbol{V} \in \mathbb{R}^{d \times r}$.*

As the number of retraining tasks grows to infinity, the squared error between the test task recovered parameter and the ground truth is determined by the under-specification of the fine-tuning rank $r$ relative to the ambient dimension $d$.

The above propositions show the cost of under-specifying the fine-tuning rank relative to the large intrinsic dimension of the test task which results from standard retraining. Conversely, applying the necessarily large fine-tuning rank $r = \min\{d, k(T+1)\}$ both defeats the purpose of *low-rank* adaptation and still incurs large prediction error when fine-tuning with limited samples.

**Remark 2.** *Consider the finite-sample loss (9) using $\hat{\boldsymbol{A}}_{SR}$ adapted with LoRA using rank $r = \min\{d, k(T+1)\}$. This can achieve optimal population risk but suffers in the finite-sample setting. Using Lemma 1, we can only hope to achieve squared prediction error of order $\mathcal{O}\left(\frac{d \min\{kT, d\}}{n}\right)$ when fine-tuning, which is much larger than the optimal rate $\mathcal{O}\left(\frac{kd}{n}\right)$ if we had in fact recovered the ground truth $\boldsymbol{A}^*$ during retraining.*

Thus, **standard retraining recovers parameters that cannot be efficiently low-rank adapted to new tasks**. In contrast, our analysis of LoRA-ML for retraining shows much improved performance.

## 3.2 Results for LoRA-Meta-Learning

Consider applying (6) to this problem instance. We introduce low-rank adapters during the retraining phase to model the different training tasks and search for a value of $\boldsymbol{A}$ such that for all $\mathcal{T}_t$, the loss $\mathcal{L}_t^*$ after running LoRA on $\mathcal{T}_t$ is minimized. This promotes values of $\boldsymbol{A}$ that can be easily adapted to unseen tasks downstream. We use the LoRA-ML loss but with symmetric low-rank adapters $\boldsymbol{U}_t\boldsymbol{U}_t^\top$ for the $t^{\text{th}}$ task $\mathcal{T}_t$ in retraining. We still use asymmetric adapters for fine-tuning on the test task with loss $\mathcal{L}_{T+1}^n$. The LoRA-ML loss given access to infinite sample task losses $\mathcal{L}_t^*$ is then

$$\mathcal{L}_{\text{Meta}}(\boldsymbol{A}) = \sum_{t=1}^{T} \min_{\boldsymbol{U}_t} \mathcal{L}_t^*(\boldsymbol{A} + \boldsymbol{U}_t\boldsymbol{U}_t^\top). \tag{13}$$

Define the concatenation of each $\boldsymbol{U}_t$ as $\boldsymbol{U} = (\boldsymbol{U}_1, \dots, \boldsymbol{U}_T) \in \left(\mathbb{R}^{d \times k}\right)^T$. Then minimizing (13) is equivalent to solving $\min_{\boldsymbol{A}, \boldsymbol{U}} \mathcal{L}^*(\boldsymbol{A}, \boldsymbol{U})$ where

$$\mathcal{L}^*(\boldsymbol{A}, \boldsymbol{U}) = \frac{1}{2} \sum_{t=1}^{T} \left\| \boldsymbol{A}^* + \boldsymbol{U}_t^*\boldsymbol{U}_t^{*\top} - \boldsymbol{A} - \boldsymbol{U}_t\boldsymbol{U}_t^\top \right\|_F^2. \tag{14}$$

We have seen that standard retraining does not recover an optimal solution, but it is unclear what the global minima of this new objective function are and if they can be easily found. Note that by fixing $\boldsymbol{A}$, (14) is $T$ independent symmetric matrix factorization problems, and by fixing $\boldsymbol{U}$, (14) is a convex quadratic problem over $\boldsymbol{A}$. Despite these well-understood sub-problems, joint minimization over $\boldsymbol{A}$ and $\boldsymbol{U}$ presents challenging variable interactions that complicate the analysis. Nevertheless, we employ a careful landscape analysis of (14) to address these questions.

### 3.2.1 Landscape of Global Minima of (14)

We first show that the objective is well-posed, i.e., minimization of $\mathcal{L}$ leads to an adaptable solution.

**Theorem 2.** *If $\mathcal{L}^*(\hat{\boldsymbol{A}}, \hat{\boldsymbol{U}}) = 0$, then $\hat{\boldsymbol{A}} = \boldsymbol{A}^* + \boldsymbol{C}$ where $\text{rank}(\boldsymbol{C}) \leq 2k$*

Any point is a global minimum of (14) if and only if it achieves zero loss. Theorem 2 guarantees that the values of $\hat{\boldsymbol{A}}$ that achieve global minimization of (14) are at most rank-$2k$ away from the ground truth parameter $\boldsymbol{A}^*$. Then, the remaining intrinsic dimension of the test task is just $3k \ll d$.

**Corollary 1.** *If $\mathcal{L}^*(\hat{\boldsymbol{A}}, \hat{\boldsymbol{U}}) = 0$, there exists a rank-$3k$ adapter $\boldsymbol{Q}\boldsymbol{V}^\top$ such that $\mathcal{L}_{T+1}^*(\boldsymbol{Q}, \boldsymbol{V}; \hat{\boldsymbol{A}}) = 0$.*

Since the sufficient LoRA rank for fine-tuning is just $3k$, we realize a much improved fine-sample prediction error.

**Corollary 2.** *Let $\mathcal{L}^*(\hat{A}, \hat{U}) = 0$ and let $Q^*, V^* \in \mathbb{R}^{d \times 3k}$ minimize $\mathcal{L}^n_{T+1}(\hat{A} + QV^\top)$ over all $Q, V \in \mathbb{R}^{d \times 3k}$. Then, $\hat{A} + Q^* V^{*\top}$ satisfies Lemma 1 with $r = 3k$.*

Thus, retraining with LoRA-ML leads to squared prediction error on the task task which grows asymptotically as $\mathcal{O}\left(\frac{kd}{n}\right)$. Although the unnecessary factor of $T$ incurred by standard retraining is avoided when using LoRA-ML, the rate still contains an additional factor of 3 over the ideal case when $r = k$ since $A^*$ is not guaranteed to be recovered exactly. However, this minor discrepancy is mitigated when the number of tasks satisfies $T \geq 3$. In this case, exact recovery of the ground truth parameter $A^*$ is possible.

**Theorem 3.** *For any number of tasks $T \geq 3$ and ambient dimension $d \geq 3k$, if $\mathcal{L}^*(\hat{A}, \hat{U}) = 0$ then $\hat{A} = A^*$ and $U_t U_t^\top = U_t^* U_t^{*\top}$ for all $t \in [T]$*

This guarantees that the ground truth parameters are the unique global minimum up to orthogonal symmetry when there are three or more tasks, as long as the ground truth adaptation rank $k$ is relatively small compared to the ambient dimension $d$. This result is surprising, as most theoretical results for multi-task learning require stricter conditions on the number of tasks $T$, typically where $T$ is required to be larger than the effective task dimension [34, 14]. However, we establish this uniqueness result for the absolute condition $T \geq 3$. As a result, we only need a rank-$k$ adaptation to realize the test task.

**Corollary 3.** *For $T \geq 3$, if $\mathcal{L}^*(\hat{A}, \hat{U}) = 0$, then $\exists Q, V \in \mathbb{R}^{d \times k}$ such that $\mathcal{L}^*_{T+1}(Q, V ; \hat{A}) = 0$.*

We then achieve the desired fine-sample prediction error.

**Corollary 4.** *For $T \geq 3$, let $\mathcal{L}^*(\hat{A}, \hat{U}) = 0$ and let $Q^*, V^* \in \mathbb{R}^{d \times k}$ minimize $\mathcal{L}^n_{T+1}(\hat{A} + QV^\top)$ over all $Q, V \in \mathbb{R}^{d \times k}$. Then, $\hat{A} + Q^* V^{*\top}$ satisfies Lemma 1 with $r = k$.*

Note that the condition $T \geq 3$ is necessary to establish Theorem 3, as if there are only two tasks we can construct ground truth parameters such that the induced loss $\mathcal{L}^*$ has infinite solutions. See Appendix F.1 for an example.

**Summary.** These results show that all global minima of the LoRA-ML objective are low-rank adaptable to the downstream task and achieve finite-sample test task prediction error which grows as $\mathcal{O}\left(\frac{kd}{n}\right)$. Crucially, this avoids the factor of $T$ incurred by standard retraining. Further, if $T \geq 3$, minimizing the LoRA-ML objective guarantees recovery of the ground truth parameters.

### 3.2.2 Algorithms for Minimizing (14)

As shown above, minimizing the LoRA-ML objective (14) leads to recovery of the ground truth parameters, with a small rank-$2k$ error term when $T = 2$. We prove that this minimization problem can always be solved by local optimization methods when there are two retraining tasks.

**Theorem 4.** *If $T = 2$ and the ambient dimension $d \geq 2k$, then $\mathcal{L}^*(\hat{A}, \hat{U}) = 0$ if and only if $(\hat{A}, \hat{U})$ is a second order stationary point (SOSP) of $\mathcal{L}^*$.*

When $T = 2$, local optimization algorithms for finding SOSPs, such as perturbed gradient descent and cubic-regularized Newton method, can efficiently minimize the meta-learning objective. Surprisingly, when there are three or more tasks, numerical experiments (see Appendix F.2) show that adversarially picking $U_t^*$ can result in specific instantiations of (14) with spurious local minima. In the next section, we perform extensive numerical experiments for various values of $T$ which show that these spurious minima are almost never found in practice and vanilla gradient descent is sufficient to minimize (14).

## 4 Experiments

We test our framework across three settings. We first consider synthetic data for linear models using LoRA fine-tuning to validate our theory in Section 3. Next, we conduct real-data experiments on vision and language tasks, comparing our general PEFT-ML retraining approach to standard retraining and Reptile [13], a gradient-based meta-learning method. To highlight the flexibility of our framework, we consider two different fine-tuning schemes: LoRA and fine-tuning just the last layer.

**Synthetic Data**. We first test our model on synthetic regression tasks, relaxing the assumptions from our theoretical results. We consider data $y_{t,j} = (A^* + R_t^*)x_{t,j} + \epsilon_{t,j}$ for task $t$ and $j \in [n_t]$, where

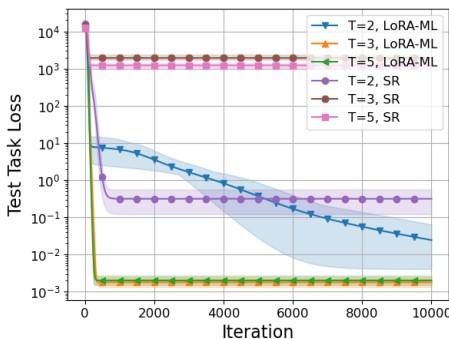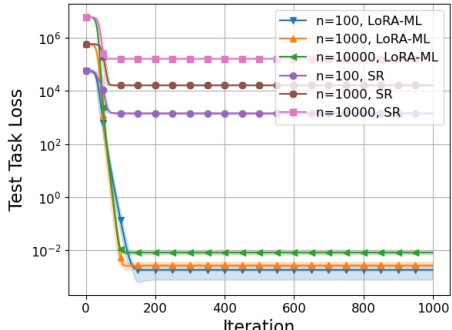

Figure 1: Linear model fine-tuning performance varying number of retraining tasks $T$ (left) and number of fine-tuning samples $n$ (right) for LoRA-ML (ours) and standard retraining (SR).

Table 1: Mean test accuracies and standard errors for PEFT-ML and standard retraining methods using LoRA-based and last-layer-based fine-tuning for adapting to a subset of CIFAR-10 classes.

| LoRA Fine-Tuning | | | | Last-Layer Fine-Tuning | | |
|---|---|---|---|---|---|---|
| **Method** | **Mean** | **Std. Err.** | | **Method** | **Mean** | **Std. Err.** |
| LoRA-ML | **86.09** | 0.35 | | Last-Layer-ML | **83.90** | 0.17 |
| SR+LoRA | 85.29 | 0.17 | | SR+Last-Layer | 81.16 | 0.22 |
| Reptile+LoRA | 81.30 | 0.42 | | Reptile+Last-Layer | 72.55 | 0.39 |

$\boldsymbol{x}_{t,j}$, $\boldsymbol{\epsilon}_{t,j}$, and $\boldsymbol{R}_t^*$ are generated just as in Section 3. $\boldsymbol{A}^*$ is constructed by sampling each entry as an i.i.d. $\mathcal{N}(0,1)$ random variable. We define parameters $N, n$ such that the number of samples per retraining task $n_t = N$ for all $t \leq T$, and the number of test task samples $n_{T+1} = n$. Setting $\mathcal{L}$ to be the mean squared error loss, we run gradient descent on the standard retraining (1) and the LoRA-ML (5) objectives. After recovering $\hat{\boldsymbol{A}}$ during retraining, we apply the low-rank adaptation $\boldsymbol{QV}^\top$ when fine-tuning to the test task. When $T = 2$, we use a rank-$3k$ adaptation during fine-tuning to account for the inexact recovery explained in Theorem 2, and otherwise use a rank-$k$ adaptation.

In each experiment we vary one hyperparameter from a fixed set of values and plot the prediction error between the recovered model and the ground truth model $\frac{1}{n} \sum_j \|\boldsymbol{y}_{T+1,j} - (\hat{\boldsymbol{A}} + \boldsymbol{QV}^\top)\boldsymbol{x}_{T+1,j}\|_2^2$. Results are averaged over 5 trials, with the shaded region showing the full range of values. Figure 1 shows that LoRA-ML retraining significantly outperforms standard retraining (SR) for all data settings. For $T > 2$, we see that applying gradient descent to the LoRA-ML objective is sufficient to achieve global minimization and recover an adaptable solution. Thus, even though there may exist spurious local minimizers, we do not encounter them in practice. Further, the fine-tuning performance after standard retraining worsens with larger $T$. This is supported by our theory in Section 3.1 which shows that as $T$ increases, standard retraining recovers worse solutions that leave a larger intrinsic dimension for the fine-tuning stage. See Appendix D for hyperparameter details and further ablations.

**Vision Experiments**. We use CIFAR-10 [21], and define $T = 4$ binary retraining tasks involving classification between consecutive class labels. Specifically, task 1 classifies between classes 1 and 2, task 2 between classes 3 and 4, etc. The test task is binary classification between classes 9 and 10. We evaluate our PEFT-ML framework using a model based on the MLP-Mixer architecture [35]. We compare two different adaptation methods: (i) LoRA with rank 1 adapters, and (ii) fine-tuning only the last layer. For both PEFT strategies, we evaluate three retraining strategies: (a) PEFT-ML for each PEFT method (LoRA-ML, Last-Layer-ML), (b) standard retraining, and (c) Reptile, a popular gradient-based meta-learning method. We report mean results along with standard errors across 5 trials in Table 1. Across both fine-tuning methods, PEFT-ML retraining consistently yields the best performance. Interestingly, standard retraining outperforms Reptile, as Reptile assumes gradient-based adaptation which is misaligned with the PEFT methods used to adapt at test-time.

**Language Experiments.** We test our PEFT-ML framework using the ConvAI2 dataset with the RoBERTa-base model [22]. ConvAI2 comprises conversations between two personas, where each

Table 2: Mean accuracies $\pm$ standard error for 10 test tasks using different retraining and fine-tuning method combinations on ConvAI2.

| Algorithm | T1 | T2 | T3 | T4 | T5 | T6 | T7 | T8 | T9 | T10 | Avg |
|---|---|---|---|---|---|---|---|---|---|---|---|
| LoRA-ML | $57 \pm 2$ | $\mathbf{41} \pm 4$ | $\mathbf{51} \pm 4$ | $\mathbf{50} \pm 4$ | $\mathbf{51} \pm 4$ | $\mathbf{30} \pm 2$ | $\mathbf{66} \pm 5$ | $\mathbf{47} \pm 4$ | $\mathbf{43} \pm 4$ | $\mathbf{38} \pm 2$ | $\mathbf{47.4} \pm 1.9$ |
| SR+LoRA | $\mathbf{59} \pm 5$ | $31 \pm 4$ | $50 \pm 7$ | $40 \pm 4$ | $24 \pm 5$ | $20 \pm 2$ | $41 \pm 10$ | $36 \pm 2$ | $23 \pm 5$ | $26 \pm 6$ | $35.0 \pm 4.1$ |
| Reptile+LoRA | $45 \pm 7$ | $29 \pm 5$ | $35 \pm 6$ | $36 \pm 2$ | $19 \pm 6$ | $21 \pm 3$ | $28 \pm 10$ | $29 \pm 7$ | $21 \pm 5$ | $21 \pm 7$ | $28.4 \pm 5.0$ |
| Last-Layer-ML | $\mathbf{55} \pm 2$ | $\mathbf{27} \pm 5$ | $\mathbf{51} \pm 5$ | $\mathbf{44} \pm 2$ | $\mathbf{36} \pm 4$ | $\mathbf{25} \pm 4$ | $\mathbf{55} \pm 4$ | $\mathbf{43} \pm 5$ | $\mathbf{33} \pm 4$ | $\mathbf{34} \pm 4$ | $\mathbf{40.2} \pm 1.5$ |
| SR+Last-Layer | $41 \pm 4$ | $9 \pm 5$ | $29 \pm 3$ | $36 \pm 5$ | $28 \pm 8$ | $15 \pm 4$ | $31 \pm 8$ | $24 \pm 5$ | $15 \pm 6$ | $17 \pm 6$ | $24.5 \pm 4.2$ |
| Reptile+Last-Layer | $35 \pm 6$ | $13 \pm 4$ | $18 \pm 3$ | $29 \pm 3$ | $19 \pm 9$ | $18 \pm 4$ | $15 \pm 9$ | $17 \pm 5$ | $15 \pm 6$ | $16 \pm 4$ | $19.4 \pm 4.3$ |

Table 3: Mean accuracies (averaged over all test tasks) $\pm$ standard error for different retraining methods at varying LoRA ranks during fine-tuning.

| Algorithm | Rank 1 | Rank 4 | Rank 8 | Rank 16 |
|---|---|---|---|---|
| LoRA-ML | $\mathbf{44.7} \pm 0.8$ | $\mathbf{48.6} \pm 1.6$ | $\mathbf{47.4} \pm 1.9$ | $\mathbf{48.2} \pm 1.5$ |
| SR+LoRA | $36.3 \pm 4.1$ | $35.5 \pm 4.3$ | $35.0 \pm 4.1$ | $37.1 \pm 4.1$ |
| Reptile+LoRA | $26.6 \pm 5.8$ | $27.7 \pm 5.3$ | $28.4 \pm 5.0$ | $27.8 \pm 5.9$ |

persona is associated with a list of facts that informs their responses. We model learning the dialogue continuations of each persona as a different task, where we aim to select the correct continuation from a set of choices given the conversation history. We retrain with $T = 10$ tasks and then fine-tune the model from the best-performing epoch to each of the 10 test tasks. We run 5 trials and report the mean accuracy and standard error on the held-out data for each test task in Table 2, using the same setup and method naming conventions as in the vision experiments. PEFT-based meta-learning performs best for both adaptation methods.

Table 2 reports results where LoRA-ML retraining and all fine-tuning uses a LoRA rank of 8. To assess robustness to the choice of rank, Table 3 reports performance across different values. LoRA-ML consistently outperforms the baselines, and while choosing a rank larger than 1 clearly improves the performance of LoRA-ML, ranks 4, 8, and 16 yield similar results. We note that LoRA-ML is always retrained with the same rank used for fine-tuning. Additional details for both real-data experiments are provided in Appendix E.

# 5 Conclusion

We presented PEFT-ML for retraining an FM on a collection of tasks to prepare the model for subsequent downstream fine-tuning. We theoretically demonstrated strict performance gaps between standard retraining and the PEFT-ML objective using LoRA (LoRA-ML). Empirically, retraining with PEFT-ML outperformed standard retraining for adapting to unseen downstream tasks using LoRA and last-layer fine-tuning. Future research includes extending our theoretical analysis to more general adapters and more complex model architectures, such as transformers.

# Acknowledgments

This work was supported in part by NSF Grants 2019844, 2107037, and 2112471, ONR Grant N00014-19-1-2566, the Machine Learning Lab (MLL) at UT Austin, the NSF AI Institute for Foundations of Machine Learning (IFML), and Qualcomm through the Wireless Networking and Communications Group (WNCG) Industrial Affiliates Program. We are grateful for computing support on the Vista GPU Cluster through the Center for Generative AI (CGAI) and the Texas Advanced Computing Center (TACC) at the University of Texas at Austin.

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

# A    Related Work on LoRA-Style PEFT

There is a vast amount of work in developing PEFT methods for FMs. The LoRA algorithm [7] has established itself as a popular and successful PEFT strategy and has inspired various extensions such as QLoRA, DoRA, and others [26, 25, 27]. These algorithms are heuristics for mimicking the full finetuning of an FM to a specific downstream task and have proven to be empirically successful in various settings. However, there is a lack of theoretical analysis on the adaptability of PFMs under LoRA-style adaptations, the ability to efficiently optimize LoRA-style objectives, and the kinds of solutions they recover. Some recent works have attempted to analyze different parts of these theoretical questions.

**Convergence of LoRA.** [29] analyzes the optimization landscape for LoRA for the Neural Tangent Kernel regime. The authors show that LoRA finetuning converges in this setting as they prove that the objective function satisfies a strict saddle property, ensuring that there are no spurious local minima. However, this focuses on the actual ability of LoRA to converge to the optimal low-rank adapter given an FM, and does not consider the adaptability of the FM in the first place.

**Expressivity of LoRA.** [30] derives the expressive power of LoRA as a function of model depth. This work shows that under some mild conditions, fully connected and transformer networks when respectively adapted with LoRA can closely approximate arbitrary smaller networks. They quantify the required LoRA rank to achieve this approximation as well as the resulting approximation error.

# B    Proofs

## B.1    Proof of Theorem 1

By Equation (12) we have that $\hat{\boldsymbol{A}}_{\text{SR}} = \boldsymbol{A}^* + \frac{1}{T}\sum_{t=1}^{T}\boldsymbol{U}_t^*\boldsymbol{U}_t^{*\top}$. In the following the expectation is always taken over $\boldsymbol{U}^* = (\boldsymbol{U}_1^*, \ldots, \boldsymbol{U}_T^*)$, where $\boldsymbol{U}_t^* \in \mathbb{R}^{d \times k}$ satisfies $\text{vec}(\boldsymbol{U}_t^*) \sim \mathcal{N}(\boldsymbol{0}, \boldsymbol{I}_{dk})$. Then,

$$
\begin{aligned}
\mathbb{E}_{\boldsymbol{U}^*}\left[\sum_{t=1}^{T}\mathcal{L}_t(\hat{\boldsymbol{A}}_{\text{SR}})\right] &= \sum_{t=1}^{T}\mathbb{E}\left[\left\|\boldsymbol{A}^* + \boldsymbol{U}_t^*\boldsymbol{U}_t^{*\top} - \boldsymbol{A}^* - \frac{1}{T}\sum_{s=1}^{T}\boldsymbol{U}_s^*\boldsymbol{U}_s^{*\top}\right\|_F^2\right] \\
&= \sum_{t=1}^{T}\mathbb{E}\left[\left\|\boldsymbol{U}_t^*\boldsymbol{U}_t^{*\top} - \frac{1}{T}\sum_{s=1}^{T}\boldsymbol{U}_s^*\boldsymbol{U}_s^{*\top}\right\|_F^2\right] \\
&= \sum_{t=1}^{T}\mathbb{E}\left[\left\|\boldsymbol{U}_t^*\boldsymbol{U}_t^{*\top} - k\boldsymbol{I} - \frac{1}{T}\sum_{s=1}^{T}\left(\boldsymbol{U}_s^*\boldsymbol{U}_s^{*\top} - k\boldsymbol{I}\right)\right\|_F^2\right] \\
&= \sum_{t=1}^{T}\mathbb{E}\left[\left\|\boldsymbol{U}_t^*\boldsymbol{U}_t^{*\top} - k\boldsymbol{I}\right\|_F^2 + \left\|\frac{1}{T}\sum_{s=1}^{T}\left(\boldsymbol{U}_s^*\boldsymbol{U}_s^{*\top} - k\boldsymbol{I}\right)\right\|_F^2 - \right. \\
&\qquad\qquad \left. \frac{2}{T}\operatorname{tr}\left\{\left(\boldsymbol{U}_t^*\boldsymbol{U}_t^{*\top} - k\boldsymbol{I}\right)\left(\sum_{s=1}^{T}\boldsymbol{U}_s^*\boldsymbol{U}_s^{*\top} - k\boldsymbol{I}\right)\right\}\right] \\
&= T\mathbb{E}\left[\left\|\boldsymbol{U}_1^*\boldsymbol{U}_1^{*\top} - k\boldsymbol{I}\right\|_F^2\right] + \frac{1}{T}\mathbb{E}\left[\left\|\sum_{s=1}^{T}\left(\boldsymbol{U}_s^*\boldsymbol{U}_s^{*\top} - k\boldsymbol{I}\right)\right\|_F^2\right] \\
&\qquad\qquad - 2\mathbb{E}\left[\operatorname{tr}\left\{\left(\boldsymbol{U}_1^*\boldsymbol{U}_1^{*\top} - k\boldsymbol{I}\right)\left(\sum_{s=1}^{T}\boldsymbol{U}_s^*\boldsymbol{U}_s^{*\top} - k\boldsymbol{I}\right)\right\}\right],
\end{aligned}
$$

where the last equality follows from the fact that each $\boldsymbol{U}_t^*\boldsymbol{U}_t^{*\top}$ are i.i.d.

Note that the second term $\frac{1}{T}\mathbb{E}\left[\left\|\sum_{s=1}^{T}\left(\boldsymbol{U}_s^*\boldsymbol{U}_s^{*\top}-k\boldsymbol{I}\right)\right\|_F^2\right]$ is the total variance of the sum of i.i.d. matrices $\boldsymbol{U}_s^*\boldsymbol{U}_s^{*\top}-k\boldsymbol{I}$, so it is equal to the sum of the individual total variances:

$$\frac{1}{T}\mathbb{E}\left[\left\|\sum_{s=1}^{T}\left(\boldsymbol{U}_s^*\boldsymbol{U}_s^{*\top}-k\boldsymbol{I}\right)\right\|_F^2\right]=\frac{1}{T}\sum_{s=1}^{T}\mathbb{E}\left[\left\|\left(\boldsymbol{U}_s^*\boldsymbol{U}_s^{*\top}-k\boldsymbol{I}\right)\right\|_F^2\right]=\mathbb{E}\left[\left\|\boldsymbol{U}_1^*\boldsymbol{U}_1^{*\top}-k\boldsymbol{I}\right\|_F^2\right].$$

Further, the third term $-2\mathbb{E}\left[\operatorname{tr}\left\{\left(\boldsymbol{U}_1^*\boldsymbol{U}_1^{*\top}-k\boldsymbol{I}\right)\left(\sum_{s=1}^{T}\boldsymbol{U}_s^*\boldsymbol{U}_s^{*\top}-k\boldsymbol{I}\right)\right\}\right]$ is the sum of total covariances between zero mean random matrices $\boldsymbol{U}_1^*\boldsymbol{U}_1^{*\top}-k\boldsymbol{I}$ and each $\boldsymbol{U}_s^*\boldsymbol{U}_s^{*\top}-k\boldsymbol{I}$. Since each $\boldsymbol{U}_s^*$ is drawn independently, we only pick up the first term of $\sum_{s=1}^{T}\boldsymbol{U}_s^*\boldsymbol{U}_s^{*\top}-k\boldsymbol{I}$:

$$\mathbb{E}\left[\operatorname{tr}\left\{\left(\boldsymbol{U}_1^*\boldsymbol{U}_1^{*\top}-k\boldsymbol{I}\right)\left(\sum_{s=1}^{T}\boldsymbol{U}_s^*\boldsymbol{U}_s^{*\top}-k\boldsymbol{I}\right)\right\}\right]=\mathbb{E}\left[\operatorname{tr}\left\{\left(\boldsymbol{U}_1^*\boldsymbol{U}_1^{*\top}-k\boldsymbol{I}\right)\left(\boldsymbol{U}_1^*\boldsymbol{U}_1^{*\top}-k\boldsymbol{I}\right)\right\}\right]$$

$$=\mathbb{E}\left[\left\|\boldsymbol{U}_1^*\boldsymbol{U}_1^{*\top}-k\boldsymbol{I}\right\|_F^2\right].$$

Combining the above simplifications gives that

$$\mathbb{E}_{\boldsymbol{U}^*}\left[\sum_{t=1}^{T}\mathcal{L}_t(\hat{\boldsymbol{A}}_{\mathrm{SR}})\right]=(T-1)\,\mathbb{E}\left[\left\|\boldsymbol{U}_1^*\boldsymbol{U}_1^{*\top}-k\boldsymbol{I}\right\|_F^2\right].$$

Then using the fact that each $\boldsymbol{U}_t^*\boldsymbol{U}_t^*$ is an independent sample of a $d\times d$ Wishart distribution with scale matrix $\boldsymbol{I}$ and $k$ degrees of freedom, we have that

$$\mathbb{E}\left[\left\|\boldsymbol{U}_1^*\boldsymbol{U}_1^{*\top}-k\boldsymbol{I}\right\|_F^2\right]=kd(d+1)$$

## B.2 Proof of Propositions 1, 2

Recall $\hat{\boldsymbol{A}}_{\mathrm{SR}}=\boldsymbol{A}^*+\frac{1}{T}\sum_{t=1}^{T}\boldsymbol{U}_t^*\boldsymbol{U}_t^{*\top}$. Then for any $\boldsymbol{Q},\boldsymbol{V}\in\mathbb{R}^{d\times r}$,

$$\mathcal{L}_{T+1}^*(\boldsymbol{Q},\boldsymbol{V}\,;\hat{\boldsymbol{A}}_{\mathrm{SR}})=\frac{1}{2}\left\|\boldsymbol{A}^*+\boldsymbol{U}_{T+1}^*\boldsymbol{U}_{T+1}^{*\top}-\hat{\boldsymbol{A}}_{\mathrm{SR}}-\boldsymbol{Q}\boldsymbol{V}^\top\right\|_F^2$$

$$=\frac{1}{2}\left\|\boldsymbol{U}_{T+1}^*\boldsymbol{U}_{T+1}^{*\top}-\sum_{t=1}^{T}\boldsymbol{U}_t^*\boldsymbol{U}_t^{*\top}-\boldsymbol{Q}\boldsymbol{V}^\top\right\|_F^2$$

By Remark 1 we have that $\operatorname{rank}\left(\boldsymbol{U}_{T+1}^*\boldsymbol{U}_{T+1}^{*\top}-\sum_{t=1}^{T}\boldsymbol{U}_t^*\boldsymbol{U}_t^{*\top}\right)=\min\{d,k(T+1)\}$. Then Proposition 1 follows from that fact that $\operatorname{rank}(\boldsymbol{Q}\boldsymbol{V}^\top)\le r$.

Further, as $T\to\infty$, the strong law of large numbers implies that $\frac{1}{T}\sum_{t=1}^{T}\boldsymbol{U}_t^*\boldsymbol{U}_t^{*\top}\to\mathbb{E}\left[\boldsymbol{U}_t^*\boldsymbol{U}_t^{*\top}\right]=k\boldsymbol{I}$. Thus for large $T$,

$$\left\|\boldsymbol{U}_{T+1}^*\boldsymbol{U}_{T+1}^{*\top}-\frac{1}{T}\sum_{t=1}^{T}\boldsymbol{U}_t^*\boldsymbol{U}_t^{*\top}-\boldsymbol{Q}\boldsymbol{V}^\top\right\|_F^2\to\left\|\boldsymbol{U}_{T+1}^*\boldsymbol{U}_{T+1}^{*\top}-k\boldsymbol{I}-\boldsymbol{Q}\boldsymbol{V}^\top\right\|_F^2 \qquad (15)$$

Using classic low-rank matrix factorization results, the $\boldsymbol{Q}^*\boldsymbol{V}^{*\top}$ that minimizes $\left\|\boldsymbol{U}_{T+1}^*\boldsymbol{U}_{T+1}^{*\top}-k\boldsymbol{I}-\boldsymbol{Q}\boldsymbol{V}^\top\right\|_F^2$ will exactly capture the $r$ eigenvectors of $\boldsymbol{U}_{T+1}^*\boldsymbol{U}_{T+1}^{*\top}-k\boldsymbol{I}$ with largest magnitude eigenvalue. But, $\boldsymbol{U}_{T+1}^*\boldsymbol{U}_{T+1}^{*\top}-k\boldsymbol{I}$ has $d-k$ eigenvalues of magnitude $k$, so $\boldsymbol{Q}^*\boldsymbol{V}^{*\top}$ can only capture $r$ of them. Thus, $\left\|\boldsymbol{U}_{T+1}^*\boldsymbol{U}_{T+1}^{*\top}-k\boldsymbol{I}-\boldsymbol{Q}^*\boldsymbol{V}^{*\top}\right\|_F^2\ge(d-k-r)k^2$. Since $\boldsymbol{Q}^*\boldsymbol{V}^{*\top}$ minimized this quantity, we have that

$$\mathcal{L}_{T+1}^*(\boldsymbol{Q},\boldsymbol{V}\,;\hat{\boldsymbol{A}}_{\mathrm{SR}})\ge(d-k-r)k^2 \quad \forall\boldsymbol{Q},\boldsymbol{V}\in\mathbb{R}^{d\times r}$$

Thus, $\mathcal{L}_{T+1}(\boldsymbol{Q},\boldsymbol{V}\,;\hat{\boldsymbol{A}}_{\mathrm{SR}})$ scales as $(d-k-r)k^2\approx(d-r)k^2$ since $k\ll d$.

## B.3 Proof of Theorem 2

Since $\mathcal{L}^*(\hat{A}, \hat{U}) = 0$ and $\mathcal{L}^* \geq 0$ we must have that $\nabla_A \mathcal{L}^* = 0$.

Thus, $\hat{A} = A^* - \frac{1}{T} \sum_{j=1}^{T} \left( \hat{U}_j \hat{U}_j^\top - U_j^* U_j^{*\top} \right)$. Plugging this into $\mathcal{L}^*$ gives

$$
\begin{aligned}
0 = \mathcal{L}^*(\hat{A}, \hat{U}) &= \frac{1}{2} \sum_{t=1}^{T} \left\| A^* + U_t^* U_t^{*\top} - \left( A^* - \frac{1}{T} \sum_{s=1}^{T} \left( \hat{U}_s \hat{U}_s^\top - U_s^* U_s^{*\top} \right) \right) - U_t U_t^\top \right\|_F^2 \\
&= \frac{1}{2} \sum_{t=1}^{T} \left\| U_t^* U_t^{*\top} - U_t U_t^\top - \frac{1}{T} \sum_{s=1}^{T} \left( \hat{U}_s \hat{U}_s^\top - U_s^* U_s^{*\top} \right) \right\|_F^2 .
\end{aligned}
$$

Thus each term of the summation is zero, so for all $t, s \in [T]$,

$$
\hat{U}_t \hat{U}_t^T - U_t^* U_t^{*\top} = \hat{U}_s \hat{U}_s^T - U_s^* U_s^{*\top} .
$$

Combining these results gives that

$$
\begin{aligned}
\hat{A} &= A^* - \frac{1}{T} \sum_{s=1}^{T} \left( \hat{U}_s \hat{U}_s^\top - U_s^* U_s^{*\top} \right) \\
&= A^* - \left( \hat{U}_1 \hat{U}_1^\top - U_1^* U_1^{*\top} \right)
\end{aligned}
$$

Let $C = -\hat{U}_1 \hat{U}_1^\top + U_1^* U_1^{*\top}$.

Then $\hat{A} = A^* + C$ and $rank(C) \leq rank(\hat{U}_1 \hat{U}_1^\top) + rank(U_1^* U_1^{*\top}) \leq 2k$.

Note the the effective remaining test-task dimension is

$$
\begin{aligned}
\text{rank} \left( A^* + U_{T+1}^* U_{T+1}^{*\top} - A^* - C \right) &= \text{rank} \left( U_{T+1}^* U_{T+1}^{*\top} - C \right) \\
&\leq \text{rank} \left( U_{T+1}^* U_{T+1}^{*\top} \right) + \text{rank} \left( C \right) \\
&\leq 3k
\end{aligned}
$$

## B.4 Proof of Theorem 3

*Proof.* Since $\mathcal{L}^*(\hat{A}, \hat{U}) = 0$, we have that for all $t, s \in [T]$,

$$
\hat{U}_t \hat{U}_t^\top - U_t^* U_t^{*\top} = \hat{U}_s \hat{U}_s^\top - U_s^* U_s^{*\top} \tag{16}
$$

Applying this to the first three tasks and rearranging gives that

$$
\begin{aligned}
U_1^* U_1^{*\top} &= \hat{U}_1 \hat{U}_1^\top + U_2^* U_2^{*\top} - \hat{U}_2 \hat{U}_2^\top \tag{17} \\
&= \hat{U}_1 \hat{U}_1^\top + U_3^* U_3^{*\top} - \hat{U}_3 \hat{U}_3^\top . \tag{18}
\end{aligned}
$$

We first show that $\text{im}(\hat{U}_1) = \text{im}(U_1^*)$.

Since $U_1^* U_1^{*\top} \succcurlyeq 0$, we must have that $\text{im}(\hat{U}_2) \subseteq \text{im}(\hat{U}_1) + \text{im}(U_2^*)$ and $\text{im}(\hat{U}_3) \subseteq \text{im}(\hat{U}_1) + \text{im}(U_3^*)$, as otherwise there would exist a vector on $\ker \left( \hat{U}_1 \hat{U}_1^\top + U_2^* U_2^{*\top} \right) \cap \ker(\hat{U}_2 \hat{U}_2^\top)^\perp$ whose existence contradicts the positive semi-definiteness of $U_1^* U_1^{*\top}$.

Thus,

$$
\begin{aligned}
\text{im}(U_1^*) &\subseteq \text{im}(\hat{U}_1) + \text{im}(U_2^*) \tag{19} \\
\text{im}(U_1^*) &\subseteq \text{im}(\hat{U}_1) + \text{im}(U_3^*) \tag{20}
\end{aligned}
$$

Using that fact that for subspaces $X, Y, Z$, $X \subseteq Y \implies X + Z \subseteq Y + Z$, we can add $\mathrm{im}(U_2^*)$ and $\mathrm{im}(U_3^*)$ to both sides of 19 and 20 respectively. This gives:

$$\mathrm{im}(U_1^*) \oplus \mathrm{im}(U_2^*) \subseteq \mathrm{im}(\hat{U}_1) + \mathrm{im}(U_2^*) \tag{21}$$

$$\mathrm{im}(U_1^*) \oplus \mathrm{im}(U_3^*) \subseteq \mathrm{im}(\hat{U}_1) + \mathrm{im}(U_3^*). \tag{22}$$

For $t \in \{2, 3\}$, we clearly have that $\dim\left(\mathrm{im}(\hat{U}_1) + \mathrm{im}(U_t^*)\right) \leq \dim \mathrm{im}(\hat{U}_1) + \dim \mathrm{im}(U_t^*) \leq 2k$, and $\dim\left(\mathrm{im}(U_1^*) + \mathrm{im}(U_t^*)\right) = 2k$. Thus,

$$(\mathrm{im}(U_1^*) \oplus \mathrm{im}(U_2^*)) = \left(\mathrm{im}(\hat{U}_1) \oplus \mathrm{im}(U_2^*)\right) \tag{23}$$

$$(\mathrm{im}(U_1^*) \oplus \mathrm{im}(U_3^*)) = \left(\mathrm{im}(\hat{U}_1) \oplus \mathrm{im}(U_3^*)\right) \tag{24}$$

**Lemma 2.** $\left([\mathrm{im}(\hat{U}_1) \oplus \mathrm{im}(U_2^*)] \cap [\mathrm{im}(\hat{U}_1) \oplus \mathrm{im}(U_3^*)]\right) = \mathrm{im}(\hat{U}_1)$

*Proof.* Clearly, $\mathrm{im}(\hat{U}_1) \subseteq \left([\mathrm{im}(\hat{U}_1) \oplus \mathrm{im}(U_2^*)] \cap [\mathrm{im}(\hat{U}_1) \oplus \mathrm{im}(U_3^*)]\right)$. To show the converse, consider $x \in \left([\mathrm{im}(\hat{U}_1) \oplus \mathrm{im}(U_2^*)] \cap [\mathrm{im}(\hat{U}_1) \oplus \mathrm{im}(U_3^*)]\right)$.

By assumption there exists some $a, b, c, d \in \mathbb{R}^k$ such that

$$x = \hat{U}_1 a + U_2^* b \tag{25}$$

$$= \hat{U}_1 c + U_3^* d \tag{26}$$

Thus,

$$\hat{U}_1(a - c) + U_2^* b - U_3^* d = 0. \tag{27}$$

By Equation 23, we can write

$$\mathrm{im}(U_2^*) = ([\mathrm{im}(U_1^*) \oplus \mathrm{im}(U_2^*)] \cap [\mathrm{im}(U_2^*) \oplus \mathrm{im}(U_3^*)])$$

$$= \left([\mathrm{im}(\hat{U}_1) \oplus \mathrm{im}(U_2^*)] \cap [\mathrm{im}(U_2^*) \oplus \mathrm{im}(U_3^*)]\right)$$

Thus, $\mathrm{im}(\hat{U}_1) \cap [\mathrm{im}(U_2^*) \oplus \mathrm{im}(U_3^*)] \subseteq \mathrm{im}(\hat{U}_1) \cap \mathrm{im}(U_2^*) = \{0\}$, so

$$\mathrm{im}(\hat{U}_1) \cap [\mathrm{im}(U_2^*) \oplus \mathrm{im}(U_3^*)] = \{0\} \tag{28}$$

Applying Equation (28) to Equation (27) implies that $a = c$ and $b = d = 0$. Thus $x = \hat{U}_1 a \in \mathrm{im}(\hat{U}_1)$, so $\left([\mathrm{im}(\hat{U}_1) \oplus \mathrm{im}(U_2^*)] \cap [\mathrm{im}(\hat{U}_1) \oplus \mathrm{im}(U_3^*)]\right) \subseteq \mathrm{im}(\hat{U}_1)$. $\square$

Then Equations (19) and (20) combined with Lemma (2) implies that $\mathrm{im}(U_1^*) \subseteq \mathrm{im}(\hat{U}_1)$ but $\dim(\mathrm{im}(U_1^*)) = \dim(\mathrm{im}(\hat{U}_1)) = k$, so $\mathrm{im}(U_1^*) = \mathrm{im}(\hat{U}_1)$.

Since the initial assumptions about $\hat{U}_1$ and $U_1^*$ analogously hold for the corresponding matrices for tasks 2 and 3, by the exact same argument we can show that

$$\mathrm{im}(U_t^*) = \mathrm{im}(\hat{U}_t) \quad \forall t \in [T]. \tag{29}$$

Then by equation (16), $\mathrm{im}(U_1^*) \supseteq \mathrm{im}\left(\hat{U}_1 \hat{U}_1^\top - U_1^* U_1^{*\top}\right) = \mathrm{im}\left(\hat{U}_2 \hat{U}_2^\top - U_2^* U_2^{*\top}\right) \subseteq \mathrm{im}(U_2^*)$. Thus,

$$\mathrm{im}\left(\hat{U}_1 \hat{U}_1^\top - U_1^* U_1^{*\top}\right) \subseteq \mathrm{im}(U_1^*) \cap \mathrm{im}(U_2^*)$$

$$= \{0\}.$$

Thus $\hat{U}_1 \hat{U}_1^\top = U_1^* U_1^{*\top}$. Then by Equation (16), $\hat{U}_t \hat{U}_t^\top = U_t^* U_t^{*\top}$ for all $t \in [T]$. Lastly, since $\mathcal{L}^*(\hat{A}, \hat{U}) = 0$, we have that $\nabla_A \mathcal{L}^*(\hat{A}, \hat{U}) = 0$, so

$$\hat{A} = A^* + \frac{1}{T} \sum_{t=1}^{T} U_t^* U_t^{*\top} - U_t U_t^\top = A^*$$

$\square$

## B.5  Proof of Theorem 4

*Proof.* Clearly if $\mathcal{L}^*(\hat{A}, \hat{U}) = 0$, then $(\hat{A}, \hat{U})$ is an SOSP. The reverse direction is the challenging part of the proof. We equivalently prove that if $(\hat{A}, \hat{U})$ is a critical point and $\mathcal{L}^*(\hat{A}, \hat{U}) \neq 0$, then $\nabla^2 \mathcal{L}^*(\hat{A}, \hat{U})$ has a negative eigenvalue.

Assume for the sake of contradiction that $(\hat{A}, \hat{U})$ is a critical point and $\mathcal{L}^*(\hat{A}, \hat{U}) \neq 0$. Then,

$$\nabla_A \mathcal{L}^*(\hat{A}, \hat{U}) = T(\hat{A} - A^*) + \sum_{t=1}^{T} \left( \hat{U}_t \hat{U}_t^\top - U_t^* U_t^{*\top} \right) = 0 \tag{30}$$

$$\nabla_{U_t} \mathcal{L}^*(\hat{A}, \hat{U}) = 2 \left( \hat{A} - A^* + \hat{U}_t \hat{U}_t^\top - U_t^* U_t^{*\top} \right) \hat{U}_t = 0 \tag{31}$$

Thus,

$$\hat{A} = A^* - \frac{1}{T} \sum_{t=1}^{T} \left( \hat{U}_t \hat{U}_t^\top - U_t^* U_t^{*\top} \right). \tag{32}$$

Define $B_t(\hat{U}) = \hat{U}_t \hat{U}_t^\top - U_t^* U_t^{*\top} - \frac{1}{T} \sum_{s=1}^{T} \left( \hat{U}_s \hat{U}_s^\top - U_s^* U_s^{*\top} \right)$. Despite being a slight abuse of notation, we refer to $B_t(\hat{U})$ as just $B_t$ for the remainder of the proof.

Then (31) equivalently states:

$$B_t \hat{U}_t = 0. \tag{33}$$

Note that by construction, $\sum_{t=1}^{T} B_t = 0$.

Considering $\mathcal{L}$ as a function of the flattened vector $[\text{vec}(A); \text{vec}(U_1); \text{vec}(U_2)]$, and let $U_1 = [x_1 \ \dots \ x_k]$, $U_2 = [y_1 \ \dots \ y_k]$, we compute the Hessian

$$\nabla^2 \mathcal{L} = \begin{bmatrix} \nabla_A^2 \mathcal{L} & \nabla_{U_1} \nabla_A \mathcal{L} & \nabla_{U_2} \nabla_A \mathcal{L} \\ (\nabla_{U_1} \nabla_A \mathcal{L})^\top & \nabla_{U_1}^2 \mathcal{L} & 0 \\ (\nabla_{U_2} \nabla_A \mathcal{L})^\top & 0 & \nabla_{U_2}^2 \mathcal{L} \end{bmatrix} \tag{34}$$

where

$$\nabla_{\boldsymbol{A}}^2 \mathcal{L}^* = 2\boldsymbol{I}_{d^2}$$

$$\nabla_{\boldsymbol{U}_1}\nabla_{\boldsymbol{A}}\mathcal{L}^* = [(\boldsymbol{x}_1 \oplus \boldsymbol{x}_1) \ \ldots \ (\boldsymbol{x}_k \oplus \boldsymbol{x}_k)] \in \mathbb{R}^{d^2 \times dk}$$

$$\nabla_{\boldsymbol{U}_2}\nabla_{\boldsymbol{A}}\mathcal{L}^* = [(\boldsymbol{y}_1 \oplus \boldsymbol{y}_1) \ \ldots \ (\boldsymbol{y}_k \oplus \boldsymbol{y}_k)] \in \mathbb{R}^{d^2 \times dk}$$

$$\nabla_{\boldsymbol{U}_1}^2 \mathcal{L}^* = 2(\boldsymbol{A} + \boldsymbol{U}_1\boldsymbol{U}_1^\top - \boldsymbol{A}^* - \boldsymbol{U}_1^*\boldsymbol{U}_1^{*\top}) \otimes \boldsymbol{I}_k$$

$$+ 2\begin{bmatrix} \boldsymbol{x}_1\boldsymbol{x}_1^\top + \|\boldsymbol{x}_1\|_2^2\boldsymbol{I} & \boldsymbol{x}_1^\top\boldsymbol{x}_2\boldsymbol{I} + \boldsymbol{x}_2\boldsymbol{x}_1^\top & \ldots & \boldsymbol{x}_1^\top\boldsymbol{x}_k\boldsymbol{I} + \boldsymbol{x}_k\boldsymbol{x}_1^\top \\ \boldsymbol{x}_2^\top\boldsymbol{x}_1\boldsymbol{I} + \boldsymbol{x}_1\boldsymbol{x}_2^\top & \boldsymbol{x}_2\boldsymbol{x}_2^\top + \|\boldsymbol{x}_2\|_2^2\boldsymbol{I} & \ldots & \boldsymbol{x}_2^\top\boldsymbol{x}_k\boldsymbol{I} + \boldsymbol{x}_k\boldsymbol{x}_2^\top \\ \vdots & \vdots & \ddots & \vdots \\ \boldsymbol{x}_k^\top\boldsymbol{x}_1\boldsymbol{I} + \boldsymbol{x}_1\boldsymbol{x}_k^\top & \ldots & \ldots & \boldsymbol{x}_k\boldsymbol{x}_k^\top + \|\boldsymbol{x}_k\|_2^2\boldsymbol{I} \end{bmatrix}$$

$$\nabla_{\boldsymbol{U}_2}^2 \mathcal{L}^* = 2(\boldsymbol{A} + \boldsymbol{U}_2\boldsymbol{U}_2^\top - \boldsymbol{A}^* - \boldsymbol{U}_2^*\boldsymbol{U}_2^{*\top}) \otimes \boldsymbol{I}_k$$

$$+ 2\begin{bmatrix} \boldsymbol{y}_1\boldsymbol{y}_1^\top + \|\boldsymbol{y}_1\|_2^2\boldsymbol{I} & \boldsymbol{y}_1^\top\boldsymbol{y}_2\boldsymbol{I} + \boldsymbol{y}_2\boldsymbol{y}_1^\top & \ldots & \boldsymbol{y}_1^\top\boldsymbol{y}_k\boldsymbol{I} + \boldsymbol{y}_k\boldsymbol{y}_1^\top \\ \boldsymbol{y}_2^\top\boldsymbol{y}_1\boldsymbol{I} + \boldsymbol{y}_1\boldsymbol{y}_2^\top & \boldsymbol{y}_2\boldsymbol{y}_2^\top + \|\boldsymbol{y}_2\|_2^2\boldsymbol{I} & \ldots & \boldsymbol{y}_2^\top\boldsymbol{y}_k\boldsymbol{I} + \boldsymbol{y}_k\boldsymbol{y}_2^\top \\ \vdots & \vdots & \ddots & \vdots \\ \boldsymbol{y}_k^\top\boldsymbol{y}_1\boldsymbol{I} + \boldsymbol{y}_1\boldsymbol{y}_k^\top & \ldots & \ldots & \boldsymbol{y}_k\boldsymbol{y}_k^\top + \|\boldsymbol{y}_k\|_2^2\boldsymbol{I} \end{bmatrix}$$

Note that $\oplus$ denotes the Kronecker sum defined as $\boldsymbol{X} \oplus \boldsymbol{Y} = \boldsymbol{I} \otimes \boldsymbol{X} + \boldsymbol{Y} \otimes \boldsymbol{I}$ where $\otimes$ is the Kronecker product.

**Lemma 3.** $\mathcal{L}^*(\hat{\boldsymbol{A}}, \hat{\boldsymbol{U}}) = 0$ *if and only if* $\boldsymbol{B}_t = \boldsymbol{0}$ *for each* $t \in [T]$.

*Proof.* Since $(\hat{\boldsymbol{A}}, \hat{\boldsymbol{U}})$ is a critical point, then plugging Equation (32) into the definition of $\mathcal{L}$ gives that

$$\mathcal{L}^*(\hat{\boldsymbol{A}}, \hat{\boldsymbol{U}}) = \frac{1}{2}\sum_{t=1}^{T} \|\boldsymbol{B}_t\|_F^2 \,.$$

Thus $\mathcal{L}^*(\hat{\boldsymbol{A}}, \hat{\boldsymbol{U}}) = 0$ if and only if $\boldsymbol{B}_t = \boldsymbol{0} \quad \forall t$. $\qquad\square$

**Lemma 4.** *If* $\nabla_{\boldsymbol{U}}^2 \mathcal{L}^*(\hat{\boldsymbol{A}}, \hat{\boldsymbol{U}}) \succeq \boldsymbol{0}$*, then the eigenvectors corresponding to the non-zero eigenvalues of* $\hat{\boldsymbol{U}}_t\hat{\boldsymbol{U}}_t^\top$ *are the leading non-negative eigenvectors of* $\boldsymbol{A}^* + \boldsymbol{U}_t^*\boldsymbol{U}_t^{*\top} - \hat{\boldsymbol{A}}$ *for all* $t \in [T]$.

*Proof.* Consider the function $\bar{f}_t(\boldsymbol{U}_t; \hat{\boldsymbol{A}}) = \frac{1}{2}\left\|\boldsymbol{A}^* + \boldsymbol{U}_t^*\boldsymbol{U}_t^{*\top} - \hat{\boldsymbol{A}} - \boldsymbol{U}_t\boldsymbol{U}_t^\top\right\|_F^2$. $\bar{f}_t$ is simply the $t$th summand in $\mathcal{L}^*$ where $\boldsymbol{A} = \hat{\boldsymbol{A}}$ is fixed and we only consider the variable $\boldsymbol{U}_t$. Minimizing $\bar{f}_t$ is identical to the problem of symmetric matrix factorization.

Using well-known properties of symmetric matrix factorization, since $\nabla\bar{f}_t(\hat{\boldsymbol{U}}_t) = \boldsymbol{0}$, we must have that $\hat{\boldsymbol{U}}_t = \boldsymbol{V}_t\boldsymbol{\Gamma}$ where the columns of $\boldsymbol{V}_t$ are the properly scaled eigenvectors of $\boldsymbol{A}^* + \boldsymbol{U}_t^*\boldsymbol{U}_t^{*\top} - \hat{\boldsymbol{A}}$ with non-negative eigenvalues where each column has norm equal to the square root of its corresponding eigenvalue, and $\boldsymbol{\Gamma} \in O_k$ is some orthogonal matrix. Further, if the eigenvectors corresponding to the non-zero eigenvalues of $\hat{\boldsymbol{U}}_t\hat{\boldsymbol{U}}_t^\top$ are not the leading non-negative eigenvectors, then $\nabla^2\bar{f}_t(\hat{\boldsymbol{U}}) \nsucceq \boldsymbol{0}$ by [36]. Since $\nabla^2\bar{f}_t(\hat{\boldsymbol{U}}_t)$ is a diagonal block of $\nabla^2\mathcal{L}^*(\hat{\boldsymbol{A}}, \hat{\boldsymbol{U}})$, $\nabla^2\bar{f}_i(\hat{\boldsymbol{U}}_t) \nsucceq \boldsymbol{0}$ would imply $\nabla^2\mathcal{L}^*(\hat{\boldsymbol{A}}, \hat{\boldsymbol{U}}) \nsucceq \boldsymbol{0}$. $\qquad\square$

**Remark 3.** *Without loss of generality, we can assume that the eigenvectors corresponding to the non-zero eigenvalues of* $\hat{\boldsymbol{U}}_t\hat{\boldsymbol{U}}_t^\top$ *are the leading non-negative eigenvectors of* $\boldsymbol{A}^* + \boldsymbol{U}_t^*\boldsymbol{U}_t^{*\top} - \hat{\boldsymbol{A}}$ *for all* $i$.

**Lemma 5.** $\left(\hat{\boldsymbol{U}}_2\hat{\boldsymbol{U}}_2^\top - \hat{\boldsymbol{U}}_1\hat{\boldsymbol{U}}_1^\top\right)\boldsymbol{x} = \left(\boldsymbol{U}_2^*\boldsymbol{U}_2^{*\top} - \boldsymbol{U}_1^*\boldsymbol{U}_1^{*\top}\right)\boldsymbol{x}$ *for all* $\boldsymbol{x} \in \mathrm{im}(\hat{\boldsymbol{U}}_1) + \mathrm{im}(\hat{\boldsymbol{U}}_2)$.

*Proof.* Recall $\boldsymbol{B}_1 = \frac{1}{2}\left(\hat{\boldsymbol{U}}_1\hat{\boldsymbol{U}}_1^\top - \boldsymbol{U}_1^*\boldsymbol{U}_1^{*\top} - \hat{\boldsymbol{U}}_2\hat{\boldsymbol{U}}_2^\top + \boldsymbol{U}_2^*\boldsymbol{U}_2^{*\top}\right)$. Then applying first-order stationarity and the fact that $\boldsymbol{B}_2 = -\boldsymbol{B}_1$, we have

$$\left(\hat{\boldsymbol{U}}_2\hat{\boldsymbol{U}}_2^\top - \hat{\boldsymbol{U}}_1\hat{\boldsymbol{U}}_1^\top\right)\hat{\boldsymbol{U}}_1 = \left(\boldsymbol{U}_2^*\boldsymbol{U}_2^{*\top} - \boldsymbol{U}_1^*\boldsymbol{U}_1^{*\top}\right)\hat{\boldsymbol{U}}_1$$

$$\left(\hat{\boldsymbol{U}}_2\hat{\boldsymbol{U}}_2^\top - \hat{\boldsymbol{U}}_1\hat{\boldsymbol{U}}_1^\top\right)\hat{\boldsymbol{U}}_2 = \left(\boldsymbol{U}_2^*\boldsymbol{U}_2^{*\top} - \boldsymbol{U}_1^*\boldsymbol{U}_1^{*\top}\right)\hat{\boldsymbol{U}}_2.$$

$\square$

**Corollary 5.** $\hat{\boldsymbol{U}}_2\hat{\boldsymbol{U}}_2^\top - \hat{\boldsymbol{U}}_1\hat{\boldsymbol{U}}_1^\top$ and $\boldsymbol{U}_2^*\boldsymbol{U}_2^{*\top} - \boldsymbol{U}_1^*\boldsymbol{U}_1^{*\top}$ *share an eigenbasis.*

*Proof.* Using the lemma, any non-zero eigenvector-eigenvalue pair of $\hat{\boldsymbol{U}}_2\hat{\boldsymbol{U}}_2^\top - \hat{\boldsymbol{U}}_1\hat{\boldsymbol{U}}_1^\top$ is also an eigenvector-eigenvalue pair of $\boldsymbol{U}_2^*\boldsymbol{U}_2^{*\top} - \boldsymbol{U}_1^*\boldsymbol{U}_1^{*\top}$. Denote the space defined by the span of these eigenvectors as $\boldsymbol{S}$. Then all other eigenvectors of $\boldsymbol{U}_2^*\boldsymbol{U}_2^{*\top} - \boldsymbol{U}_1^*\boldsymbol{U}_1^{*\top}$ are orthogonal to $\boldsymbol{S}$, so they are also 0-eigenvectors of $\hat{\boldsymbol{U}}_2\hat{\boldsymbol{U}}_2^\top - \hat{\boldsymbol{U}}_1\hat{\boldsymbol{U}}_1^\top$. Thus the two matrices share an eigenbasis. $\square$

**Corollary 6.** $\dim\left(\operatorname{im}\hat{\boldsymbol{U}}_1 + \operatorname{im}\hat{\boldsymbol{U}}_2\right) \leq 2k-1$, *i.e., the set of columns of $\hat{\boldsymbol{U}}_1$ and $\hat{\boldsymbol{U}}_2$ are not linearly independent.*

*Proof.* Assume for contradiction that the vectors in the set $\boldsymbol{S} = \{\hat{\boldsymbol{U}}_1\boldsymbol{e}_i \mid i = 1,\ldots,k\} \cup \{\hat{\boldsymbol{U}}_2\boldsymbol{e}_i \mid i = 1,\ldots,k\}$ are linearly independent, where $\boldsymbol{e}_i$ is the $i$th standard basis vector in $\mathbb{R}^k$.

Then note that $\left(\hat{\boldsymbol{U}}_1\hat{\boldsymbol{U}}_1^\top - \hat{\boldsymbol{U}}_2\hat{\boldsymbol{U}}_2^\top\right)\boldsymbol{x} \neq \boldsymbol{0}$ and $\left(\boldsymbol{U}_1^*\boldsymbol{U}_1^{*\top} - \boldsymbol{U}_2^*\boldsymbol{U}_2^{*\top}\right)\boldsymbol{x} \neq \boldsymbol{0}$ for all $\boldsymbol{x} \in \boldsymbol{S}$. By Lemma (5), $\hat{\boldsymbol{U}}_1\hat{\boldsymbol{U}}_1^\top - \hat{\boldsymbol{U}}_2\hat{\boldsymbol{U}}_2^\top$ and $\boldsymbol{U}_1^*\boldsymbol{U}_1^{*\top} - \boldsymbol{U}_2^*\boldsymbol{U}_2^{*\top}$ agree for each vector on the $2k$-dimensional space $\operatorname{span}(\boldsymbol{S})$. But, both $\operatorname{rank}(\hat{\boldsymbol{U}}_1\hat{\boldsymbol{U}}_1^\top - \hat{\boldsymbol{U}}_2\hat{\boldsymbol{U}}_2^\top), \operatorname{rank}(\boldsymbol{U}_1^*\boldsymbol{U}_1^{*\top} - \boldsymbol{U}_2^*\boldsymbol{U}_2^{*\top}) \leq 2k$ by construction. Then by dimension counting, $\hat{\boldsymbol{U}}_1\hat{\boldsymbol{U}}_1^\top - \hat{\boldsymbol{U}}_2\hat{\boldsymbol{U}}_2^\top$ and $\boldsymbol{U}_1^*\boldsymbol{U}_1^{*\top} - \boldsymbol{U}_2^*\boldsymbol{U}_2^{*\top}$ must send $\operatorname{span}\{\boldsymbol{S}\}^\perp$ to $\boldsymbol{0}$. Thus, $\hat{\boldsymbol{U}}_1\hat{\boldsymbol{U}}_1^\top - \hat{\boldsymbol{U}}_2\hat{\boldsymbol{U}}_2^\top$ and $\boldsymbol{U}_1^*\boldsymbol{U}_1^{*\top} - \boldsymbol{U}_2^*\boldsymbol{U}_2^{*\top}$ agree on the entire basis formed by concatenating basis vectors of $\operatorname{span}\{\boldsymbol{S}\}^\perp$ with those of $\operatorname{span}\{\boldsymbol{S}\}$. This implies that $\hat{\boldsymbol{U}}_1\hat{\boldsymbol{U}}_1^\top - \hat{\boldsymbol{U}}_2\hat{\boldsymbol{U}}_2^\top = \boldsymbol{U}_1^*\boldsymbol{U}_1^{*\top} - \boldsymbol{U}_2^*\boldsymbol{U}_2^{*\top}$ and thus $\boldsymbol{B}_1 = \hat{\boldsymbol{U}}_1\hat{\boldsymbol{U}}_1^\top - \hat{\boldsymbol{U}}_2\hat{\boldsymbol{U}}_2^\top - \boldsymbol{U}_1^*\boldsymbol{U}_1^{*\top} + \boldsymbol{U}_2^*\boldsymbol{U}_2^{*\top} = \boldsymbol{0}$. Then $\boldsymbol{B}_2 = -\boldsymbol{B}_1 = \boldsymbol{0}$ so by Lemma 3, $\mathcal{L}^*(\hat{\boldsymbol{A}},\hat{\boldsymbol{U}}) = 0$ which is a contradiction. $\square$

**Lemma 6.** $\boldsymbol{U}_2^*\boldsymbol{U}_2^{*\top} - \boldsymbol{U}_1^*\boldsymbol{U}_1^{*\top}$ *has exactly $k$ positive and $k$ negative eigenvalues.*

*Proof.* First, note that $\boldsymbol{U}_2^*\boldsymbol{U}_2^{*\top}$ has exactly $k$ positive eigenvalues and $k - d$ eigenvalues of $\boldsymbol{0}$. Then $\boldsymbol{U}_2^*\boldsymbol{U}_2^{*\top} - (\boldsymbol{U}_1^*\boldsymbol{e}_1)(\boldsymbol{U}_1^*\boldsymbol{e}_1)^\top$ has rank $k + 1$ because of the linear independence of the columns of the combined set of columns $\boldsymbol{U}_1^*$ and $\boldsymbol{U}_2^*$. Further, since we subtract $(\boldsymbol{U}_1^*\boldsymbol{e}_1)(\boldsymbol{U}_1^*\boldsymbol{e}_1)^\top$, we must be accumulating an additional negative eigenvalue relative to $\boldsymbol{U}_2^*\boldsymbol{U}_2^{*\top}$. Continuing this process shows that subtracting $(\boldsymbol{U}_1^*\boldsymbol{e}_{j+1})(\boldsymbol{U}_1^*\boldsymbol{e}_{j+1})^\top$ from $\boldsymbol{U}_2^*\boldsymbol{U}_2^{*\top} - \sum_{t=1}^{j}(\boldsymbol{U}_1^*\boldsymbol{e}_i)(\boldsymbol{U}_1^*\boldsymbol{e}_i)^\top$ contributes exactly one more negative eigenvalue, since $\boldsymbol{U}_1^*\boldsymbol{e}_{j+1}$ can never be written as a linear combination of $\{\boldsymbol{U}_1^*\boldsymbol{e}_1,\ldots\boldsymbol{U}_1^*\boldsymbol{e}_k,\boldsymbol{U}_2^*\boldsymbol{e}_1,\ldots\boldsymbol{U}_2^*\boldsymbol{e}_j\}$ for $0 < j < k$. The result then follows from induction. $\square$

**Lemma 7.** $\operatorname{rank}(\hat{\boldsymbol{U}}_1) = \operatorname{rank}(\hat{\boldsymbol{U}}_2) = k$.

*Proof.* Assume for contradiction that $\operatorname{rank}(\hat{\boldsymbol{U}}_1) = m < k$ without loss of generality. Since by Remark (3) we assume the columns of $\hat{\boldsymbol{U}}_1$ are the leading $k$ non-negative eigenvectors of $\boldsymbol{A}^* + \boldsymbol{U}_1^*\boldsymbol{U}_1^{*\top} - \hat{\boldsymbol{A}} = \hat{\boldsymbol{U}}_1\hat{\boldsymbol{U}}_1^\top - \boldsymbol{B}_1$, this must imply that $\boldsymbol{A}^* + \boldsymbol{U}_1^*\boldsymbol{U}_1^{*\top} - \hat{\boldsymbol{A}} - \hat{\boldsymbol{U}}_1\hat{\boldsymbol{U}}_1^\top = -\boldsymbol{B}_1 \preccurlyeq \boldsymbol{0}$.

Plugging in the definition of $\boldsymbol{B}_1$ gives that $\frac{1}{2}\left(\hat{\boldsymbol{U}}_1\hat{\boldsymbol{U}}_1^\top - \boldsymbol{U}_1^*\boldsymbol{U}_1^{*\top} - \hat{\boldsymbol{U}}_2\hat{\boldsymbol{U}}_2^\top + \boldsymbol{U}_2^*\boldsymbol{U}_2^{*\top}\right) \succcurlyeq \boldsymbol{0}$. Thus, $\hat{\boldsymbol{U}}_1\hat{\boldsymbol{U}}_1^\top \succcurlyeq \boldsymbol{U}_1^*\boldsymbol{U}_1^{*\top} + \hat{\boldsymbol{U}}_2\hat{\boldsymbol{U}}_2^\top - \boldsymbol{U}_2^*\boldsymbol{U}_2^{*\top} \succcurlyeq \boldsymbol{U}_1^*\boldsymbol{U}_1^{*\top} - \boldsymbol{U}_2^*\boldsymbol{U}_2^{*\top}$. This contradicts the fact from Lemma (6) that $\boldsymbol{U}_1^*\boldsymbol{U}_1^{*\top} - \boldsymbol{U}_2^*\boldsymbol{U}_2^{*\top}$ has $k$ positive eigenvalues. $\square$

With this lemma, we will prove the existence of a direction of $\nabla^2\mathcal{L}^*$ with negative curvature. Instead of directly working with this matrix, we instead use the Schur complement to work with a different form.

**Theorem 5.** *(Schur Complement) Since $\nabla_A^2 \mathcal{L}^*(\hat{A}, \hat{U}) = 2I \succ 0$, $\nabla^2 \mathcal{L}^*(\hat{A}, \hat{U}) \succcurlyeq 0$ if and only if*

$$\nabla_U^2 \mathcal{L}^*(\hat{A}, \hat{U}) - \left(\nabla_A \nabla_U \mathcal{L}^*(\hat{A}, \hat{U})\right) \left(\nabla_A^2 \mathcal{L}^*(\hat{A}, \hat{U})\right)^{-1} \left(\nabla_U \nabla_A \mathcal{L}^*(\hat{A}, \hat{U})\right) \succcurlyeq 0.$$

Define $M = \nabla_U^2 \mathcal{L}^*(\hat{A}, \hat{U}) - \left(\nabla_A \nabla_U \mathcal{L}^*(\hat{A}, \hat{U})\right) \left(\nabla_A^2 \mathcal{L}^*(\hat{A}, \hat{U})\right)^{-1} \left(\nabla_U \nabla_A \mathcal{L}^*(\hat{A}, \hat{U})\right)$.

For example, when $k = 2$ and letting $U_1 = [x_1 \; x_2]$, $U_2 = [y_1 \; y_2]$, we have

$$M = \begin{bmatrix} M_{11} & M_{12} \\ M_{12}^\top & M_{22} \end{bmatrix},$$

where

$$M_{11} = \begin{bmatrix} 2B_1 + x_1 x_1^\top + \|x_1\|_2^2 & x_1^\top x_2 I + x_2 x_1^\top \\ x_2^\top x_1 I + x_1 x_2^\top & 2B_1 + x_2 x_2^\top + \|x_2\|_2^2 \end{bmatrix}$$

$$M_{12} = \begin{bmatrix} -x_1^\top y_1 I - y_1 x_1^\top & -x_1^\top y_2 I - y_2 x_1^\top \\ -x_2^\top y_1 I - y_1 x_2^\top & x_2^\top y_2 I - y_2 x_2^\top \end{bmatrix}$$

$$M_{22} = \begin{bmatrix} 2B_2 + y_1 y_1^\top + \|y_1\|_2^2 & y_1^\top y_2 I + y_2 y_1^\top \\ y_2^\top y_1 I + y_1 y_2^\top & 2B_2 + y_2 y_2^\top + \|y_2\|_2^2 \end{bmatrix}$$

For brevity, we do not include the full form of $M$ for general $k$. However, we can make an easy simplification that will allow for a much cleaner expression.

Using Corollaries (5) and (6), there is an eigenvector $z$ of $U_2^* U_2^{*\top} - U_1^* U_1^{*\top}$ with eigenvalue $\lambda \neq 0$ such that $z \in \ker\left(\hat{U}_2 \hat{U}_2^\top - \hat{U}_1 \hat{U}_1^\top\right)$. Assume without loss of generality that $\lambda > 0$, and consider $\alpha \in \mathbb{R}^{2k}$. Define the function $g(\cdot \, ; z) : \mathbb{R}^{2k} \to \mathbb{R}$ parameterized by $z$ such that $g(\alpha; z) = (\alpha \otimes z)^\top M (\alpha \otimes z)$, where we partition $\alpha = [\alpha_1; \alpha_2]$, $\alpha_1, \alpha_2 \in \mathbb{R}^k$. Then after some algebra,

$$g(\alpha; z) = \left\|\hat{U}_1 \alpha_1 + \hat{U}_2 \alpha_2\right\|_2^2 + \lambda \left(\|\alpha_1\|_2^2 - \|\alpha_2\|_2^2\right). \tag{35}$$

We prove the existence of $\alpha \in \mathbb{R}^{2k}$, $x \in \mathbb{R}^d$ such that $g(\alpha; x) < 0$ considering two different cases. Define $N^- : S_d \to \mathbb{Z}$ as the function that returns the number of strictly negative eigenvalues of its input.

**Case 1**: $N^-\left(\hat{U}_2 \hat{U}_2^\top - \hat{U}_1 \hat{U}_1^\top\right) < k$.

Using Corollary (6), we can pick $\alpha$ such that $\hat{U}_1 \alpha_1 + \hat{U}_2 \alpha_2 = 0$, $\alpha_1, \alpha_2 \neq 0$.

Because $N^-\left(\hat{U}_2 \hat{U}_2^\top - \hat{U}_1 \hat{U}_1^\top\right) < k$, $N^-\left(U_2^* U_2^{*\top} - U_1^* U_1^{*\top}\right) = k$, and $\hat{U}_2 \hat{U}_2^\top - \hat{U}_1 \hat{U}_1^\top$ and $U_2^* U_2^{*\top} - U_1^* U_1^{*\top}$ share an eigenbasis by Corollary 5, there exists $z^- \in \mathbb{R}^d$ that is a $\lambda^-$-eigenvector of $U_2^* U_2^{*T} - U_1^* U_1^{*T}$, $\lambda^- < 0$, where $z \in \ker\left(\hat{U}_2 \hat{U}_2^\top - \hat{U}_1 \hat{U}_1^\top\right)$

Then for the same choice of $\alpha$,

$$\text{sign}\left(g(\alpha; z)\right) = \text{sign}\left(\|\alpha_1\|_2^2 - \|\alpha_2\|_2^2\right)$$

$$\text{sign}\left(g(\alpha; z^-)\right) = \text{sign}\left(\|\alpha_2\|_2^2 - \|\alpha_1\|_2^2\right).$$

Then if $\|\alpha_1\|_2 \neq \|\alpha_2\|_2$, one of the above expressions is negative and thus $M$ has a negative eigenvalue. This then implies $\nabla^2 \mathcal{L}^*(\hat{A}, \hat{U}) \not\succcurlyeq 0$.

Otherwise $\|\boldsymbol{\alpha}_1\|_2 = \|\boldsymbol{\alpha}_2\|_2$. Then $g(\boldsymbol{\alpha}; \boldsymbol{z}) = 0$, but $\nabla_{\boldsymbol{\alpha}_1} g(\boldsymbol{\alpha}; \boldsymbol{z}) = \hat{\boldsymbol{U}}_1^\top \left( \hat{\boldsymbol{U}}_1 \bar{\boldsymbol{\alpha}}_1 + \hat{\boldsymbol{U}}_2 \boldsymbol{\alpha}_2 \right) - 2\lambda\boldsymbol{\alpha}_2 = -2\lambda\boldsymbol{\alpha}_2 \neq 0$. Thus $g(\boldsymbol{\alpha}; \boldsymbol{z}) = 0$ and $\nabla g(\boldsymbol{\alpha}; \boldsymbol{z}) \neq 0$ so there exists $\bar{\boldsymbol{\alpha}}$ in an infinitesimal neighborhood around $\boldsymbol{\alpha}$ where $g(\bar{\boldsymbol{\alpha}}; \boldsymbol{z}) < 0$. Thus $\boldsymbol{M}$ has a negative eigenvalue so $\nabla^2 \mathcal{L}^*(\hat{\boldsymbol{A}}, \hat{\boldsymbol{U}}) \not\succeq 0$.

**Case 2**: $N^- \left( \hat{\boldsymbol{U}}_2 \hat{\boldsymbol{U}}_2^\top - \hat{\boldsymbol{U}}_1 \hat{\boldsymbol{U}}_1^\top \right) = k$.

Define $m = \dim \left( \operatorname{im}(\hat{\boldsymbol{U}}_1) \cap \operatorname{im}(\hat{\boldsymbol{U}}_2) \right)$. By Corollary 6, $m \geq 1$, so we can select orthogonal matrix $\boldsymbol{\Gamma} \in O_k$ such that $\hat{\boldsymbol{U}}_2 \boldsymbol{\Gamma} \boldsymbol{e}_1 \in \left( \operatorname{im}(\hat{\boldsymbol{U}}_1) \cap \operatorname{im}(\hat{\boldsymbol{U}}_2) \right)$. Define $\boldsymbol{y} = \hat{\boldsymbol{U}}_2 \boldsymbol{\Gamma} \boldsymbol{e}_1$.

Clearly for any $\boldsymbol{B} \in S_d$ and $\boldsymbol{R} \in S_d^+$, $N^-(\boldsymbol{B}) \geq N^-(\boldsymbol{B} + \boldsymbol{R})$. Then since $N^- \left( -\hat{\boldsymbol{U}}_1 \hat{\boldsymbol{U}}_1^\top \right) = k$ by Lemma (7), we have that

$$k = N^-(-\hat{\boldsymbol{U}}_1 \hat{\boldsymbol{U}}_1^\top) \geq N^-(\boldsymbol{y}\boldsymbol{y}^\top - \hat{\boldsymbol{U}}_1 \hat{\boldsymbol{U}}_1^\top) = N^- \left( \left( \hat{\boldsymbol{U}}_2 \boldsymbol{\Gamma} \boldsymbol{e}_1 \right) \left( \hat{\boldsymbol{U}}_2 \boldsymbol{\Gamma} \boldsymbol{e}_1 \right)^\top - \hat{\boldsymbol{U}}_1 \hat{\boldsymbol{U}}_1^\top \right)$$

$$\geq N^- \left( \left( \hat{\boldsymbol{U}}_2 \boldsymbol{\Gamma} \right) \left( \hat{\boldsymbol{U}}_2 \boldsymbol{\Gamma} \right)^\top - \hat{\boldsymbol{U}}_1 \hat{\boldsymbol{U}}_1^\top \right) = N^- \left( \hat{\boldsymbol{U}}_2 \hat{\boldsymbol{U}}_2^\top - \hat{\boldsymbol{U}}_1 \hat{\boldsymbol{U}}_1^\top \right) = k,$$

Thus, $N^-(\boldsymbol{y}\boldsymbol{y}^\top - \hat{\boldsymbol{U}}_1 \hat{\boldsymbol{U}}_1^\top) = k$. But, since $\boldsymbol{y} \in \operatorname{im}(\hat{\boldsymbol{U}}_1)$, $\operatorname{rank} \left( \boldsymbol{y}\boldsymbol{y}^\top - \hat{\boldsymbol{U}}_1 \hat{\boldsymbol{U}}_1^\top \right) = k$. Thus,

$$\boldsymbol{y}\boldsymbol{y}^\top - \hat{\boldsymbol{U}}_1 \hat{\boldsymbol{U}}_1^\top \preccurlyeq 0. \tag{36}$$

Take $\boldsymbol{\alpha}$ such that $\hat{\boldsymbol{U}}_1 \boldsymbol{\alpha}_1 = -\boldsymbol{y}$ and $\boldsymbol{\alpha}_2 = \boldsymbol{\Gamma} \boldsymbol{e}_1$. Then

$$\boldsymbol{y}_1 \boldsymbol{y}_1^\top - \hat{\boldsymbol{U}}_1 \hat{\boldsymbol{U}}_1^\top = \left( \hat{\boldsymbol{U}}_1 \boldsymbol{\alpha} \right) \left( \hat{\boldsymbol{U}}_1 \boldsymbol{\alpha} \right)^\top - \hat{\boldsymbol{U}}_1 \hat{\boldsymbol{U}}_1^\top \tag{37}$$

$$= \hat{\boldsymbol{U}}_1 \left( \boldsymbol{\alpha}_1 \boldsymbol{\alpha}_1^\top - \boldsymbol{I} \right) \hat{\boldsymbol{U}}_1^\top \preccurlyeq 0. \tag{38}$$

Therefore $\|\boldsymbol{\alpha}_1\|_2 \leq 1$.

Then $g(\boldsymbol{\alpha}; \boldsymbol{z}) = \left\| \hat{\boldsymbol{U}}_1 \boldsymbol{\alpha}_1 + \hat{\boldsymbol{U}}_2 \boldsymbol{\alpha}_2 \right\|_2^2 + \lambda \left( \|\boldsymbol{\alpha}_1\|_2^2 - \|\boldsymbol{\alpha}_2\|_2^2 \right) = \lambda \left( \|\boldsymbol{\alpha}_1\|_2^2 - 1 \right) \leq 0$.

If $g(\boldsymbol{\alpha}; \boldsymbol{z}) < 0$, then we are done. Otherwise, $g(\boldsymbol{\alpha}; \boldsymbol{z}) = 0$. Then the same analysis from Case 1 will show that $\nabla g(\boldsymbol{\alpha}; \boldsymbol{z}) \neq \boldsymbol{0}$, so there exists $\bar{\boldsymbol{\alpha}}$ in an infinitesimal neighborhood around $\boldsymbol{\alpha}$ where $g(\bar{\boldsymbol{\alpha}}; \boldsymbol{z})$ is strictly negative. This then implies our desired result. $\square$

### B.6 Derivation of Equation (8)

Recall our generative model where each input sample $\boldsymbol{x} \in \mathbb{R}^d$ satisfies $\mathbb{E}[\boldsymbol{x}] = \boldsymbol{0}$ and $\mathbb{E}\left[\boldsymbol{x}\boldsymbol{x}^\top\right] = \sigma_x^2 \boldsymbol{I}_d$, each noise sample is generated independently of $\boldsymbol{x}$ as $\boldsymbol{\epsilon} \sim \mathcal{N}(\boldsymbol{0}, \sigma_\epsilon^2 \boldsymbol{I}_d)$, and $\boldsymbol{y} = (\boldsymbol{A}^* + \boldsymbol{R}_t^*)\boldsymbol{x} + \boldsymbol{\epsilon}$. Then,

$$
\begin{aligned}
2\mathbb{E}[\mathcal{L}_t^1(\boldsymbol{A})] &= \mathbb{E}\left[ \|\boldsymbol{y} - \boldsymbol{A}\boldsymbol{x}\|_2^2 \right] \\
&= \mathbb{E}\left[ \|(\boldsymbol{A}^* + \boldsymbol{R}_t^* - \boldsymbol{A})\boldsymbol{x} + \boldsymbol{\epsilon}\|_2^2 \right] \\
&= \mathbb{E}\left[ \|(\boldsymbol{A}^* + \boldsymbol{R}_t^* - \boldsymbol{A}_t)\boldsymbol{x}\|_2^2 + \|\boldsymbol{\epsilon}\|_2^2 + 2\boldsymbol{\epsilon}^\top (\boldsymbol{A}^* + \boldsymbol{R}_t^* - \boldsymbol{A}_t)\boldsymbol{x} \right] \\
&= \mathbb{E}\left[ \operatorname{tr}\left( \boldsymbol{x}^\top (\boldsymbol{A}^* + \boldsymbol{R}_t^* - \boldsymbol{A}_t)^\top (\boldsymbol{A}^* + \boldsymbol{R}_t^* - \boldsymbol{A}_t)\boldsymbol{x} \right) \right] + \sigma_\epsilon^2 \\
&\quad + 2\mathbb{E}[\boldsymbol{\epsilon}]^\top (\boldsymbol{A}^* + \boldsymbol{R}_t^* - \boldsymbol{A}_t)\mathbb{E}[\boldsymbol{x}] \\
&= \mathbb{E}\left[ \operatorname{tr}\left\{ (\boldsymbol{A}^* + \boldsymbol{R}_t^* - \boldsymbol{A}_t)^\top (\boldsymbol{A}^* + \boldsymbol{R}_t^* - \boldsymbol{A}_t)\boldsymbol{x}\boldsymbol{x}^\top \right\} \right] + \sigma_\epsilon^2 \\
&= \operatorname{tr}\left\{ (\boldsymbol{A}^* + \boldsymbol{R}_t^* - \boldsymbol{A}_t)^\top (\boldsymbol{A}^* + \boldsymbol{R}_t^* - \boldsymbol{A}_t)\mathbb{E}\left[\boldsymbol{x}\boldsymbol{x}^\top\right] \right\} + \sigma_\epsilon^2 \\
&= \sigma_x^2 \operatorname{tr}\left( (\boldsymbol{A}^* + \boldsymbol{R}_t^* - \boldsymbol{A}_t)^\top (\boldsymbol{A}^* + \boldsymbol{R}_t^* - \boldsymbol{A}_t) \right) + \sigma_\epsilon^2 \\
&= \sigma_x^2 \|\boldsymbol{A}^* + \boldsymbol{R}_t^* - \boldsymbol{A}_t\|_F^2 + \sigma_\epsilon^2
\end{aligned}
$$

Thus, $\mathbb{E}[\mathcal{L}_t^1(\boldsymbol{A}_t)] = \frac{1}{2}\left(\sigma_x^2 \left\|\boldsymbol{A}^* + \boldsymbol{R}_t^* - \boldsymbol{A}_t\right\|_F^2 + \sigma_\epsilon^2\right)$. Then $\mathbb{E}[\mathcal{L}_t^{n_t}(\boldsymbol{A}_t)] = \mathbb{E}[\mathcal{L}_t^1(\boldsymbol{A}_t)]$ by linearity of expectation, so

$$\frac{1}{\sigma_x^2}\left(\mathbb{E}\left[\mathcal{L}_t^{n_t}(\boldsymbol{A}_t)\right] - \frac{\sigma_\epsilon^2}{2}\right) = \frac{1}{2}\left\|\boldsymbol{A}^* + \boldsymbol{U}_t^* \boldsymbol{U}_t^{*\top} - \boldsymbol{A}_t\right\|_F^2$$

## C  Limitations of Standard Retraining for Asymmetric Adapters

In this section we show that even if the ground truth rank-$k$ adapters $\boldsymbol{R}_t^*$ are asymmetric in our theoretical model defined in Section 3, standard retraining still fails to recover a retraining solution $\hat{\boldsymbol{A}}_{\mathrm{SR}}$ that guarantees that either $\mathrm{rank}\left(\hat{\boldsymbol{A}}_{\mathrm{SR}} - \boldsymbol{A}^*\right)$ or $\left\|\hat{\boldsymbol{A}}_{\mathrm{SR}} - \boldsymbol{A}^*\right\|$ is small.

Consider task-specific adapters $\boldsymbol{R}_t^* \in \mathbb{R}^{d\times d}$ drawn independently from a general distribution $\mathcal{D}_R$ such that $\mathrm{rank}(\boldsymbol{R}_t^*) = k$ and $\mathcal{D}_R$ is absolutely continuous with respect to the Lebesgue measure. Then since $\hat{\boldsymbol{A}}_{\mathrm{SR}} = \boldsymbol{A}^* + \frac{1}{T}\sum_{t=1}^T \boldsymbol{R}_t^*$, we immediately have that $\hat{\boldsymbol{A}}_{\mathrm{SR}}$ and $\boldsymbol{A}^*$ are far apart in terms of rank:

$$\mathrm{rank}\left(\hat{\boldsymbol{A}}_{\mathrm{SR}} - \boldsymbol{A}^*\right) = \min\{d, kT\} \quad \text{w.p. } 1.$$

As a result, the necessary adaptation rank during fine-tuning to realize the test task parameters $\boldsymbol{A}^* + \boldsymbol{R}_{T+1}^*$ is $\min\{d, k(T+1)\}$, just as in Proposition 1.

Additionally, we show that we can never guarantee that $\left\|\hat{\boldsymbol{A}}_{\mathrm{SR}} - \boldsymbol{A}^*\right\|$ is small for general adapter distributions $\mathcal{D}_R$. Let mean of the adapter distribution be $M_R = \mathbb{E}\left[\boldsymbol{R}_t^*\right]$. Since the error between $\hat{\boldsymbol{A}}_{\mathrm{SR}}$ and $\boldsymbol{A}^*$ is the average of retraining adapters $\frac{1}{T}\sum_{t=1}^T \boldsymbol{R}_t^*$, then by the law of large numbers we must have that as the number of retraining tasks $T \to \infty$, then $\hat{\boldsymbol{A}}_{\mathrm{SR}} - \boldsymbol{A}^* \to M_R$. Thus, we can only guarantee that standard retraining will approach a reasonable solution in the special and impractical case where both (i) the number of tasks approaches infinity, and (ii) the mean of the adapter distribution $M_R = \boldsymbol{0}$. If either of these conditions do not hold, then standard retraining will provably fail to recover $\boldsymbol{A}^*$.

We lastly note that even though we model each task $t$ by the additive parameterization $\boldsymbol{A}^* + \boldsymbol{R}_t^*$, we cannot absorb the mean $M_R$ into $\boldsymbol{A}^*$ to reduce to the case where $M_R = \boldsymbol{0}$ without destroying the low-rank structure of $\boldsymbol{R}_t^*$. Specifically, given a problem instance defined by tasks $\mathcal{T}_t$ parameterized by $\boldsymbol{A}^* + \boldsymbol{R}_t^*$ where $M_R = \mathbb{E}\left[\boldsymbol{R}_t^*\right]$, we could equivalently consider the tasks as parameterized by $(\boldsymbol{A}^* + M_R) + (\boldsymbol{R}_t^* - M_R)$. Then the transformed adapters $\boldsymbol{R}_t^* - M_R$ are zero-mean, but $\mathrm{rank}(\boldsymbol{R}_t^* - M_R)$ may be as large as the ambient dimension $d$, eliminating the low-rank structure needed for the computational and sample-efficiency gains of low-rank adaptation during test-time fine-tuning.

## D  Synthetic Experiments

### D.1  Linear Model

We first test our meta-learning framework on synthetic regression tasks. We consider data $\boldsymbol{y}_{t,j} = (\boldsymbol{A}^* + \boldsymbol{R}_t^*)\boldsymbol{x}_{t,j} + \boldsymbol{\epsilon}_{t,j}$ for task $t$ and $j \in [n_t]$, where we generate $\boldsymbol{x}_{t,j}$, $\boldsymbol{\epsilon}_{t,j}$, and $\boldsymbol{R}_t^*$ as in Section 3 with $\boldsymbol{x}_{t,j} \sim \mathcal{N}(\boldsymbol{0}, \boldsymbol{I})$, $\boldsymbol{\epsilon}_{t,j} \sim \mathcal{N}(\boldsymbol{0}, .01 \times \boldsymbol{I})$, and $\boldsymbol{R}_t^* = \boldsymbol{U}_t^* \boldsymbol{U}_t^{*\top}$ with each element of $\boldsymbol{U}_t^*$ sampled as a $\mathcal{N}(0,1)$ random variable. We similarly generate the entries of $\boldsymbol{A}^*$ as $\mathcal{N}(0,1)$ random variables. We set the number of samples per retraining task for each task $t$ as $n_t = N$, so each retraining task is equipped with $N$ samples. Further, we denote the number of test task samples $n_{T+1} = n$. We set $\mathcal{L}$ to be the mean squared error loss, and we run gradient descent on the standard retraining (1) and the LoRA-ML (5) objectives. For the LoRA-ML objective, we alternate between the outer step, updating the shared parameter $\boldsymbol{A}$, and the inner steps, where we update the task specific parameter $\boldsymbol{U}_t$ for each task independently. We run 3000 outer steps, where after each outer step we run 10 inner steps. We use a learning rate of $3 \times 10^{-2}$ for the outer steps and $3 \times 10^{-3}$ for the inner steps. For the standard retraining objective, we simply run gradient descent using the learning rate $3 \times 10^{-2}$ over 3000 epochs.

After recovering $\hat{A}$ during retraining, we apply the low-rank adaptation $QV^\top$ when fine-tuning to the test task. When $T = 2$, we use a rank-$3k$ adaptation during fine-tuning to account for the inexact recovery explained in Theorem 2, and otherwise use a rank-$k$ adaptation. To perform the adaptation, we run gradient descent on the LoRA objective (9) using a learning rate of $5 \times 10^{-3}$.

In each experiment we vary one hyperparameter from a fixed set of values and plot the prediction error between the recovered model and the ground truth model $\frac{1}{n} \sum_j \|y_{T+1,j} - (\hat{A} + QV^\top)x_{T+1,j}\|_2^2$. Results are averaged over 5 trials, with the shaded region showing the full range of values across trials. For each experiment, we fine-tuned for a different number of epochs, as some required more epochs to converge. The performance by epoch is displayed on the x-axis of each plot.

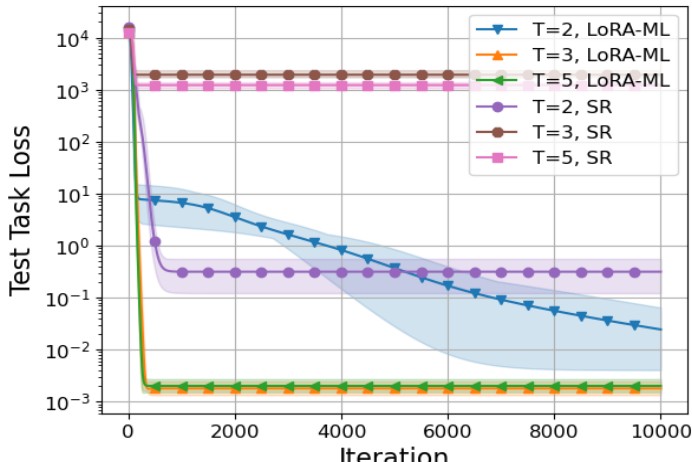

Figure 2: Linear model fine-tuning performance varying the number of retraining tasks $T$. This is an enlargement of the left subfigure of Figure 1.

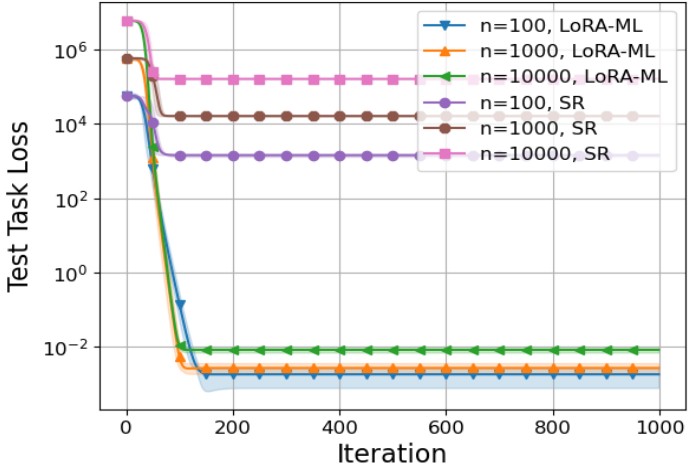

Figure 3: Linear model fine-tuning performance varying the number of samples for the test task $n$. This is an enlargement of the right subfigure of Figure 1.

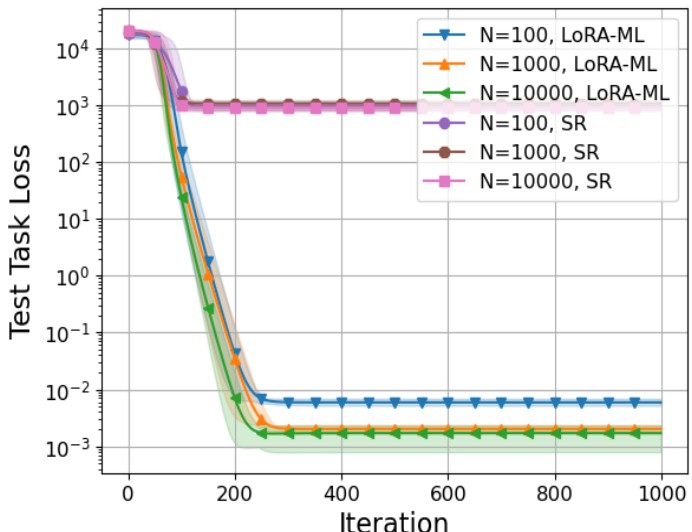

Figure 4: Linear model fine-tuning performance varying the number of samples per retraining task $N$.

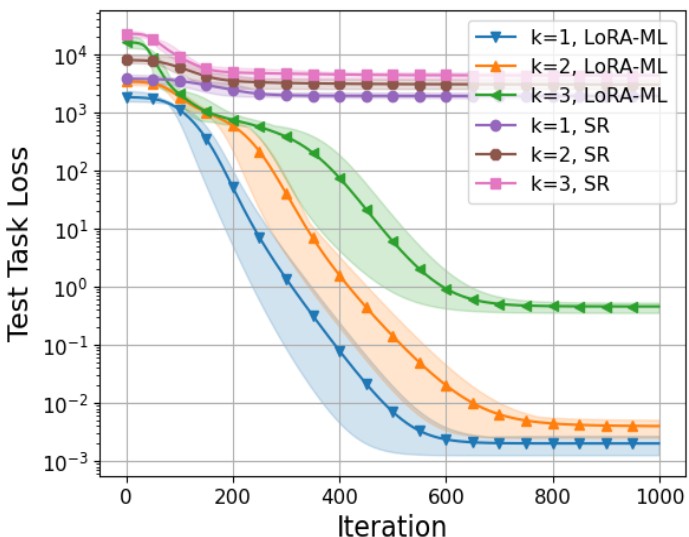

Figure 5: Linear model fine-tuning performance varying the ground truth adaptation rank $k$.

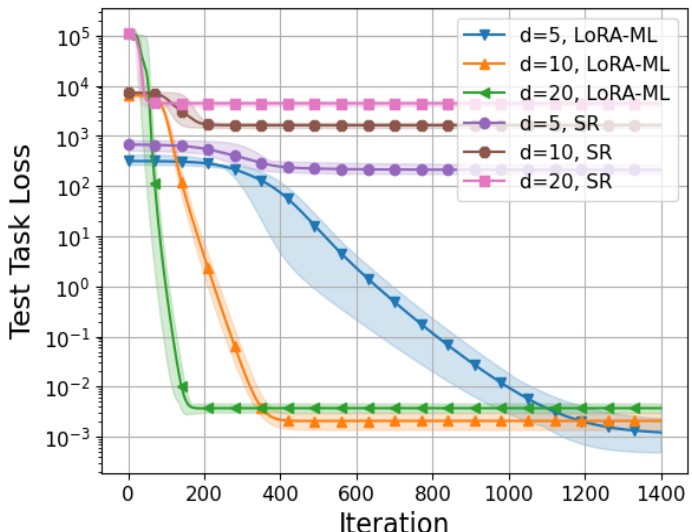

Figure 6: Linear model fine-tuning performance varying the ambient dimension $d$.

## D.2 Shallow Network

We additionally run a synthetic data experiment using data generated using a shallow network architecture. We first generate inputs $\boldsymbol{x}_{t,j} \in \mathbb{R}^d$ distributed as $\mathcal{N}(\boldsymbol{0}, \boldsymbol{I}_d)$. We then generate ground truth parameters $\boldsymbol{A}^* \in \mathbb{R}^{d \times d}$, $\boldsymbol{c}^* \in \mathbb{R}^d$, and $\boldsymbol{U}_t^* \in \mathbb{R}^{d \times k}$ for each task $t \in [T]$ by sampling the entries as $\mathcal{N}(0, 1)$ random variables. We denote $\boldsymbol{R}_t^* = \boldsymbol{U}_t^* \boldsymbol{U}_t^{*\top}$. Lastly, the outputs $y_{t,j} \in \mathbb{R}$ for task $t$ are generated as $y_{t,j} = \boldsymbol{c}^{*\top} s((\boldsymbol{A}^* + \boldsymbol{R}_t^*)\boldsymbol{x}_{t,j}) + \epsilon_{t,j}$, where $s(\cdot)$ is the element-wise sigmoid function and $\epsilon_{t,j} \sim \mathcal{N}(0, .01)$ is noise. We again define parameters $N, n$ such that $n_t = N$ for all $t \leq T$ and $n_{T+1} = n$. Setting $\mathcal{L}$ to be the mean squared error loss, we apply the AdamW optimizer with a learning rate of $1 \times 10^{-3}$ to both the standard retraining (1) and the LoRA-ML (5) objectives. After recovering $\hat{\boldsymbol{A}}$ and $\hat{\boldsymbol{c}}$ during retraining, we apply the low-rank adaptation $\boldsymbol{Q}\boldsymbol{V}^\top$ only to $\hat{\boldsymbol{A}}$ when fine-tuning to the test task. In each experiment we vary a single hyperparameter from a fixed set of values and plot the prediction error between the recovered model and the ground truth model $\frac{1}{n} \sum_{j=1}^n \left\| \hat{\boldsymbol{c}}^\top s((\hat{\boldsymbol{A}} + \boldsymbol{Q}\boldsymbol{V}^\top)\boldsymbol{x}_{T+1,j}) - \boldsymbol{c}^{*\top} s((\boldsymbol{A}^* + \boldsymbol{R}_{T+1}^*)\boldsymbol{x}_{T+1,j}) \right\|_2^2$. To mitigate the effects of outliers, we plot the median over 10 trials, where the tables that accompany each figure report the range of the central 6 values of the last epoch prediction error, clipping the two best and worst trials.

Figures 7, 8, 9, 10, and 11 again show that retraining using the LoRA-ML objective leads to much better fine-tuning performance relative to standard retraining. Figure 7 shows that the LoRA-ML objective effectively exploits the number of tasks, since fine-tuning performance improves as $T$ increases. Further, we observe in Figure 8 that fine-tuning with any number of samples after standard retraining cannot even recover the performance of fine-tuning with just 100 samples after LoRA-ML retraining. We note that the range of the results varied much more in the shallow network setting relative to the linear model experiments. Although LoRA-ML performed much better than standard retraining when looking at the median result, a few individual trials showed outlier results.

The data parameters for both synthetic experiments are summarized in Table 9. Both were performed using a single NVIDIA A40 GPU.

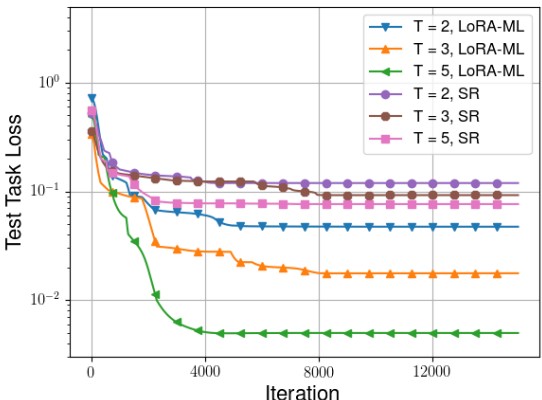

Figure 7: Shallow network fine-tuning performance while varying number of retraining tasks $T$.

Table 4: Median and central range of the test task loss after fine-tuning (smaller is better) at the last epoch for the shallow network experiment across 10 trials while varying $T$.

| Method | $T = 2$ | $T = 3$ | $T = 5$ |
|---|---|---|---|
| LoRA-ML | **0.047** (0.042, 0.127) | **0.018** (0.017, 0.028) | **0.005** (0.004, 0.024) |
| SR | 0.119 (0.104, 0.161) | 0.092 (0.086, 0.225) | 0.076 (0.046, 0.107) |

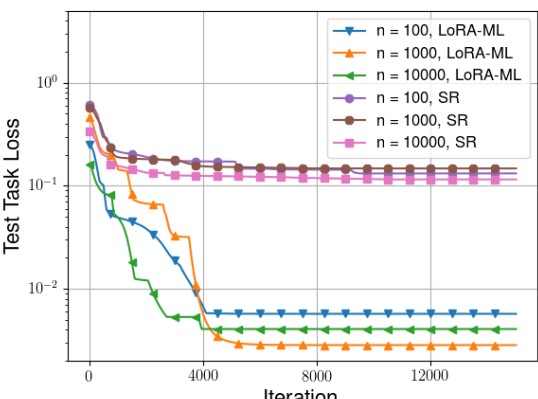

Figure 8: Shallow network fine-tuning performance while varying number of test task samples $n$.

Table 5: Median and central range of the test task loss after fine-tuning (smaller is better) at the last epoch for the shallow network experiment across 10 trials while varying $n$.

| Method | $n = 100$ | $n = 1000$ | $n = 10000$ |
|---|---|---|---|
| LoRA-ML | **0.006** (0.005, 0.034) | **0.003** (0.002, 0.061) | **0.004** (0.004, 0.017) |
| SR | 0.132 (0.114, 0.171) | 0.147 (0.109, 0.221) | 0.115 (0.112, 0.165) |

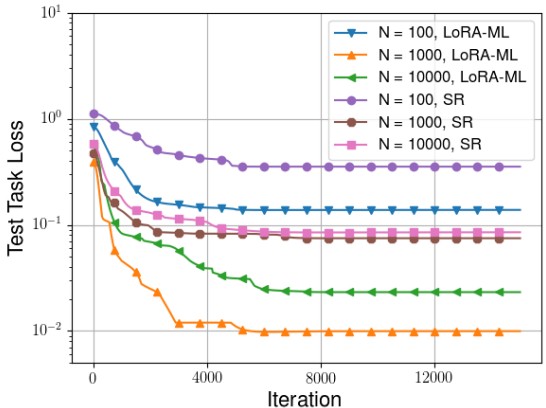

Figure 9: Shallow network fine-tuning performance while varying number of samples per retraining task $N$.

Table 6: Median and central range of the test task loss after fine-tuning (smaller is better) at the last epoch for the shallow network experiment across 10 trials while varying $N$.

| Method | $N = 100$ | $N = 1000$ | $N = 10000$ |
|---|---|---|---|
| LoRA-ML | **0.138** (0.100, 0.262) | **0.010** (0.006, 0.055) | **0.023** (0.009, 0.048) |
| SR | 0.354 (0.319, 0.646) | 0.075 (0.073, 0.092) | 0.085 (0.074, 0.108) |

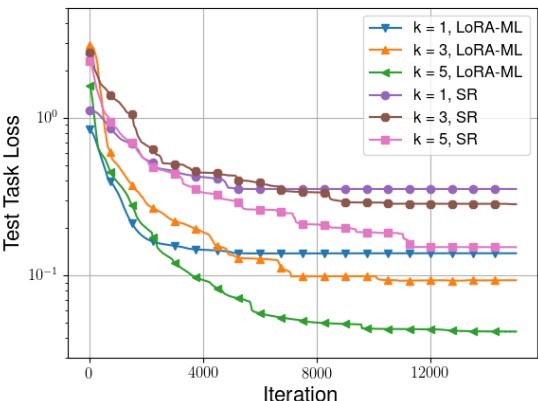

Figure 10: Shallow network fine-tuning performance while varying ground truth adaptation rank $k$.

Table 7: Median and central range of the test task loss after fine-tuning (smaller is better) at the last epoch for the shallow network experiment across 10 trials while varying $k$.

| Method | $k = 1$ | $k = 3$ | $k = 5$ |
|---|---|---|---|
| LoRA-ML | **0.138** (0.100, 0.262) | **0.093** (0.077, 0.117) | **0.044** (0.043, 0.067) |
| SR | 0.354 (0.319, 0.646) | 0.283 (0.261, 0.359) | 0.151 (0.140, 0.299) |

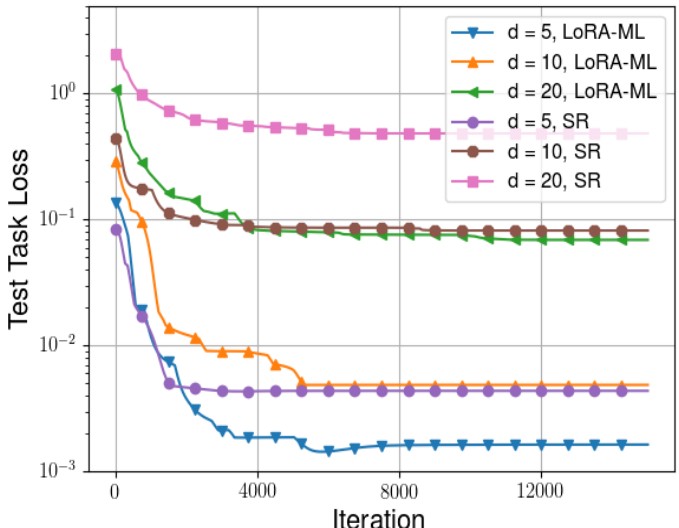

Figure 11: Shallow network fine-tuning performance while varying ambient dimension $d$.

Table 8: Median and central range of the test task loss after fine-tuning (smaller is better) at the last epoch for the shallow network experiment across 10 trials while varying $d$.

| Method | $d = 5$ | $d = 10$ | $d = 20$ |
|---|---|---|---|
| LoRA-ML | **0.002** (0.001, 0.003) | **0.005** (0.004, 0.010) | **0.069** (0.050, 0.121) |
| SR | 0.004 (0.003, 0.021) | 0.082 (0.082, 0.125) | 0.483 (0.438, 0.690) |

Table 9: Synthetic Data Parameters

| Experiment | $T$ | $n$ | $N$ | $k$ | $d$ | $\sigma_x$ | $\sigma_\epsilon$ |
|---|---|---|---|---|---|---|---|
| Linear, varying $T$ | {2,3,5} | 100 | 5000 | 1 | 10 | 1 | .1 |
| Linear, varying $n$ | 3 | {100,1000,10000} | 5000 | 1 | 10 | 1 | .1 |
| Linear, varying $N$ | 3 | 100 | {100,1000,10000} | 1 | 10 | 1 | .1 |
| Linear, varying $k$ | 3 | 100 | 1000 | {1,2,3} | 10 | 1 | .1 |
| Linear, varying $d$ | 3 | 100 | 5000 | 1 | {5,10,20} | 1 | .1 |
| Shallow Network, varying $T$ | {2,3,5} | 100 | 1000 | 1 | 10 | 1 | .1 |
| Shallow Network, varying $n$ | 3 | {100,1000,10000} | 1000 | 1 | 10 | 1 | .1 |
| Shallow Network, varying $N$ | 3 | 100 | {100,1000,10000} | 1 | 10 | 1 | .1 |
| Shallow Network, varying $k$ | 3 | 100 | 1000 | {1,3,5} | 10 | 1 | .1 |
| Shallow Network, varying $d$ | 3 | 100 | 1000 | 1 | {5,10,20} | 1 | .1 |

# E    Real Data Experiments

In this section, we describe the real data experiments in complete detail. All of the real data experiments were performed on a single NVIDIA H200 GPU.

## E.1    Vision Experiments

We use CIFAR-10 [21], and define $T = 4$ binary classification retraining tasks. Each task requires classification between consecutive CIFAR-10 class labels. Specifically, task 1 classifies between classes 1 and 2, task 2 between classes 3 and 4, etc. The test task is binary classification between classes 9 and 10. We evaluate our PEFT-ML framework using a model based on the MLP-Mixer architecture [35]. We use a depth of 1, and an embedding dimension of 512 for the patches. We compare two different adaptation methods: (i) LoRA with rank 1 adapters, and (ii) fine-tuning only the last layer. For both PEFT strategies, we evaluate three retraining strategies: (a) PEFT-ML for

Table 10: Mean test accuracies and standard errors for PEFT-ML and standard retraining methods using LoRA-based and last-layer-based fine-tuning for adapting to a subset of CIFAR-10 classes.

| LoRA Fine-Tuning | | | | Last-Layer Fine-Tuning | | |
|---|---|---|---|---|---|---|
| **Method** | **Mean** | **Std. Err.** | | **Method** | **Mean** | **Std. Err.** |
| LoRA-ML | **86.09** | 0.35 | | Last-Layer-ML | **83.90** | 0.17 |
| SR+LoRA | 85.29 | 0.17 | | SR+Last-Layer | 81.16 | 0.22 |
| Reptile+LoRA | 81.30 | 0.42 | | Reptile+Last-Layer | 72.55 | 0.39 |

each PEFT method (LoRA-ML, Last-Layer-ML), (b) standard retraining (SR), and (c) Reptile, a popular gradient-based meta-learning method. We report mean results along with standard errors across 5 trials in Table 10.

For each retraining method, we take the model from the epoch with the best average performance on the validation samples for the retraining tasks to then be fine-tuned. We also apply basic transformations of the retraining data like random cropping and flipping to reduce overfitting. During fine-tuning, we take the best accuracy on the heldout samples for the test task across epochs for each trial, and we report the average of these values in the Table 10. We summarize the hyperparameter choices in Tables 11 and 12.

Table 11: Retraining hyperparameters for vision experiments.

| Hyperparameter | Standard Retraining | LoRA-ML | Last-Layer-ML | Reptile |
|---|---|---|---|---|
| Learning Rate | $10^{-4}$ | $10^{-4}$ | $10^{-4}$ | $10^{-2}$ |
| Outer Learning Rate | N/A | N/A | N/A | $10^{-5}$ |
| Weight Decay | $10^{-5}$ | $10^{-5}$ | $10^{-5}$ | $10^{-5}$ |
| Epochs | 100 | 100 | 100 | 5 |
| Inner Epochs | N/A | N/A | N/A | 20 |
| Optimizer | Adam | Adam | Adam | Adam |
| Batch Size | 256 | 256 | 256 | 256 |
| LoRA Rank | N/A | 1 | N/A | N/A |

Table 12: Fine-Tuning hyperparameters for vision experiments

| Hyperparameter | LoRA Fine-Tuning | Last-Layer Fine-Tuning |
|---|---|---|
| Learning Rate | $5 \cdot 10^{-4}$ | $5 \cdot 10^{-4}$ |
| Weight Decay | $10^{-5}$ | $10^{-5}$ |
| Epochs | 100 | 100 |
| Optimizer | Adam | Adam |
| LoRA Rank | 1 | N/A |

Table 13: Mean accuracies ± standard error for 10 test tasks using different retraining and fine-tuning method combinations on ConvAI2.

| Algorithm | T1 | T2 | T3 | T4 | T5 | T6 | T7 | T8 | T9 | T10 | Avg |
|---|---|---|---|---|---|---|---|---|---|---|---|
| LoRA-ML | 57±2 | **41±4** | 51±4 | 50±4 | **51±4** | 30±2 | **66±5** | 47±4 | 43±4 | 38±2 | **47.4±1.9** |
| SR+LoRA | **59±5** | 31±4 | 50±7 | 40±4 | 24±5 | 20±2 | 41±10 | 36±2 | 23±5 | 26±6 | 35.0±4.1 |
| Reptile+LoRA | 45±7 | 29±5 | 35±6 | 36±2 | 19±6 | 21±3 | 28±10 | 29±7 | 21±5 | 21±7 | 28.4±5.0 |
| Last-Layer-ML | **55±2** | 27±5 | 51±5 | 44±2 | 36±4 | **25±4** | 55±4 | 43±5 | 33±4 | **34±4** | 40.2±1.5 |
| SR+Last-Layer | 41±4 | 9±5 | 29±3 | 36±5 | 28±8 | 15±4 | 31±8 | 24±5 | 15±6 | 17±6 | 24.5±4.2 |
| Reptile+Last-Layer | 35±6 | 13±4 | 18±3 | 29±3 | 19±9 | 18±4 | 15±9 | 17±5 | 15±6 | 16±4 | 19.4±4.3 |

Table 14: Mean accuracies (averaged over all tasks) ± standard error for different retraining methods at varying LoRA ranks during fine-tuning.

| Algorithm | Rank 1 | Rank 4 | Rank 8 | Rank 16 |
|---|---|---|---|---|
| LoRA-ML | **44.7 ± 0.8** | **48.6 ± 1.6** | **47.4 ± 1.9** | **48.2 ± 1.5** |
| SR+LoRA | 36.3 ± 4.1 | 35.5 ± 4.3 | 35.0 ± 4.1 | 37.1 ± 4.1 |
| Reptile+LoRA | 26.6 ± 5.8 | 27.7 ± 5.3 | 28.4 ± 5.0 | 27.8 ± 5.9 |

## E.2 Language Experiments

We use the ConvAI2 [23] dataset for the language tasks. ConvAI2 comprises conversations between two personas, where each persona is associated with a list of facts that informs their responses. We model learning the dialogue continuations of each persona as a different task. For each continuation we are given 20 candidate continuations, with one option being the correct continuation.

We retrain starting from the RoBERTa-base model [22] along with the RoBERTa tokenizer. We add a sequence classification head and use cross-entropy loss to predict whether a given continuation is in fact the correct response. We label the incorrect continuations as class 0 and the correct continuation as class 1 for training, and to perform inference we select the continuation with the highest predicted probability of being from class 1. We retrain on the ten tasks (personas) with the most data, and fine-tune on the next ten tasks (personas) with the most data. We perform 5 runs for each experiment (both retraining and fine-tuning) and report the means and the standard errors of the accuracies of the fine-tuned model on the validation sets of the fine-tuning tasks. During retraining, we save the model from the epoch which demonstrates the best average performance on the validation samples for the training tasks. We then fine-tune this retrained model and report the test task accuracies from the last fine-tuning epoch.

The hyperparameters we used are listed below in Tables 15 and 16.

Table 15: Retraining Hyperparameters for language experiments

| Hyperparameter | Standard Retraining | LoRA-ML | Last-Layer-ML | Reptile |
|---|---|---|---|---|
| Learning Rate | $5 \cdot 10^{-5}$ | $5 \cdot 10^{-5}$ | $5 \cdot 10^{-5}$ | $5 \cdot 10^{-5}$ |
| Learning Rate Schedule | Linear | Linear | Linear | Linear |
| Batch Size | 8 | 8 | 8 | 8 |
| Epochs | 10 | 10 | 10 | 10 |
| Inner Epochs | N/A | N/A | N/A | 20 |
| Optimizer | AdamW | AdamW | AdamW | AdamW |
| LoRA Rank | N/A | 8 | N/A | N/A |
| LoRA Dropout | N/A | 0.1 | N/A | N/A |
| LoRA Alpha | N/A | 16 | N/A | N/A |
| Outer Learning Rate | N/A | N/A | N/A | $10^{-4}$ |

## E.3 Asset Information

We use the CIFAR-10 [21] and ConvAI2 [23] datasets in our experiments. CIFAR-10 is publicly available but does not specify an explicit license. ConvAI2 is distributed under the Creative Commons Attribution 4.0 International (CC BY 4.0) license. We also use the MLP-Mixer [35] and RoBERTa-Base [22] model architectures, including the pretrained weights for RoBERTa. The official MLP-

Table 16: Fine-Tuning Hyperparameters for language experiments

| Hyperparameter | LoRA Fine-Tuning | Last-Layer Fine-tuning |
|---|---|---|
| Learning Rate | $10^{-4}$ | $10^{-4}$ |
| Learning Rate Schedule | Linear | Linear |
| Batch Size | 8 | 8 |
| Epochs | 10 | 10 |
| Optimizer | AdamW | AdamW |
| LoRA Rank | 8 | N/A |
| LoRA Dropout | .1 | N/A |
| LoRA Alpha | 16 | N/A |

Mixer implementation is licensed under the Apache License 2.0, while RoBERTa-Base, as accessed via Hugging Face, is licensed under the MIT License.

# F  Theory Notes

## F.1  Non-Uniqueness of Global Min for $T = 2$

Consider $T = 2$, $k = 1$, $d = 2$, $\boldsymbol{A}^* = \boldsymbol{0}$, and $\boldsymbol{u}_t^* = \boldsymbol{e}_t$ for $t = 1, 2$, where $\boldsymbol{e}_t$ is the $t^{\text{th}}$ standard basis vector. Clearly the ground truth perturbations $\boldsymbol{u}_i^*$ are orthonormal and thus linearly independent. The set of global minima of $\mathcal{L}^*$ are $(\boldsymbol{A}, \boldsymbol{U})$ such that $\boldsymbol{A} = \frac{1}{T} \sum_{t=1}^T \left( \boldsymbol{u}_t^* \boldsymbol{u}_t^{*\top} - \boldsymbol{u}_t \boldsymbol{u}_t^\top \right)$ and $\boldsymbol{u}_t \boldsymbol{u}_t^\top - \boldsymbol{u}_t^* \boldsymbol{u}_t^{*\top} - \frac{1}{T} \sum_{s=1}^T \left( \boldsymbol{u}_s \boldsymbol{u}_s^\top - \boldsymbol{u}_s^* \boldsymbol{u}_s^{*\top} \right) = \boldsymbol{0}$. It is not hard to see that a global minimum follows from any set values of $\boldsymbol{u}_1, \boldsymbol{u}_2$ such that $\boldsymbol{u}_1 \boldsymbol{u}_1^\top - \boldsymbol{u}_2 \boldsymbol{u}_2^\top = \begin{bmatrix} 1 & 0 \\ 0 & -1 \end{bmatrix}$. When properly parameterized, this system of equations defines a hyperbola where each point corresponds to a global minimum of $\mathcal{L}^*$.

## F.2  Spurious Local Minima

We observe that for $T \geq 3$, for certain tasks $\boldsymbol{U}^* = (\boldsymbol{U}_1^*, \boldsymbol{U}_2^*, \boldsymbol{U}_3^*)$, it is possible to find points $\boldsymbol{U}$ that are local minima, but not global minima. To find these points, we sample true tasks $\boldsymbol{U}^*$ from a normal distribution and use a numerical solver to find zeros of the gradient of the reduced loss

$$\hat{\mathcal{L}}(\boldsymbol{U}) = \sum_{t=1}^T \left\| \boldsymbol{U}_t \boldsymbol{U}_t^\top - \boldsymbol{U}_t^* \boldsymbol{U}_t^{*\top} - \frac{1}{T} \sum_{s=1}^T (\boldsymbol{U}_s \boldsymbol{U}_s^\top - \boldsymbol{U}_s^* \boldsymbol{U}_s^{*\top}) \right\|_F^2 .$$

Through the Schur complement argument used to prove Theorem 4, we can see that $\hat{\mathcal{L}}$ has a spurious local minimum only if $\mathcal{L}$ has a spurious local minimum.

Typically, these zeros are close to the global minimum. Occasionally, it is possible to find a point $\hat{\boldsymbol{U}}$ with gradients close to 0 and with positive definite Hessians. We then confirm that these are close to the spurious local minimum through the following argument.

Consider the function
$$r(\boldsymbol{U}) = \text{vec}(\boldsymbol{U} - \hat{\boldsymbol{U}})^\top \text{vec}(\nabla \hat{\mathcal{L}}(\boldsymbol{U})).$$

Clearly, there is a minimum of $\hat{\mathcal{L}}$ in the $\delta$-ball of $\hat{\boldsymbol{U}}$ if $r(\boldsymbol{U}) > 0$ for all $\boldsymbol{U}$ on the boundary of the $\delta$-ball. As $r$ is continuous, if for some small enough $\epsilon, \gamma > 0$ if $r(\boldsymbol{U}) > \gamma > 0$ for all $\boldsymbol{U}$ on the $\epsilon$-net of the boundary of the $\delta$-ball, then there exists a spurious local minimum in the $\delta$-ball around $\hat{\boldsymbol{U}}$. Numerically, such points and $\epsilon, \delta$, and $\gamma$ can be found which would imply that spurious local minima exist, barring any errors due to numerical computation. To confirm, we run gradient descent from this point and observe that the loss stays constant in Figure 12

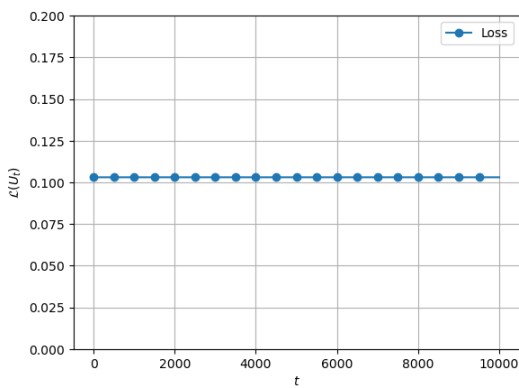

Figure 12: Loss does not decrease near these spurious local minima

# G Example Pseudocode for Minimizing (5)

---

**Algorithm 1** Meta-Adapter Training

---

1: **Input:** Tasks $\mathcal{T}_t$, $t \in [T]$, learning rate $\eta$, number of epochs $N_e$, batches per epoch $N_b$
2: **Initialize:** Model parameters $\boldsymbol{W}_0, \boldsymbol{\theta}_0^{(t)}$ for all $t = 1, \ldots, T$
3: **for** epoch $e = 1$ to $N_e$ **do**
4:     **for** $b = 1, \ldots, N_b$ **do**
5:         **for** $t = 1, \ldots, T$ **do**
6:            Load next batch $\beta_{t,b}$ from $\mathcal{T}_i$
7:            Compute gradient $\boldsymbol{g}^{(t)} = \nabla_{\boldsymbol{W}, \boldsymbol{\theta}^{(t)}} \left( \sum_{(\boldsymbol{x}, \boldsymbol{y}) \in \beta_{t,b}} \mathcal{L}((\Phi_{\text{FT}}(\boldsymbol{x}\,;\boldsymbol{W}, \boldsymbol{\theta}^{(t)}), \boldsymbol{y})) \right)$
8:            Update adapter parameters: $\boldsymbol{\theta}_{e+1}^{(t)} \leftarrow \boldsymbol{\theta}_e^{(t)} - \eta_e \boldsymbol{g}_{\boldsymbol{\theta}^{(t)}}$
9:         **end for**
10:     Update base parameters: $\boldsymbol{W}_{e+1} \leftarrow \boldsymbol{W}_e - \eta_e \sum_{t=1}^T \boldsymbol{g}_{\boldsymbol{W}}^{(t)}$
11:     **end for**
12: **end for**

---

