# OpenReview forum: "Provable Meta-Learning with Low-Rank Adaptations"
_NeurIPS.cc/2025/Conference — NeurIPS 2025 poster_

### Official Review · Reviewer_qeBR · 2025-06-25

**Clarity:** 4
**Significance:** 3
**Originality:** 4
**Rating:** 3
**Confidence:** 3

**Summary:**

This paper provides an analysis of the retraining and fine-tuning stages that foundation models require for downstream applications. The authors argue that the retraining phase should be tailored to the nature of the fine-tuning stage, which usually employs parameter-efficient fine-tuning methods. Meta-learning is proposed as the overall framework to achieve this within the retraining phase. To derive theoretical results, the authors propose a multiple linear regression setting where each ground truth is a low-rank perturbation away from a ground truth matrix $A^\star$. Given many tasks, the authors show that standard retraining (i.e., minimizing the sum of the losses for all tasks) results in a high-rank estimate and does not lead to efficient fine-tuning. In contrast, they show that the meta-learning approach in the high-sample limit is low-rank adaptable, proving that approaches such as Low-Rank Adaptation lead to fast fine-tuning.

**Questions:**

1. Could you explain why have you chosen $R_t^\star = U_t^\star (U_t^\star)^\top$. I know that it is difficult justify the theoretical setting as it is unclear what is the structure of the real-world data. However, since this choice is a bit unusual, it requires some commentary from the authors.

2. Is it possible to derive the same results with $R_t^\star \in \mathbb{R}^{d \times d}$ that is of low-rank but not necessarily PSD? I believe then the minimizer (analog of Equation 12) converges to $A^\star$ with a decaying rate in $T$. This means that standard retraining might actually work without any issues if there are many tasks. Could you comment on this claim of mine?

3. Remark 1 is not suited for many task regimes. Could you comment on how reasonable Remark 1 is in practice? In conjunction with the previous question, wouldn't your results nullify then as standard retraining benefits from a high number of tasks?

4. Can you comment a bit more in detail about the technical challenges in the paper?

**Ethical Concerns:**

["NO or VERY MINOR ethics concerns only"]

**Final Justification:**

I keep my original score as the main results of the paper are due to choices that are made to simplify the analysis. With a more forthcoming presentation, my recommendation would be "borderline accept" as it has results that are technical correct and interesting.

**Limitations:**

The work is theoretical in nature, and its applicability is limited to the setting introduced by the authors. This is evident from the text, and I consider this limitation to be acknowledged. The authors also chose to run experiments to verify if their intuitions extend to real-world settings, which is a plus. The only suggestion I have is that the authors could discuss their technical assumptions (see questions section) so that any limitations are more clearly presented.

**Quality:**

2

**Strengths And Weaknesses:**

**Strengths:**

The authors introduce an original framework to study retraining and fine-tuning. This is well-suited to theoretical insights and intuition. In terms of significance, it can lead to follow-up works on the area. Overall, the paper is written clearly.

**Weaknesses:**

The design choices in the theoretical setting are not well-discussed. This hinders the overall quality of the theoretical work. Examples include the positive-definite structure of $R_t^\star$ and if Remark 1 holds in practice. In addition, I believe the results are technically non-challenging.

---

> ### Author Rebuttal · Authors · 2025-07-31
>
> Dear Reviewer qeBR, thank you for your positive feedback regarding the clarity of our presentation and the originality of our contributions. We address your remaining questions below.
>
> **(Q1-2) Could you explain...commentary from the authors... Is it possible... claim of mine**
>
> (A1-2) Thank you for raising this important point.
>
> As you note, the structure of real-world adaptations is not well understood. We analyze the symmetric case to avoid complications in the analysis that do not primarily relate to our goal of studying meta-learning with low-rank adaptation. The LoRA fine-tuning approach we analyze is closely related to problems in matrix sensing, where assuming the ground truth matrix is symmetric can lead to a cleaner analysis. While it is not immediately clear how to extend the guarantees of our proposed LoRA-ML objective to the asymmetric case (please see our response (A4) for details on technical challenges), we emphasize that **even in the case of asymmetric adapters, standard retraining provably fails**.
>
> If the ground truth adapters $R_t^\ast = U_t^\ast V_t^\ast$ are asymmetric, the analog of Eqn. (12) shows that standard retraining (SR) recovers $ \hat{A}_{SR} = A^\ast + \frac{1}{T} \sum _{t=1} ^T R_t^\ast $. For any $T$, this estimate will deviate from $A^\ast$ by an error of large rank under a mild condition:
>
> **Theorem**: For finite $T$, assume the distribution over adapters $R_t^\ast$ is absolutely continuous with respect to the Lebesgue measure and that the ambient dimension $d$ is large enough so $kT \leq d$. Then with probability 1,
> $ rank(\hat{A}_{\text{SR}} - A^\ast) = kT $
>
> Thus for finite $T$, an analog of Proposition 1 still holds: a rank-$k(T+1)$ adapter is required to exactly fit to a test task, even though recovering $A^\ast$ would have allowed us to adapt using only a rank-$k$ adapter.
>
> For large $T$, we emphasize that in general **$\hat{A}_{SR}$ does not converge to $A^*$**. The error term $\frac{1}{T} \sum_{t=1}^T R_t^\ast$ is the average of the adapters. If we assume that $R_t^\*$ are drawn i.i.d. from a fixed distribution with mean $M_R$, then this term converges almost surely to $M_R$ as $T \to \infty$:
>
> **Theorem:** Consider ground truth adapters $R_t^*$ which are drawn from some arbitrary distribution with mean $M_R$. Then, $\lim_{T \rightarrow \infty} \hat{A}_{SR} = A^\ast +  M_R$
>
> Even this limit, SR cannot recover $A^\ast$ unless $M_R = 0$. We emphasize that this $0$-mean scenario is highly restrictive. To be effective in realistic applications, an algorithm must succeed under general task distributions.
>
> We now clarify our assumption that $U_t^\ast$ is generated according to a $0$-mean distribution in our symmetric setting, even though we have just argued against requiring such assumptions. This choice was for notational and analytical convenience, as **all of our guarantees for LoRA-ML in Section 3.2 hold deterministically for any $\\{U_t^*\\}_{t=1}^T$ as long as the diversity condition in Remark 1 is satisfied**. We use the assumed distribution over $U_t^*$ only to quantify the error of SR in an average sense in Theorem 1 and Proposition 2. These assumptions do not restrict the generality of our results on the performance of LoRA-ML, as long as the diversity condition holds.
>
> We will update the revised paper to include the above discussion and clarify (i) why we assumed symmetric adapters, (ii) why standard retraining still fails in the asymmetric case, and (iii) why the distributional parameters over $U_t^\ast$ do not affect our performance guarantees for LoRA-ML.
>
> **(Q3) Remark 1 is not suited ... number of tasks?**
>
> (A3) Thank you for this comment. To clarify, we make the technical assumption of independent adapters in Remark 1 to naturally formalize the notion of task diversity for our theoretical setting. This is analogous to prior works [1-4], which consider the problem of recovering a shared linear representation across tasks. To guarantee recovery, they require that the tasks represent a set of independent vectors on this shared subspace.
>
> We note that it is difficult to make claims about the structure of idealized adapters for real data. When adapting deep networks via LoRA across multiple layers, it is also unclear which properties of these adapters meaningfully capture task diversity. Linear independence was natural for our setting, but a more complex notion is likely required for deep networks. Further, in low diversity settings, the diminishing marginal benefit of meta-learning relative to standard retraining is expected. If the tasks are entirely unrelated, or conversely nearly identical, then meta-learning is not the appropriate tool: in the former case, no shared structure exists to exploit, and in the latter, modeling task-specific variation is unnecessary.
>
> We agree with the reviewer that the characterization of task diversity deserves further discussion, particularly in relation to the relative performance of meta-learning versus standard retraining. We will update the paper to include these additional insights, clarifying both the importance of our task diversity assumptions and their implications for performance.
>
> **(Q4) Can you comment a bit more in detail about the technical challenges in the paper?**
>
> (A4) Thank you for this comment. We are happy to discuss the technical challenges of our analysis and theoretical framework as a whole.
>
> Firstly, we note that our framework differs fundamentally from standard meta-learning analyses focused on recovering a shared linear representation across tasks [1-4]. In those settings, parameter recovery reduces to a subspace recovery problem. In contrast, our framework involves recovering $A^\ast$ from low-rank perturbations $A^\ast + U_t^\ast U_t^{\ast \top}$. Here, the measurements are additive and non-linear (algebraically) in $A^\ast$, making the recovery problem fundamentally different.
>
> The first technical challenge we faced was understanding the solution set of our LoRA-ML objective in Eqn. (14). Remarkably, we show in Theorem 3 that if (i) $T \geq 3$ and (ii) there exist distinct adapters $U_s^\ast$, $U_t^\ast$, $U_r^\ast$ with independent column spaces, then the *only* solutions minima of Eqn. (14) are the exact ground truth parameters. This is counterintuitive to prior meta-learning analyses, which require that the number of tasks scale with the intrinsic dimension of each task to ensure recovery. However, in our case, no matter the ground truth adapter rank $k$, we can always guarantee recovery given the above assumptions.
>
> The crux of the proof relies on using the set of equalities $U_1^\ast U_1^{\ast \top} = U_1 U_1^{ \top} + U_2^\ast U_2^{\ast \top} - U_2 U_2^{\top} = U_1 U_1^{ \top} + U_3^\ast U_3^{\ast \top} - U_3 U_3^{\top}$ along with the assumption that the combined columns of $U_1^\ast$, $U_2^\ast$, and $U_3^\ast$ are linearly independent to show that $im(U_1) = im(U_1^\ast)$, where $im$ denotes the matrix image. We show that for any $\\{U_i,U_i^\ast\\}_{i=1}^3$ which satisfy both of the above conditions, we must have $im(U_1) = im(U_1^\ast)$. The rest of the proof then quickly follows.
>
> The second challenge we faced related to the landscape of our proposed objective in Eqn. (14). In Theorem 4, we show with $T=2$ tasks, our objective Eqn. (14) has a strict saddle property, meaning that even though it is non-convex, every second-order stationary point (SOSP) is in fact a global minimizer.
>
> This proof is more involved, but we discuss the key steps along with the barrier we face going beyond $T=2$. We note that for a fixed $A$, the objective in Eqn. (14) reduces to separate matrix factorization objectives, but joint minimization leads to challenging interactions between the variables, which form the technical complications. When $T=2$, the analysis reduces to relating the matrix $U_1 U_1^\top-U_2U_2^\top$ to its ground truth counterpart $U_1^\ast(U_1^\ast)^\top-U_2^\ast(U_2^\ast)^\top$. For larger $T $ we cannot make this reduction. The proof follows by contradiction, where we assume the existence of an SOSP $(A,U)$ of Eqn. (14), which does not achieve the minimum function value. For such an $(A,U)$, we prove that $U_1U_1^\top-U_2U_2^\top$ shares an eigenbasis with $U_1^\ast(U_1^\ast)^\top-U_2^\ast(U_2^\ast)^\top$, but $U_1U_1^\top-U_2U_2^\top$ is rank deficient whereas $U_1^\ast(U_1^\ast)^\top-U_2^\ast(U_2^\ast)^\top$ has rank-$2k$ due to the diversity assumption. Thus, there exists a non-zero eigenvector of $U_1^\ast(U_1^\ast)^\top-U_2^\ast(U_2^\ast)^\top$ which belongs to the nullspace of $U_1U_1^\top-U_2U_2^\top$. We use this eigenvector to construct a negative direction of the Hessian, contradicting the fact that $(A,U)$ was an SOSP.
>
> We note that our analysis relied on many subtle properties of the problem, which followed from the assumption of only two tasks. Through numerical experiments, we found strong evidence that our objective in Eqn. (14) can exhibit spurious local minima in some degenerate cases when $T>2$ (please see Appendix E). This sharp transition from $T=2$ to $T=3$ underscores the unique complexity of the meta-learning landscape we consider and highlights the broader challenges of our analysis.
>
> **Concluding Remark:**
>
> We hope that our responses have adequately addressed your concerns regarding our theoretical assumptions and their practicality, as well as the technical challenges we encountered. If so, we kindly request you to consider revising your score. We are committed to addressing any additional questions you may have during the discussion phase.
>
> [1] Saunshi, N., et al. "A sample complexity separation between non-convex and convex meta-learning." ICML 2020.
>
> [2] Collins, L., et al. "Maml and anil provably learn representations." ICML 2022.
>
> [3] Thekumparampil, K., et al. "Statistically and computationally efficient linear meta-representation learning." NeurIPS 2021.
>
> [4] Du, S., et al. "Few-shot learning via learning the representation, provably." ICLR 2021.

---

> > ### Comment · Area_Chair_6kfd · 2025-08-05
> >
> > Dear Reviewer qeBR,
> >
> > The deadline of the Author-Reviewer discussion period is approaching. I would really appreciate if you can initiate the discussion with authors as soon as possible. Reviewers are expected to stay engaged in discussions, initiate them and respond to authors’ rebuttal, ask questions and listen to answers to help clarify remaining issues.
> >
> > If you have any difficulty in starting the discussion, please let us know.
> >
> > Sincerely,
> > AC

---

> > ### Comment · Reviewer_qeBR · 2025-08-06
> >
> > Thanks for your explanations.
> >
> > **Q1-Q2**: I understand that the same limitation of a high-rank offset to $A^\star$ remains. However, when the mean $M_R = 0$, wouldn't standard retraining plus fine-tuning achieve good performance? In addition, why do you think a non-zero mean for $M_R$ makes sense given your model? I think you can incorporate any non-zero mean into the $A^\star$ as the model is just additive.
> >
> > Assuming these are true, the main results of the paper are then coincidentally due to "avoid(ing) complications in the analysis that do not primarily relate to our goal of studying meta-learning with low-rank adaptation." Instead, you should **justify** this choice as it is the basis of the main result.
> >
> > **Q3**: I agree that it is unclear what the right structure is in practice or whether there is even a single one. However, I do not see why Remark 1 is analogous to [1-3] (I haven't checked [4]). My main question is that the condition $d > k T$ limits to the number of tasks, whereas [1-3] does not have any such issues. I believe this is not a mild condition and could be problematic given your setting as it is aimed at many tasks. As far as I am aware, previous work in meta-learning generally operate in regimes where $T \gg d$. I know it is very difficult, but it could be useful to justify this choice with a real-world example.
> >
> > **Q4**: Thank you for your explanations. I think the proposed structure of the adaptors is a novel and important addition to the literature over [1-4]. However, it seems like the key assumptions $d > kT$ and the PSD structure required for the results are due to the challenges of the analysis and not motivated by the real phenomena. This is acceptable to a point in theory as we lack rigorous understanding even in limited settings.

---

> > > ### Author Response · Authors · 2025-08-06
> > >
> > > Dear Reviewer qeBR,
> > >
> > > Thank you for your response.
> > >
> > > **(A1-2)** Your point is valid for asymmetric adapters if two specific conditions hold. (i) The mean of the adapters is zero, i.e., $M_R = 0$, and (ii) the number of tasks $T$ is approaching infinity. Under these two conditions, standard retraining (SR) plus fine-tuning succeeds. However, if either of these two conditions is not satisfied, then the above conclusion does not hold. Next, we explain why your suggestion that the case when $M_R$ is non-zero can be reduced to the zero-mean case may not hold in general. Specifically, we explain why the transformed adapters $R_t^\ast - M_R$ may no longer be low-rank, as $M_R = \mathbb{E}[R_t^\ast]$ which is the expectation of rank-$k$ matrices may have full rank $d$.
> > >
> > > Consider the following generative model for $R_t^\ast$. Fix two arbitrary full-rank matrices $U,V \in \mathbb{R}^{d \times d}$ with columns given by $u_i$ and $v_i$ respectively. Let each rank-$k$ adapter $R_t^\ast$ be generated by first picking 1 random index $j$ uniformly from $\\{1,…d\\}$ and set $R_t^\ast = u_j v_j^\top + Z_L Z_R^\top$, where $Z_L$ and $Z_R$ are random zero-mean Gaussian matrices in $\mathbb{R}^{d \times (k-1)}$. Then each $R_t^*$ is rank-$k$ almost surely with non-zero mean $M_R = \frac{1}{d} UV^\top$, but $rank(R_t^\ast - M_R) = d$ almost surely.
> > >
> > > Thus, even in the asymmetric case, we must consider task distributions which may have arbitrary adapter mean $M_R$ since we cannot always reduce to the case of $M_R = 0$ without destroying the original low-rank structure. Rather than enforcing any specific choice for $M_R$, which is a parameter of the task distribution, we would like to consider more general task distributions with arbitrary mean.
> > >
> > > We further note that even when $M_R = 0$, SR would need an infinite number of tasks, each with an infinite number of samples, in order to recover the desired parameter $A^\ast$. In contrast, in the symmetric case we prove that our method can exactly recover $A^\ast$ when given only $T=3$ tasks, regardless of adapter mean $M_R$. We emphasize that this is a major practical benefit, as a large number of tasks may not be available and training using fewer tasks is more computationally efficient. While in this work we do not provide guarantees for our method in the asymmetric setting, we emphasize that the key result which shows that SR provably fails with finite tasks is not simply an artifact of the symmetric adapter assumption.
> > >
> > > **(A3)** The reason that we mention the related works [1-4] when discussing our notion of task diversity assumed in Remark 1 is simply to justify the need for task diversity when analyzing meta-learning algorithms. Both our assumption in Remark 1 and the assumptions in [1-4] use the notion of linear independence of task-specific parameters to formalize task diversity. Beyond this high-level similarity, we emphasize that the structure of our model is very different from the linear representation learning setting considered in [1-4].
> > >
> > > Regarding our assumption that $k(T+1) < d$, this is mainly for notational brevity. We explain below how each of our key results only requires a less restrictive assumptions, where the bounds involving $k(T+1)$ instead become $ min \\{ k(T+1), d \\} $ when $k(T+1) < d$ no longer holds. In Proposition 1, we show that standard retraining recovers the matrix $\hat{A}_{SR}$ which incurs a rank-$kT$ error to $A^\ast$, and thus requires a rank-$k(T+1)$ adapter to fit a new, unseen test task. In the general case, we can update the bound in Proposition 1 as SR will require an adapter of rank  $min \\{ k(T+1), d \\} $ to fit the test task. Thus, even when this bound reduces to $d$, it shows that SR requires a full-rank-$d$ adapter to fit the test task, defeating the purpose of low-rank adaptation. Secondly, for Theorem 3 which shows that minimizing our LoRA-ML objective leads to recovery of $A^\ast$, we only rely on the existence of 3 independent tasks. Thus, this result holds as long as $3k < d$, which does not depend on the total number of tasks.
> > >
> > > In summary, we emphasize that our main results are not limited by the assumption that $k(T+1) < d$ and our method can robustly operate in the large task regime.
> > >
> > > **(A4)** As we mention above, we clarify that the assumption that $k(T+1) < d$ does not change our main results. Further, we recognize that the assumption of the PSD adapter structure allows us to provide guarantees for our method and is not directly motivated by a specific real-world observation. While we agree it is a natural next step to consider extending our positive results for LoRA-ML to the asymmetric case, we aim to show in our response (A1-2) that standard retraining is still inherently limited in the more general asymmetric setting.
> > >
> > > Thank you for your continued engagement. We hope this addresses your concerns and we are happy to answer any follow up questions you may have.

---

### Official Review · Reviewer_JxEW · 2025-07-02

**Clarity:** 2
**Significance:** 3
**Originality:** 3
**Rating:** 4
**Confidence:** 4

**Summary:**

This paper provides a theoretical framework based on a linear regression model to analyze why meta-learning is more suitable than standard retraining during the retraining stage for preparing a model for future low-rank adaptation.

**Questions:**

See Strengths And Weaknesses above.

**Ethical Concerns:**

["NO or VERY MINOR ethics concerns only"]

**Final Justification:**

My primary concern was the distinction between the proposed PEFT-Meta-Learning (PEFT-ML) and standard Multi-Task Learning (MTL). This concern was effectively addressed through discussions with the authors, particularly their clarification of the formulation.

While the theoretical contributions are currently limited to linear models, I acknowledge the value of the work in providing a principled perspective on the connection between meta-learning and foundation models. This perspective, in my view, is meaningful for the meta-learning community.

**Limitations:**

The authors acknowledge that the primary limitation of their work is that the theoretical analysis applies only to linear models with LoRA fine-tuning.

**Paper Formatting Concerns:**

No concerns on formatting.

**Quality:**

3

**Strengths And Weaknesses:**

**Strengths:**

1. The paper provides a principled theoretical framework to explain why meta-learning is more suitable than multi-task learning (standard retraining) for retraining stage to prepare for future low-rank adaptation.
2. The theoretical analysis looks solid.
3. The paper is well-organized.

**Weaknesses:**

1. **Unfair theoretical setting between standard retraining and meta-learning:** The comparison between meta-learning and standard retraining appears to be unfair. In the standard retraining setup, only the base model \$W\$ is trained (as in Equation 1), while in the meta-learning setup, both the base model \$W\$ and the adapter \$\theta\$ are trained (as in Equations 5 and 6). A fair comparison should involve the same model architecture and trainable parameters, with only the training algorithm differing.

2. **Unclear definition and distinction between meta-learning and multi-task learning objectives, which could lead to unreliable conclusions:** Equation 14 is presented as the meta-learning objective. However, it is unclear how this differs from multi-task learning when both the base model and the adapter are trained. In traditional meta-learning, a key characteristic is the bi-level optimization on the same set of parameters, whereas the current formulation appears to reduce to a multi-task learning objective. If my interpretation is correct, the paper may actually be comparing multi-task learning on both the base model and adapter vs. multi-task learning on the base model only, rather than demonstrating a true meta-learning strategy.

3. **Limited theoretical scope:** The theoretical analysis is confined to linear regression tasks. It remains unclear how well the conclusions generalize to more complex models.

4. **Counterintuitive result in Theorem 3:** Theorem 3 suggests that there is an absolute condition on the number of tasks needed for generalization in meta-learning. However, empirical studies in the literature [1,2,3] often highlight the importance of *task diversity* for generalization performance, rather than merely the number of tasks. The paper should provide a discussion clarifying this discrepancy. Additionally, it would be valuable to explain the practical implications of the theorem for task construction in real meta-learning scenarios.


[1] Meta-Learning with Fewer Tasks through Task Interpolation, ICLR 2022.

[2] Data Augmentation for Meta-Learning, ICML 2021.

[3] Meta-Learning Requires Meta-Augmentation, NeurIPS 2020.

---

> ### Author Rebuttal · Authors · 2025-07-31
>
> Dear Reviewer JxEW, thank you for recognizing the strength of our analysis as well as the organization of our presentation. We address your questions below.
>
> **(Q1) Unfair theoretical setting between standard retraining and meta-learning...**
>
> (A1) Thank you for this question which touches on a central difference in the way meta-learning models the different tasks relative to learning them in aggregate, which we refer to as standard retraining (SR).
>
> We consider the problem of preparing a model for efficient adaptation to an unseen, downstream task, using a set of accessible representative tasks drawn from the same task distribution. Meta-learning is a well-established framework for this problem, as it explicitly aims to learn model parameters that can be "easily" adapted. A key aspect of any meta-learning method is defining this "ease" of adaptability. In MAML, for example, the adaptation mechanism is chosen to be a few steps of gradient descent on the task-specific loss. As you mention, this adaptation does not introduce additional parameters, as MAML simply uses a different training algorithm on the same parameter vector compared to standard retraining, which minimizes the average task loss directly.
>
> We focus on parameter-efficient fine-tuning (PEFT) adaptation methods, as these are standard for fine-tuning large-scale models. A common PEFT paradigm is to freeze the given model weights and introduce a small set of additional trainable parameters, like LoRA adapters. To incorporate PEFT with meta-learning, we must include these additional parameters to form the inner meta-learning loss, just as MAML devises its inner loss as the result of applying a fixed number of gradient steps on the task-specific loss. Therefore, when we compare our PEFT-based meta-learning framework to SR, we are comparing two fundamentally different modeling strategies: (i) our approach, which models each task as a PEFT adaptation of a shared model, and (ii) SR, which learns a single parameter by aggregating all task data. Because PEFT methods can rely on additional task-specific parameters for fine-tuning, the fine-tuning adaptation procedure and the parameterization are inseparable.
>
> However, to compare our framework to a method which models all tasks in the aggregate but also includes the same number of trainable parameters as our proposed method, we analyze the following method, which we denote SR+. Consider our theoretical setting where we are given $T$ tasks, where each task $t$ is equipped with the infinite-sample loss function $\mathcal{L}_t^\ast$ defined by the shared parameter $A^\ast \in \mathbb{R}^{d \times d}$ along with the task-specific rank-$k$ adapter with factor $U_t^\ast \in \mathbb{R}^{d \times k}$:
>
> \begin{equation*}
>     \mathcal{L}_t^\ast (A) = \|| A - A^\ast - U_t^\ast U_t^{\ast \top} \||_F^2
> \end{equation*}
>
> We define the SR+ method to minimize $ \sum_t \mathcal{L}_t^\ast $ just like SR, except it uses the parameterization $A + U U^\top$, where $A \in \mathbb{R}^{d \times d}$ aims to recover the ground truth matrix $A^\ast$, while $U \in \mathbb{R}^{d \times kT}$ represents a rank $kT$ adaptation applied to each task. Formally, SR+ minimizes:
>
> $$
> \mathcal{L}_{SR+} (A,U) =  \sum _{t=1}  ^T  \|| A + U U^\top - A^\ast - U_t^\ast U_t^{\ast \top} \||_F^2
> $$
>
> Thus, SR+ uses the same number of parameters as LoRA-ML but still models the tasks in aggregate like SR. We then directly show that it is impossible to recover $A^\ast$ by minimizing this SR+ objective.
>
> **Theorem:** Let $\hat{U}$ be any $d \times kT$ matrix. Then for the matrix $\hat{A} = -\hat{U} \hat{U}^\top + A^\ast + \frac{1}{T} \sum_{t=1}^T U_t^\ast U_t^{\ast \top}$, we have that $(\hat{A}, \hat{U})$ globally minimizes $\mathcal{L}_{\text{SR+}}$
>
> Thus $A^\ast$ is completely unidentifiable by the SR+ loss, as any value of $\hat{U}$ corresponds to a global minimizer, whereas the only value that leads to recovery $\hat{A} = A^\ast$ is when $\hat{U}$ contains the properly scaled columns of the ground truth factors $U_t^\ast$. However, this is impossible to recover, since any of the infinite values of $\hat{U} \in \mathbb{R}^{d \times kT}$ can result in global minimization. This is in contrast to our LoRA-ML objective, where we show that global minimization always leads to exact recovery of the ground truth parameter $A^\ast$, with a rank $2k$ error only when $T=2$.
>
> We hope this clarifies the rationale behind our setup and addresses your concern about a fair comparison between our method and standard retraining. We are happy to provide further details if helpful.
>
> **(Q2) Unclear definition and distinction between meta-learning and multi-task learning...rather than demonstrating a true meta-learning strategy.**
>
> (A2) Thank you for this comment. We clarify the interpretation of our method meta-learning rather than standard multi-task learning.
>
> A key feature of meta-learning is the bi-level optimization structure, where the outer objective minimizes a sum of task-specific losses over shared parameters, where each task-specific loss is itself a minimization problem over a task-specific adaptation. This precisely underlies our derivation: in the linear case in Equation (13), the inner problem minimizes the loss for each task after fine-tuning with the task-specific adapter $U_t$, and the outer problem optimizes the shared base model parameter $A$ to improve post-adaptation performance.
>
> Because the LoRA adaptation introduces new parameters to update, the entire meta loss can be rewritten in Eqn. (14) as the joint minimization over $A$ and $\\{ U_t\\}_{t=1}^T$. While this resembles a multi-task learning objective, it is the outcome of a meta-learning formulation with a specific adaptation rather than simply a multi-task loss.
>
> As you mention, in prior model agnostic meta-learning methods such as MAML or Reptile, the inner adaptation runs full gradient steps over all model parameters. Our method aligns with model-based meta-learning, where the focus is on structuring the model (e.g., via PEFT modules) to enable fast adaptation. This explains why our meta-objective can be equivalently expressed in the form  of Eqn. (14), since the inner problem introduces new parameters to update with respect to the outer problem.
>
> **(Q3) Limited theoretical scope..**
>
> (A3) We recognize that our theoretical contributions only directly apply to adapting linear models with LoRA. While this setting allows us to derive precise and interpretable guarantees within the new paradigm of meta-learning via PEFT-based adaptation, we acknowledge that these results do not directly extend to the non-linear or deep networks where we ultimately aim to apply these ideas.
>
> To address the practical setting, we include a set of empirical experiments across vision and language tasks using deep networks, demonstrating that the key principles established theoretically, such as improved adaptation via retraining with our proposed framework, continue to hold in practice. We agree that extending our framework to more complex models is a natural and interesting next step for future work.
>
> **(Q4) Counterintuitive result in Theorem 3...**
>
> (A4) Thank you for this comment.
>
> Theorem 3 states that, given infinite samples per task, access to 3 independent tasks guarantees exact recovery of the ground truth parameters, no matter the rank of the ground truth adapters. This contrasts with prior meta-learning analyses [1-4] that consider linear representation learning settings, where the number of tasks needed for ground truth parameter recovery depends on the intrinsic task dimension. This difference arises since these previously analyzed settings correspond to subspace recovery problems, but our setting requires estimating a matrix $A^\ast$ from low-rank perturbations $A^\ast + U_t^\ast U_t^{\ast \top}$. For more details on the technical challenges of our setting, please see our response (A4) to Reviewer qeBR.
>
> In terms of the empirical observations that highlight the importance of task-diversity rather than just the number of tasks, we note that this is in fact supported by our result in Theorem 3 which shows that recovery fails with many dependent tasks but succeeds with as few as three independent ones.
>
> Regarding practical task construction, we first note that the result of Theorem 3 is specific to LoRA-based meta-learning for linear models. For other PEFT strategies or model architectures, 3 tasks may not be enough to guarantee exact parameter recovery. Further, in practical settings, we only have access to tasks with a finite number of samples, whereas Theorem 3 considers the infinite sample case. With only finite samples, each task does not give an exact measurement of the true parameters, so increasing the number of tasks can effectively denoise the parameter estimate by providing additional measurements of the true parameters. Thus, it is likely beneficial to construct a larger set of diverse, high-quality tasks in practice.
>
> We will add this discussion relating the result in Theorem 3 to the empirical studies you mention as well as its implications for practical task construction to the revised paper.
>
> **Concluding Remark**
>
> We hope that our responses have adequately addressed your concerns regarding the theoretical characterization of our framework and the guarantees we provide, along with how they relate to practical insights. If so, we kindly request you to consider revising your score. We are committed to addressing any additional questions you may have during the discussion phase.
>
> [1] Saunshi, N., et al. ``A sample complexity separation between non-convex and convex meta-learning." ICML 2020.
>
> [2] Collins, L., et al. ``Maml and anil provably learn representations." ICML 2022.
>
> [3] Thekumparampil, K., et al. ``Statistically and computationally efficient linear meta-representation learning." NeurIPS 2021.
>
> [4] Du, S., et al. ``Few-shot learning via learning the representation, provably." ICLR 2021.

---

> > ### Comment · Reviewer_JxEW · 2025-08-04
> >
> > Thank you for your detailed response. However, I still have a few concerns:
> >
> > 1. **Difference between the PEFT-based meta-learning and SR+  in formulation.**
> >    Could you please clarify the difference between the objective function of SR+ (as presented in your response A1, i.e., $\mathcal{L}_{\rm SR+}$) and the objective of PEFT-based meta-learning presented in Eq. (14), from a formulation perspective?
> >
> > 2. **Setting of PEFT-based meta-learning. & Difference between PEFT-based meta-learning and multi-task learning in formulation.**
> >
> >    (i) In your proposed PEFT-based meta-learning framework, it appears that the PEFT modules are updated in the inner loop, while only the base model is updated in the outer loop. Why do you not update the PEFT modules in the outer loop as well? In standard meta-learning setups, the parameters updated in the inner and outer loops are typically the same, so that the meta parameters can be quickly adapted to new tasks.
> >
> >    (ii) Since the parameters updated in your inner and outer loops are disjoint, it seems that this framework (Eq.14) reduces to a multi-task learning in formulation, rather than meta-learning. I would recommend that the authors provide a more formal and explicit comparison between the PEFT-based meta-learning objective and a standard multi-task learning objective, from a formulation perspective.
> >
> > 3. **Discussion of related work.**
> >    Given that this paper is theory-heavy, I recommend that the authors include a discussion of additional related works that practically combine meta-learning and PEFT methods (e.g., \[1, 2, 3]). This would help readers better understand the practical motivations, setting, and broader application.
> >
> > [1] Context-Aware Meta-Learning. ICLR 2024.
> >
> > [2] LoRA Recycle: Unlocking Tuning-Free Few-Shot Adaptability in Visual Foundation Models by Recycling Pre-Tuned LoRAs. CVPR 2025.
> >
> > [3] Meta-Adapters: Parameter Efficient Few-shot Fine-tuning through Meta-Learning. AutoML 2022.

---

> > > ### Author Response · Authors · 2025-08-04
> > >
> > > **Difference between SR+ and Eqn. (14)**
> > >
> > > In Eqn. (14), we model each task as a rank-$k$ perturbation of a shared parameter $A^\ast$. This formulation is motivated by the widespread empirical success of LoRA in adapting foundation models to downstream tasks. Specifically, Eqn. (14) fits task $t$ using parameters $A + U_t U_t^\top$, where $A \in \mathbb{R}^{d \times d}$ is a shared parameter across tasks and $U_t \in \mathbb{R}^{d \times k}$ is a task-specific rank-$k$ adapter.
> > >
> > > To address your concern that comparing our method to standard retraining (SR) may be unfair due to parameter count, we introduce SR+ to show that the key advantage of our approach lies not in using more parameters, but in modeling each task as a distinct PEFT perturbation of a shared parameter. SR+ matches our method in trainable parameter count but still models tasks in aggregate. SR+ fits to task $t$ using $A + U U^\top$, where $U \in \mathbb{R}^{d \times kT}$ is a rank-$kT$ adapter which is the same for all $T$ tasks. Both SR+ and our method use the same number of parameters, but SR+ models all uniformly, similar to SR. Even though SR+ includes an additional adapter $UU^\top$ compared to SR, we show in our previous response that it still fails to recover the ground truth parameters. We clarify the differences below:
> > >
> > > | Method  | Params | Num. Params.   | Task-$t$ Loss |
> > > |-|-|-|-|
> > > | SR | $A$ | $d^2$ | $\|A - A^\ast - U_t^\ast U_t^{\ast \top} \|_F^2$ |
> > > | LoRA-ML | $A, \\{U_t\\}_{t=1}^T$ | $d^2 + dkT$  | $ \|A + U_t U_t^\top -  A^\ast - U_t^\ast U_t^{\ast \top} \| _F ^2 $|
> > > | SR+| $A,U$ | $d^2 + dkT$  | $\| A + U U^\top -  A^\ast - U_t^\ast U_t^{\ast \top} \| _F ^2 $| |
> > >
> > > **Difference to Multi-Task Learning**
> > >
> > > (i) We would like to clarify that our framework provides an *objective function* (Eqns. 5, 14) that seeks a shared parameter achieving low loss on each task after task-specific adaptation, rather than prescribing a particular optimization algorithm. While the meta-loss has a natural “inner” and “outer” structure, we are free to apply different optimization strategies to minimize it. For instance, in our vision and language experiments, we jointly optimize the meta-loss in Eqn. (5) over both the shared parameter $W$ and task-specific adapters $\theta^{(t)}$, without separate inner or outer updates. This approach achieves strong performance in adapting to new tasks. We are grateful to the reviewer for raising this point, and emphasize that the choice of optimization procedure is independent of the formulation, which is grounded in PEFT-based adaptation within a meta-learning framework.
> > >
> > > (ii) We agree that our objective in Eqn. (14) can be viewed as a form of multi-task learning, as it fits each retraining task using a combination of shared and task-specific parameters. This is conceptually similar to approaches like Meta-Adapters (Bansal et al.), which learn a shared base parameter that enables low task loss after injecting additional task-specific trainable layers within the network. In this sense, PEFT-based meta-learning objectives can resemble multi-task ones, and we thank the reviewer for highlighting this connection.
> > >
> > > However, we emphasize that the key difference between multi-task learning and meta-learning lies in the learning goal. Multi-task learning aims to improve performance on a set of related tasks by training jointly on their data, and it is evaluated on new *samples* from the same tasks. In contrast, meta-learning seeks to extract shared task structure to enable *fast adaptation* to entirely new tasks, and is evaluated by adaptation performance on new, unseen *tasks*. Our framework is designed for the latter setting.
> > >
> > > If our analysis focused solely on minimizing Eqn. (14), it would fall under the multi-task paradigm. However, our interest lies in the *adaptability* of the recovered parameters. Corollaries 2 and 4 quantify the sample complexity of adapting to new tasks, and Theorem 2 shows that while Eqn. (14) can be perfectly minimized with two tasks (satisfying the multi-task objective), this does not guarantee optimal adaptation due to a rank-$2k$ ambiguity in recovering $A^\ast$. Thus, while the form of our objective resembles multi-task learning, our framework and analysis apply to the meta-learning setting.
> > >
> > > **Related Work**
> > >
> > > Thank you for pointing out these works. We are familiar with Meta-Adapters ([3]), as in our Related Work (Line 96) we mention they show empirical benefits of using meta-learning with the adaptation method of inserting task-specific trainable layers into the network. We will add citations to the other works and mention that “Context-Aware Meta-Learning” trains a sequence model to predict the class of a test image conditioned on the embeddings of support examples, while “LoRA Recycle...” reverse-engineers fine-tuning data from existing LoRA adapters and trains a base model that fits these inferred samples via tuning-free adaptation.
> > >
> > > We are happy to address any remaining questions.

---

> > > > ### Comment · Reviewer_JxEW · 2025-08-05
> > > >
> > > > Thank you for your detailed response. It has effectively addressed my concerns, especially regarding the difference between PEFT-ML and MTL. Therefore, I would like to revise my score to 4.

---

### Official Review · Reviewer_eo3S · 2025-07-03

**Clarity:** 3
**Significance:** 3
**Originality:** 3
**Rating:** 5
**Confidence:** 3

**Summary:**

This paper introduces a framework for meta-learning with low-rank adaptations, aiming to enhance the adaptability of foundation models to unseen tasks. ​ The authors argue (and present proofs) that standard retraining methods aggregate task data to optimize model parameters, but they fail to account for the downstream fine-tuning, often resulting in suboptimal fine-tuning. ​ The proposed framework explicitly incorporates the fine-tuning mechanism during retraining to prepare the model for efficient adaptation to new tasks. ​The paper offers theoretical insights and guarantees for the proposed method, and tests the framework in three settings, demonstrating performance improvements over standard retraining. ​

**Questions:**

1. Do you have insights for why Reptile outperforms on some tasks, given the theoretical guarantees?
2. How did you conclude that "these spurious minima are almost never found in practice?" I do not think the presented experiments suffice for this claim IMO.

**Ethical Concerns:**

["NO or VERY MINOR ethics concerns only"]

**Final Justification:**

The rebuttal acknowledges dataset scale, adds a simple rank ablation, and clarifies the claim about spurious minima with synthetic evidence and planned plots. A broader evaluation would strengthen the case; however, the gains across settings are convincing, so I will maintain the original rating.

**Limitations:**

yes

**Quality:**

3

**Strengths And Weaknesses:**

Strengths:
1. The new framework combines PEFT with meta-learning to address the limitations of standard retraining methods. ​
2. The paper provides rigorous theoretical guarantees and insights for the proposed framework, demonstrating that the proposed method outperforms standard retraining in terms of adaptability to unseen tasks. These results are supported by proofs and detailed analysis (under some theoretical constraints).
3. Empirical results show that the proposed framework outperforms standard retraining and other meta-learning methods (e.g., Reptile) across synthetic, vision, and language tasks. ​
4. The framework is tested with two PEFT strategies (LoRA and last-layer fine-tuning), demonstrating flexibility across different adaptation methods.

Weakness:
1. The results are not tested on strong datasets (CIFAR is a small dataset, and ConvAI2 does not properly capture task diversity IMO). Additionally, the paper assumes "task diversity" and "linear independence" of task-specific adaptations, which, of course, do not hold in real-world scenarios where tasks can be highly correlated; therefore, I think the method should be tested on more realistic datasets.
2. Lack of ablation for the impact of different ranks or architectures.
3. This can be subjective, while I appreciate the formal notations and explicit definitions, I do believe many parts of the write-up can be simplified (or written in simpler notation).

---

> ### Author Rebuttal · Authors · 2025-07-31
>
> Dear Reviewer eo3s, thank you for highlighting our rigorous guarantees as well as our practical experiments. We address your remaining questions below.
>
> **(Q1) The results are not tested on strong datasets (CIFAR is a small dataset, and ConvAI2 does not properly capture task diversity IMO). Additionally, the paper assumes "task diversity" and "linear independence" of task-specific adaptations, which, of course, do not hold in real-world scenarios where tasks can be highly correlated; therefore, I think the method should be tested on more realistic datasets.**
>
> Thank you for raising this important point regarding the connection between our experimental setup and the assumptions made in our theoretical analysis. Regarding our theoretical framework, we note that assuming some level of task diversity is standard for meta-learning analyses [1-4], as if the tasks are extremely similar, modeling them individually provides little benefit. In our setting, the linear independence among the column spaces of the adapters forms a natural way to model this notion of task diversity.
>
> We acknowledge that this may not hold in practice for real data, and we agree that evaluating our method on additional large-scale and realistic datasets would further strengthen the empirical evidence for its effectiveness. While our goal is to develop a theoretically principled framework for preparing a model for efficient test-time adaptation using PEFT, we recognize that larger-scale evaluation is a natural next step for future work.
>
> **(Q2) Lack of ablation for the impact of different ranks or architectures.**
>
> Thank you for this suggestion. While we did not have time to incorporate additional architectures during the rebuttal period, we include an ablation for different ranks during the LoRA-ML retraining phase below for the language modeling task, where we aim to learn the responses of different personas in the ConvAI2 dataset. In Table 2 of the paper, we show results using rank-8 adaptations for LoRA-ML along with various other methods. Below, we show how changing the rank affects performance. As a baseline, we include standard retraining followed by rank-$k$ LoRA (SR+LoRA), where we use the same rank for both methods in each column.
>
> **Test Task Prediction Acc. (↑)**
> |Rank|1|4|8|12|16|
> |-|-|-|-|-|-|
> |LoRA-ML|0.37|0.36|0.39|0.31|0.37|
> |SR+LoRA|0.20|0.20|0.26|0.19|0.20|
>
> We report the mean accuracy over the 10 test tasks using 5 random trials. The results demonstrate that LoRA-ML outperforms SR across a variety of ranks, demonstrating the effectiveness of our framework and its insensitivity to the adapter rank hyperparameter. We will include these results in the revised version of the paper.
>
> **(Q3) This can be subjective, while I appreciate the formal notations and explicit definitions, I do believe many parts of the write-up can be simplified (or written in simpler notation).**
>
> Thank you for this comment. We note that Reviewer bDJi raised a similar point regarding the readability of the paper, which we agree is an important aspect of our presentation. To address this, we will include a new figure illustrating our proposed meta-learning pipeline in an accessible way, along with additional text bridging the theoretical and empirical sections. Please see our response (A5) to Reviewer bDJi for the exact planned wording.
>
> **(Q4) Do you have insights for why Reptile outperforms on some tasks, given the theoretical guarantees?**
>
> Thank you for this question. Across all of our experiments, Reptile achieves the highest average performance only on the language modeling experiment for Task 1 using both adaptation mechanisms and for Task 6 using just last-layer adaptations. For our language modeling experiment, the test tasks are equipped with relatively few samples, leading to large performance variations across all methods. This leads to larger variation in the per-task performances, but when averaged over all test tasks, we observe that LoRA-ML comprehensively outperforms both Reptile and standard retraining (SR). Thus, these isolated instances where Reptile achieves better performance do not indicate a deeper trend.
>
> **(Q5) How did you conclude that "these spurious minima are almost never found in practice?" I do not think the presented experiments suffice for this claim IMO.**
>
> Thank you for bringing this point up. This claim is supported by our extensive synthetic data experiments described in Section 4, which test the LoRA-ML objective for linear models, the same setting as our theoretical analysis. We observe, over all experiments, including the additional ablations in Appendix C.1, that minimizing the LoRA-ML objective with vanilla gradient descent and then fine-tuning the recovered shared parameter estimate $ \\hat{A} $ consistently minimizes the test-task loss up to the noise floor. If the LoRA-ML objective converged to a spurious local minimum which did not correspond to the ground truth shared parameter $ A^\\ast $, then the test task performance would significantly suffer.
>
> We appreciate the reviewer's concern about this claim and will add plots that track the convergence of $ \\hat{A} $ to $ A^\\ast $ within our synthetic data experiments to further justify this claim. While we cannot upload new figures during the discussion period, we will include them in the revised version of the paper.
>
> **Concluding Remark**:
> We hope that our responses have adequately addressed your questions and are committed to addressing any additional questions you may have during the discussion phase.
>
> [1] Saunshi, N., et al. ``A sample complexity separation between non-convex and convex meta-learning." ICML 2020.
>
> [2] Collins, L., et al. ``Maml and anil provably learn representations." ICML 2022.
>
> [3] Thekumparampil, K., et al. ``Statistically and computationally efficient linear meta-representation learning." NeurIPS 2021.
>
> [4] Du, S., et al. ``Few-shot learning via learning the representation, provably." ICLR 2021.

---

> > ### Comment · Reviewer_eo3S · 2025-08-03
> >
> > Thank you for your thorough rebuttal. I look forward to improving the paper's readability.
> > I will keep my rating unchanged.

---

### Official Review · Reviewer_bDJi · 2025-07-03

**Clarity:** 2
**Significance:** 3
**Originality:** 3
**Rating:** 5
**Confidence:** 3

**Summary:**

This paper proposes a new meta-learning framework tailored for parameter-efficient fine-tuning. Even though they claim it can work with any PEFT method, the main focus is on LoRA. By providing theoretical results in a linear regression setting, the author shows that standard retraining cannot produce representations that are easily adaptable to new tasks. Their proposed framework on the other hand guarantees low-rank adaptability and lower prediction error. Experiments on synthetic data and small-scaled vision and language tasks support the theoretical claims.

**Questions:**

Suggestions:

-	This paper is not easy to read for practical and general ML audiences. A short paragraph bridging the theory and experiments and an overview figure of the pipeline can improve it.
-	Add computational complexity analysis
-	Showing large-scale benchmark experiments

**Ethical Concerns:**

["NO or VERY MINOR ethics concerns only"]

**Final Justification:**

The paper presents a novel idea supported by a solid theoretical foundation. The authors have addressed my concerns in their responses and plan to provide a revised version that includes additional results and discussion.

**Limitations:**

Yes

**Quality:**

3

**Strengths And Weaknesses:**

Strengths:

- The theoretical contribution is novel and strong. Showing that standard retraining is suboptimal and proposing a framework with provably better adaptation.
-	The problem formulation is well-defined with clear assumptions.
-	The theoretical analysis of the effect of the number of tasks on the proposed model is another strength of this paper. Especially unique recovery of the ground truth for three or more tasks.
-	Figure 1 comprehensively supports the theoretical results in the controlled linear setting.

Major issues:

-	The theoretical results are driven under linear regression setting with Gaussian noise. However, classification tasks on CIFAR-10 using an MLP-Mixer and language modeling tasks with RoBERTa involve nonlinear architectures and classification losses. The theoretical guarantees do not extend to these more complex settings.
-	The proposed framework requires solving a nested optimization problem. For each task, adapter parameters must be learned in the inner loop. This introduces computational overhead compared to standard retraining. However, the paper does not provide any analysis or empirical comparison of the computational cost, memory usage, or convergence time between the proposed method and the baselines.
-	Some results (for example, Table 1 - left) show that the performance of standard retraining is very close to the proposed method. Even in some cases better, if we consider the standard deviations. The empirical result contradicts the strong theoretical claims which predict that standard retraining should become significantly worse as the number of tasks increases. This suggests that the theoretical benefits may not transfer effectively to nonlinear and classification settings and authors should discuss it.
-	More ablation studies can add great value to the paper, such as more diverse parameter-efficient fine-tuning methods or empirical results on real-world large-scale benchmarks.

Minor issues:

-	Line 282, “task task”

---

> ### Author Rebuttal · Authors · 2025-07-31
>
> Dear Reviewer bDJi, thank you for highlighting the novelty and strength of our theoretical contribution, as well as the clarity of our formulation. We answer your remaining questions below.
>
> **(Q1) The theoretical results are derived under a linear regression setting with Gaussian noise...The theoretical guarantees do not extend to these more complex settings.**
>
> (A1): Thank you for raising this important point regarding the applicability of our theoretical guarantees to more complex model classes and a broader range of loss functions. Our goal is to develop a theoretically principled framework for preparing a model for efficient test-time adaptation using PEFT, and as you correctly note, our theoretical analysis focuses on model adaptation in the setting of linear models with squared $\ell_2$ error. This setting enabled strong theoretical guarantees, yet still gave rise to several counterintuitive phenomena and posed non-trivial technical challenges (please see our response (A4) to Reviewer qeBR for further discussion of these challenges). We agree that extending this framework to encompass more expressive architectures and diverse objectives, particularly those beyond regression, is a valuable direction for future work, especially as our empirical results demonstrate that the principles extend effectively to non-linear models and classification tasks.
>
> **(Q2) The proposed framework requires solving a nested optimization problem...the paper does not provide any analysis or empirical comparison of the computational cost, memory usage, or convergence time between the proposed method and the baselines.**
>
> (A2) Thank you for the thoughtful question regarding the computational overhead, memory usage, and convergence time of our method compared to standard retraining (SR).
>
> We first address the memory cost. We discuss the memory overhead of LoRA-ML in Appendix D.3, which arises from the additional adapter parameters. If we assume a model has $m$ layers, each parameterized by a $d \times d$ matrix, and uses rank-$k$ LoRA adapters, then SR uses $md^2$ parameters, while LoRA-ML retraining uses $m(d^2 + 2kdT)$ parameters (where $T$ is the number of tasks). Since $k(T+1) \ll d$ in our setting, the total parameter count of LoRA-ML remains $O(md^2)$ during retraining, so LoRA-ML does not incur any additional asymptotic memory complexity relative to SR. At the fine-tuning stage, both LoRA-ML and SR models require the same number of trainable parameters since standard LoRA is applied.
>
> In terms of the computational cost and nested optimization, we note that while our objective is naturally expressed in a nested form (Eqn. (5)), we emphasize that LoRA-ML differs from methods like MAML in a key way: the inner optimization is over lightweight adapter parameters, not full shared weights. This allows us to implement optimization methods for the LoRA-ML objective either via explicit inner/outer loops or through joint optimization over both the shared and task-specific parameters. In fact, we utilize this joint form in our theoretical setting as we start with the nested formulation in Eqn. (13) and instead analyze the equivalent joint formulation in Eqn. (14).
>
> As you mention, the LoRA-ML objective can be empirically optimized using either an outer/inner loop strategy or simple joint minimization, each incurring different computational cost. In our experiments, we explored both strategies. In the linear experiments, we used a structured inner/outer loop: 10 inner steps on task-specific adapters followed by 1 outer step on shared parameters, for 3000 outer epochs. We compared this to SR trained for 3000 steps, since the SR objective is convex and converges quickly. In the larger vision and language experiments, we adopted a joint optimization strategy for LoRA-ML and used the same number of epochs as SR, without explicit inner steps, achieving stronger performance with no additional training loops. We agree that this is an important design choice to highlight and will update the revised version of the paper to reflect this. Further exploration of the tradeoffs between joint optimization and inner/outer-loop strategies is an interesting direction for future work, particularly for improving the efficiency of LoRA-ML at scale.
>
> In terms of convergence time, while we do not provide a theoretical convergence rate comparison, our empirical results show that LoRA-ML achieves better performance within the same epoch budget as SR in large-scale settings. This suggests the convergence of LoRA-ML is competitive in practice.
>
> We will incorporate the discussion of memory usage, computational cost of different optimization strategies, and convergence time into the revised paper.
>
> **(Q3) Some results (for example, Table 1 - left) show that the performance...This suggests that the theoretical benefits may not transfer effectively to nonlinear and classification settings and authors should discuss it.**
>
> (A3) Thank you for highlighting this point. As you mention, even though our proposed method demonstrates stronger results than standard retraining (SR) across tasks and experiments as a whole, for some specific tasks, the performance gap can vary, especially when the performance variance is taken into account. This behavior is a function of two key characteristics of each experiment: (i) the complexity of the test task relative to the expressivity of the adaptation mechanism, and (ii) the number of available samples per task.
>
> In Table 1 (left), we test our framework through a vision task on CIFAR-10, where each task involves binary classification between two distinct classes. We employ an MLP-Mixer architecture and adapt to the test task after retraining using LoRA. We observe that the performance gain using LoRA-ML relative to SR is not drastic, and the performance variances of both methods are small. LoRA induces a fairly expressive adaptation mechanism for the relatively easier task of binary classification between CIFAR-10 classes, so it is able to recover from potentially suboptimal models recovered by SR during retraining. Further, since we have access to many samples per task, the performance variance of each method is small.
>
> In contrast, in Table 1 (right), we consider the same vision experiment except we adapt using just the last layer fine-tuning, which is significantly less expressive than LoRA. As such, we see a much wider gap between the performance of LoRA-ML and SR while still achieving small performance variance due to the large number of samples in CIFAR-10.
>
> Lastly, in Table 2, we present the performance of our framework through a language task on ConvAI2, a dataset of conversations between personas. Each task corresponds to predicting the dialogue continuation of a specific persona. Relative to CIFAR-10, this experiment is much more complex and is equipped with many fewer samples per task. This leads to a larger variance in per-task performance. However, when we compare the average performance when adapting to all 10 test tasks, we observe that LoRA-ML significantly outperforms SR, even with variances taken into account.
>
> **(Q4) More ablation studies can add great value to the paper...**
>
> (A4) Thank you for this suggestion. While we did not have time to incorporate additional PEFT methods or larger datasets during the rebuttal period, we include an ablation for the LoRA-ML method demonstrating how the choice of the adapter rank $k$ during retraining affects performance, as this is the sole additional hyperparameter of LoRA-ML relative to the standard retraining procedure. We test using our language modeling experiment, where we aim to learn the responses of different personas in the ConvAI2 dataset. In Table 2 of the paper, we show results using rank-8 adaptations for LoRA-ML along with other methods. Below, we show the effect of different ranks and include standard retraining followed by rank-$k$ LoRA (SR+LoRA) as a baseline.
>
> **Test Task Prediction Acc. (↑)**
> |Rank|1|4|8|12|16|
> |-|-|-|-|-|-|
> |LoRA-ML|0.37|0.36|0.39|0.31|0.37|
> |SR+LoRA|0.20|0.20|0.26|0.19|0.20|
>
> We report the mean accuracy over the 10 test tasks using 5 random trials. The results demonstrate that LoRA-ML outperforms SR across a variety of ranks, demonstrating the effectiveness of our framework and its insensitivity to the adapter rank hyperparameter. We will include these results in the revised version of the paper.
>
> **(Q5) This paper is not easy to read for practical and general ML audiences. A short paragraph bridging the theory and experiments, and an overview figure of the pipeline can improve it.**
>
> (A5) Thank you for this helpful suggestion. Readability is an important concern, and we have aimed to keep our exposition and theoretical statements as accessible as possible while maintaining technical precision. While we cannot upload new figures during the rebuttal phase, we appreciate your feedback and will add a pipeline diagram in the revised version of the paper, as well as the following text to better connect the theory and the experiments:
>
> ``In our theoretical analysis, we showed that minimizing a meta-learning-based retraining objective infused with LoRA (Eqn. 13) leads to provably stronger test-time adaptation compared to standard retraining for a class of linear models. Building on this, we evaluate our general framework for PEFT-based meta-learning (Eqn. 5) in more practical settings, using deeper and more complex architectures, both regression and classification tasks, and various PEFT methods beyond LoRA. Across these experiments, we find that incorporating the PEFT method directly into the retraining objective via our framework consistently improves downstream adaptation performance."
>
> **Concluding Remark:**
>
> We hope that our responses have adequately addressed your concerns. If so, we kindly request that you consider revising your score. We are committed to addressing any additional concerns you may have during the discussion phase.

---

> > ### Comment · Area_Chair_6kfd · 2025-08-05
> >
> > Dear Reviewer bDJi,
> >
> > The deadline of the Author-Reviewer discussion period is approaching. I would really appreciate if you can initiate the discussion with authors as soon as possible. Reviewers are expected to stay engaged in discussions, initiate them and respond to authors’ rebuttal, ask questions and listen to answers to help clarify remaining issues.
> >
> > If you have any difficulty in starting the discussion, please let us know.
> >
> > Sincerely,
> > AC

---

> > ### Comment · Reviewer_bDJi · 2025-08-07
> >
> > I thank the authors for their thoughtful and thorough responses to the review comments. Their responses sufficiently addressed my concerns. Therefore, I’ll revise my score accordingly as 5.

---

> > > ### Author Response · Authors · 2025-08-09
> > >
> > > Dear Reviewer bDJi,
> > >
> > > Thank you for engaging with us in the discussion period, we are glad our responses addressed your concerns.

---

### Decision · Program_Chairs · 2025-09-17

**Decision:**

Accept (poster)

**Comment:**

This paper theoretically analyzes a meta-learning problem setup based on low-rank adaptation and investigates conditions under which it outperforms simple multitask learning. While the scope of the theoretical analysis is limited to linear models, the claims are solid and the experimental results including beyond-linear setting support them well. The findings offer a principled perspective that spans meta-learning and foundation models.

Reviewer opinions are split around the borderline. In particular, qeBR gave a score of 3 and argued that the problem setup is overly simplified for theory. However, the authors’ response to this point appears sufficiently reasonable. No other major concerns are evident, and I judge that the paper’s theoretical contribution is substantial for the meta-learning community, so I recommend acceptance.